# RepLoRA: Reparameterizing Low-Rank Adaptation via the Perspective of Mixture of Experts

Tuan Truong [* 1]  Chau Nguyen [* 1 2]  Huy Nguyen [* 3]  Minh Le [1]  Trung Le [4]  Nhat Ho [3]

## Abstract

Low-rank Adaptation (LoRA) has emerged as a powerful method for fine-tuning large-scale foundation models. Despite its popularity, the theoretical understanding of LoRA has remained limited. This paper presents a theoretical analysis of LoRA by examining its connection to the Mixture of Experts models. Under this framework, we show that simple reparameterizations of the LoRA matrices can notably accelerate the low-rank matrix estimation process. In particular, we prove that reparameterization can reduce the data needed to achieve a desired estimation error from an exponential to a polynomial scale. Motivated by this insight, we propose *Reparameterized Low-Rank Adaptation* (RepLoRA), which incorporates lightweight MLPs to reparameterize the LoRA matrices. Extensive experiments across multiple domains demonstrate that RepLoRA consistently outperforms vanilla LoRA. Notably, with limited data, RepLoRA surpasses LoRA by a margin of up to **40.0%** and achieves LoRA's performance with only **30.0%** of the training data, highlighting both the theoretical and empirical robustness of our PEFT method.

## 1. Introduction

With the rapid growth in data availability and computational resources, large-scale models trained on extensive datasets have demonstrated remarkable generalization capabilities, enabling successful applications across language, vision, and multi-modal tasks (Dosovitskiy, 2021; Radford et al., 2021; Touvron et al., 2023). However, fully fine-tuning such

*Equal contribution [1]Qualcomm AI Research, Qualcomm Vietnam Company Limited [2]Work was completed while an employee at Qualcomm [3]The University of Texas at Austin, USA [4]Monash University, Australia. Correspondence to: Tuan Truong <tuantruo@qti.qualcomm.com>.

*Proceedings of the 42nd International Conference on Machine Learning*, Vancouver, Canada. PMLR 267, 2025. Copyright 2025 by the author(s).

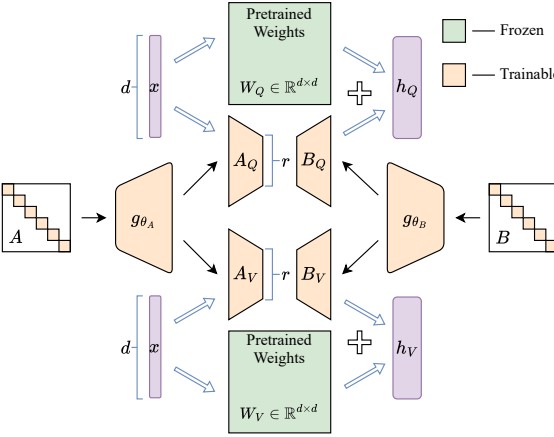

*Figure 1.* Overview of our proposed method RepLoRA, which reparameterizes the low-rank matrices as the output of a lightweight MLP, whose inputs are two diagonal matrices.

models for specific downstream tasks can be prohibitively expensive. To address this challenge, several parameter-efficient fine-tuning (PEFT) methods (Houlsby et al., 2019; Lester et al., 2021; Jia et al., 2022) have emerged, facilitating effective adaptation of large pre-trained models by adjusting a minimal set of parameters while keeping most of the backbone frozen. Among these methods, *Low-Rank Adaptation* (LoRA) (Hu et al., 2022) stands out for its simplicity and effectiveness and has been successfully applied across diverse domains (Li et al., 2022; Qin et al., 2023; Taori et al., 2023; Liu et al., 2024a). Despite its successes, theoretical understanding of LoRA has remained limited, hindering our ability to optimize its performance further.

Building on the recent finding (Le et al., 2024) about the connection between attention mechanism (Vaswani, 2017) and the mixture of experts (MoE) (Jacobs et al., 1991; Jordan & Jacobs, 1994) models, we present a rigorous theoretical study demonstrating how LoRA can be interpreted within this new framework. Leveraging this perspective, we show that a straightforward reparameterization technique (Li & Liang, 2021; Le et al., 2025), which represents low-rank matrices as the output of an MLP, can theoretically enhance the performance of LoRA. Specifically, our analysis reveals that this reparameterization can reduce the data needed to achieve a desired estimation error from an exponential

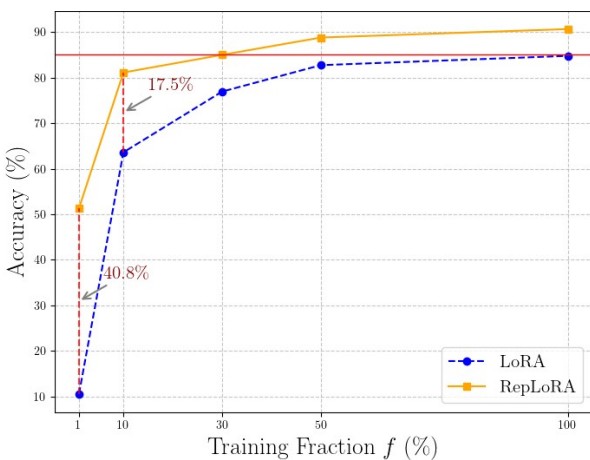

*Figure 2.* Sample Efficiency on FGVC Datasets. RepLoRA not only outperforms LoRA consistently but also achieves LoRA performance on a full dataset with only $f = 30\%$ training fraction.

scale to a polynomial scale, thereby substantially improving sample efficiency. Based on these insights, we introduce ***R***eparameterized ***Lo***w-***R***ank ***A***daptation (RepLoRA) - a novel PEFT method reparameterizes low-rank matrices through a lightweight MLP.

We conducted extensive experiments across multiple domains, including image, video, language, and multi-modal tasks. Our results indicate that RepLoRA consistently demonstrates better performance than vanilla LoRA. When only a tiny fraction of the training data is subsampled, RepLoRA improves up to **40%** over LoRA. This highlights the robustness and effectiveness of our method, both theoretically and empirically. Moreover, the MLP used for reparameterization can be discarded after training, ensuring that our approach remains as efficient as the standard counterparts at inference time.

**Contributions.** In summary, our contributions are: **(i)** We provide a rigorous theoretical analysis of LoRA from the perspective of a mixture of experts. **(ii)** Our results show that reparameterization can substantially improve sample efficiency, transitioning from an exponential rate to a polynomial rate. **(iii)** Building on these theoretical insights, we introduce RepLoRA, a novel PEFT approach that integrates reparameterization into LoRA. **(iv)** Extensive experiments across diverse domains demonstrate that RepLoRA consistently outperforms vanilla LoRA by a significant margin, thereby underscoring its effectiveness and robustness from theoretical and empirical perspectives.

**Organization.** The paper is organized as follows: Section 3 provides the background on LoRA and MoE. Section 4 establishes the connection between LoRA and MoE. Section 5 presents our theoretical analysis, including the statistical benefits of reparameterizing LoRA. Building on these in-

sights, Section 6 presents our method, RepLoRA. To demonstrate the effectiveness of RepLoRA, Section 7 presents the experimental results. Finally, Section 8 concludes the paper.

**Notation.** For any $n \in \mathbb{N}$, let $[n] := \{1, 2, \ldots, n\}$. For any set $S$, $|S|$ denotes its cardinality. Given a vector $u := (u_1, u_2, \ldots, u_d) \in \mathbb{R}^d$ and $\alpha := (\alpha_1, \alpha_2, \ldots, \alpha_d) \in \mathbb{N}^d$, we define $u^\alpha = u_1^{\alpha_1} u_2^{\alpha_2} \ldots u_d^{\alpha_d}$, $|u| := u_1 + u_2 + \ldots + u_d$ and $\alpha! := \alpha_1! \alpha_2! \ldots \alpha_d!$, while $\|u\|$ stands for its Euclidean norm. For positive sequences $(a_n)_{n \geq 1}$ and $(b_n)_{n \geq 1}$, we write $a_n = \mathcal{O}(b_n)$ or $a_n \lesssim b_n$ if there exists a constant $C > 0$ such that $a_n \leq Cb_n$ for all $n \in \mathbb{N}$. The notation $a_n = \mathcal{O}_P(b_n)$ indicates $a_n/b_n$ is stochastically bounded.

## 2. Related Works

**Parameter-Efficient Fine-tuning (PEFT).** With the recent rise of large models, PEFT methods are growing in popularity for their ability to fine-tune large-scale models by training a relatively small number of parameters for adapting to specific downstream tasks. Existing PEFT methods can be divided into three categories. The first category is referred to as *adapter-based* methods, which introduce additional trainable parameters to the frozen backbone. For example, *Series Adapter* (Houlsby et al., 2019) proposes adding linear modules in sequence to the existing layer, or *Parallel Adapter* (He et al., 2022) proposes integrating these modules in parallel. The second category of PEFT methods is *Prompt-based* methods that add extra trainable soft tokens, referred to as prompts, to the input (Lester et al., 2021; Razdaibiedina et al., 2023; Wang et al., 2023). A weakness of these methods is that they increase inference latency compared to the original model.

**Low-Rank Adaptation (Hu et al., 2022).** LoRA and its variants are among the third category of the PEFT method, which is well-known for its simplicity and for not adding extra inference burden. To fine-tune the linear layers of a large model, LoRA applies low-rank matrices to approximate the weight changes and then merges them with the pre-trained weights for inference. A recent variant of LoRA is DoRA (Liu et al., 2024b), which proposes to decompose the weight change into a learnable magnitude and directional component. Another example is AdaLoRA (Zhang et al., 2023), which parameterizes the incremental updates in the form of singular value decomposition and prunes less significant singular values for more efficient updates. Orthogonal Fine-tuning (OFT) (Qiu et al., 2023) exploits the orthogonal factorization to fine-tune diffusion models. Recently, VeRA (Kopiczko et al., 2024) significantly reduces the number of trainable parameters compared to LoRA by using learnable scaling vectors to adjust a shared pair of frozen random matrices across layers. Notably, our method, which will be proposed in the following few sections, also

falls within this category, and we validate its efficacy alongside LoRA and its variants through theoretical analysis and comprehensive experimentation.

**Mixture of Experts.** Expanding upon the foundational principles of mixture models (Jacobs et al., 1991; Jordan & Jacobs, 1994), earlier research (Eigen et al., 2014; Shazeer et al., 2017) established the Mixture of Experts (MoE) layer as a crucial mechanism for effectively scaling model capacity. Over time, MoE models have gained significant recognition due to their versatility across multiple domains, including large language models (Du et al., 2022; Zhou et al., 2023), computer vision (Riquelme et al., 2021; Puigcerver et al., 2024), multimodal learning (Han et al., 2024), and multi-task learning (Ma et al., 2018). Recent studies have focused on analyzing the convergence rates of expert estimation in MoE models, exploring different assumptions and configurations related to gating mechanisms and expert functions. For instance, in the context of softmax gating, Nguyen et al. (2023; 2024a;b) revealed that the expert estimation rates are shaped by the solvability of polynomial systems arising from interactions between gating and expert parameters. More recently, Nguyen et al. (2024d;c; 2025) employed least squares estimation to establish an identifiability condition for expert functions, particularly in feedforward networks with nonlinear activation functions. These findings indicate that under these conditions, estimation rates improve significantly compared to models relying on polynomial experts.

## 3. Preliminaries

This section briefly reviews the background for multi-head self-attention in transformers, low-rank adaptation, and a mixture of expert models.

**Multi-head Self-attention.** We begin by revisiting the architecture of the multi-head self-attention (MSA) layer in Transformer (Vaswani, 2017; Dosovitskiy, 2021). Let $\boldsymbol{X} = [\boldsymbol{x}_1, \ldots, \boldsymbol{x}_N]^\top \in \mathbb{R}^{N \times d}$ denote an input sequence of embeddings, where $N$ is the sequence length and $d$ denotes the embedding dimension. The MSA layer processes this sequence as follows:

$$\text{MSA}(\boldsymbol{X}_Q, \boldsymbol{X}_K, \boldsymbol{X}_V) = \text{Concat}(\boldsymbol{h}_1, ..., \boldsymbol{h}_m)\boldsymbol{W}^O, \quad (1)$$

where each attention head is defined by $\boldsymbol{h}_i = \text{Attention}(\boldsymbol{X}\boldsymbol{W}_i^Q, \boldsymbol{X}\boldsymbol{W}_i^K, \boldsymbol{X}\boldsymbol{W}_i^V)$ for $i \in [m]$. Here, $\boldsymbol{X}_Q = (\boldsymbol{X}\boldsymbol{W}_1^Q, \ldots, \boldsymbol{X}\boldsymbol{W}_m^Q)$, $\mathbb{X}_K = (\boldsymbol{X}\boldsymbol{W}_1^K, \ldots, \boldsymbol{X}\boldsymbol{W}_m^K)$, and $\mathbb{X}_V = (\boldsymbol{X}\boldsymbol{W}_1^V, \ldots, \boldsymbol{X}\boldsymbol{W}_m^V)$ are the query, key, and value matrices, respectively. Furthermore, $m$ is the number of heads, and $\boldsymbol{W}^O \in \mathbb{R}^{md_v \times d}$ is the output projection matrix. Each attention head $\boldsymbol{h}_i$ is parameterized by $\boldsymbol{W}_i^Q \in \mathbb{R}^{d \times d_k}, \boldsymbol{W}_i^K \in \mathbb{R}^{d \times d_k}$, and $\boldsymbol{W}_i^V \in \mathbb{R}^{d \times d_v}$, with

$d_k = d_v = \frac{d}{m}$.

**Low-Rank Adaptation.** LoRA (Hu et al., 2022) has emerged as an efficient method for adapting large pretrained transformer models to downstream tasks. Building upon the hypothesis that the updates during fine-tuning exhibit a low "intrinsic rank", LoRA proposes to fine-tune the transformer architectures' linear layers by incrementally updating the pre-trained weights with the product of two low-rank matrices. For a given pre-trained weight matrix $\boldsymbol{W}_0 \in \mathbb{R}^{m \times n}$, LoRA represents its update as $\Delta \boldsymbol{W} = \boldsymbol{B}\boldsymbol{A}$, where $\boldsymbol{B} \in \mathbb{R}^{m \times r}$, and $\boldsymbol{A} \in \mathbb{R}^{r \times n}$ with $r \ll \min\{m, n\}$. Consequently, the output of the fine-tuned model is:

$$\hat{\boldsymbol{y}} = \boldsymbol{W}'\boldsymbol{x} = \boldsymbol{W}_0\boldsymbol{x} + \boldsymbol{B}\boldsymbol{A}\boldsymbol{x}. \quad (2)$$

During training, $\boldsymbol{W}_0$ remains fixed, while $\boldsymbol{A}$ and $\boldsymbol{B}$ are updated. Typically, LoRA adjusts the linear layers that generate transformer models' queries and values (or keys, queries, and values). In line with prior work (Hu et al., 2022; Liu et al., 2024b; Xin et al., 2024), here we fine-tune the query and value projection matrices, leading to the following output expression:

$$f_{\text{LoRA}}(\boldsymbol{X}; \boldsymbol{A}, \boldsymbol{B}) = \text{Concat}(\widetilde{\boldsymbol{h}}_1, \cdots, \widetilde{\boldsymbol{h}}_m)\boldsymbol{W}^O, \quad (3)$$

where for each $i \in [m]$, $\widetilde{\boldsymbol{h}}_i = \text{Attention}(\boldsymbol{X}\boldsymbol{W}_i^Q + \boldsymbol{X}\boldsymbol{B}_{Q,i}\boldsymbol{A}_{Q,i}, \boldsymbol{X}\boldsymbol{W}_i^K, \boldsymbol{X}\boldsymbol{W}_i^V + \boldsymbol{X}\boldsymbol{B}_{V,i}\boldsymbol{A}_{V,i})$. Here, we denote $\boldsymbol{A} = [\boldsymbol{A}_Q, \boldsymbol{A}_V]$, and $\boldsymbol{B} = [\boldsymbol{B}_Q, \boldsymbol{B}_V]$, where $\boldsymbol{A}_Q = (\boldsymbol{A}_{Q,1}, \ldots, \boldsymbol{A}_{Q,m})$, and $\boldsymbol{A}_V = (\boldsymbol{A}_{V,1}, \ldots, \boldsymbol{A}_{V,m})$. Likewise, $\boldsymbol{B}_Q = (\boldsymbol{B}_{Q,1}, \ldots, \boldsymbol{B}_{Q,m})$, and $\boldsymbol{B}_V = (\boldsymbol{B}_{V,1}, \ldots, \boldsymbol{B}_{V,m})$. For each head $i \in [m]$, the dimensions of these matrices are $\boldsymbol{A}_{Q,i} \in \mathbb{R}^{r \times d_k}, \boldsymbol{B}_{Q,i} \in \mathbb{R}^{d \times r}, \boldsymbol{A}_{V,i} \in \mathbb{R}^{r \times d_v}$, and $\boldsymbol{B}_{V,i} \in \mathbb{R}^{d \times r}$.

**Mixture of Experts.** A mixture of experts (MoE) (Jacobs et al., 1991; Jordan & Jacobs, 1994) model consists of $N$ expert networks, $f_i : \mathbb{R}^d \to \mathbb{R}^{d_v}$ for $i \in [N]$, and a gating function $G : \mathbb{R}^d \to \mathbb{R}^N$ that allocates contributions of each expert based on the input $\boldsymbol{x}$ to the model. The output of the MoE model is given by:

$$\hat{\boldsymbol{y}} = \sum_{i=1}^{N} G(\boldsymbol{x})_i \cdot f_i(\boldsymbol{x}),$$

where $G(\boldsymbol{x}) = \text{softmax}(s_1(\boldsymbol{x}), \cdots, s_N(\boldsymbol{x}))$, and $s_i : \mathbb{R}^d \to \mathbb{R}$ is a score function. In the subsequent sections, we discuss how MoE relates to LoRA and provide a theoretical analysis of the proposed method.

## 4. LoRA from the perspective of MoE

Prior work (Le et al., 2024; 2025) has shown that each attention head in the MSA layer can be viewed as a specialized architecture of multiple MoE models. Specifically, from Eq. (1), consider the output of the $l$-th head $\boldsymbol{h}_l =$

$[\boldsymbol{h}_{l,1}, \ldots, \boldsymbol{h}_{l,N}]^\top \in \mathbb{R}^{N \times d_v}$. Let $\mathbb{X} = [\boldsymbol{x}_1^\top, \ldots, \boldsymbol{x}_N^\top]^\top \in \mathbb{R}^{Nd}$ denote the concatenated input embeddings. We then define $N$ experts $f_j : \mathbb{R}^{Nd} \to \mathbb{R}^{d_v}$ encoded within the MSA layer as follows:

$$f_j(\mathbb{X}) = \boldsymbol{W}_l^{V^\top} \boldsymbol{E}_j \mathbb{X} = \boldsymbol{W}_l^{V^\top} \boldsymbol{x}_j, \qquad (4)$$

for $j \in [N]$, where the matrix $\boldsymbol{E}_j \in \mathbb{R}^{d \times Nd}$ is such that $\boldsymbol{E}_j \mathbb{X} = \boldsymbol{x}_j$. Next, we introduce $N \times N$ score functions $s_{i,j} : \mathbb{R}^{Nd} \to \mathbb{R}$ associated with these experts:

$$s_{i,j}(\mathbb{X}) = \frac{\mathbb{X}^\top \boldsymbol{E}_i^\top \boldsymbol{W}_l^Q \boldsymbol{W}_l^{K^\top} \boldsymbol{E}_j \mathbb{X}}{\sqrt{d_v}} = \frac{\boldsymbol{x}_i^\top \boldsymbol{W}_l^Q \boldsymbol{W}_l^{K^\top} \boldsymbol{x}_j}{\sqrt{d_v}}, \qquad (5)$$

for $i \in [N]$ and $j \in [N]$. Consequently, each output vector $\boldsymbol{h}_{l,i}$ can be formulated as the result of an MoE model, utilizing the experts and score functions defined above:

$$\boldsymbol{h}_{l,i} = \sum_{j=1}^N \frac{\exp(s_{i,j}(\mathbb{X}))}{\sum_{k=1}^N \exp(s_{i,k}(\mathbb{X})))} \cdot f_j(\mathbb{X}). \qquad (6)$$

Within a pre-trained MSA layer, all parameters of these experts and score functions $\boldsymbol{W}_l^Q$, $\boldsymbol{W}_l^K$, and $\boldsymbol{W}_l^V$ remain fixed. When LoRA is applied, it refines these experts and score functions with low-rank updates:

$$\tilde{f}_j(\mathbb{X}) = (\boldsymbol{W}_l^V + \boldsymbol{B}_{V,l} \boldsymbol{A}_{V,l})^\top \boldsymbol{E}_j \mathbb{X}, \qquad (7)$$

$$\tilde{s}_{i,j}(\mathbb{X}) = \frac{\mathbb{X}^\top \boldsymbol{E}_i^\top (\boldsymbol{W}_l^Q + \boldsymbol{B}_{Q,l} \boldsymbol{A}_{Q,l}) \boldsymbol{W}_l^{K^\top} \boldsymbol{E}_j \mathbb{X}}{\sqrt{d_v}}, \qquad (8)$$

for $i \in [N]$ and $j \in [N]$. From these definitions and Eq. (3), the modified output of the $l$-th head $\tilde{\boldsymbol{h}}_l = [\tilde{\boldsymbol{h}}_{l,1}, \ldots, \tilde{\boldsymbol{h}}_{l,N}]^\top \in \mathbb{R}^{N \times d_v}$ can be expressed as:

$$\tilde{\boldsymbol{h}}_{l,i} = \sum_{j=1}^N \frac{\exp(\tilde{s}_{i,j}(\mathbb{X}))}{\sum_{k=1}^N \exp(\tilde{s}_{i,k}(\mathbb{X})))} \cdot \tilde{f}_j(\mathbb{X}). \qquad (9)$$

From this perspective, LoRA effectively fine-tunes the pre-trained MoE models contained within each MSA head by incorporating low-rank modifications to both the expert and the score functions. Next section will leverage this MoE viewpoint to analyze the theoretical properties of LoRA.

# 5. Theoretical Analysis of LoRA: With and Without Reparameterization

This section presents the theoretical benefits of applying the reparameterization technique in LoRA via its connection to MoE as formulated in Section 4. For simplicity, we will take into account only the first row of the first attention head $\tilde{\boldsymbol{h}}_{1,1}$ specified in Eq. (9). Under this simplified setting, we will investigate the convergence behavior of low-rank matrices within the following MoE-based regression framework:

**Problem setup.** Let $(\mathbb{X}_1, \boldsymbol{Y}_1), (\mathbb{X}_2, \boldsymbol{Y}_2), \ldots, (\mathbb{X}_n, \boldsymbol{Y}_n) \in \mathbb{R}^{\bar{d}} \times \mathbb{R}^{\bar{d}}$ be i.i.d. samples of size $n$ generated from the following regression model:

$$\boldsymbol{Y}_i = f_{G_*}(\mathbb{X}_i) + \varepsilon_i, \quad i = 1, 2, \ldots, n. \qquad (10)$$

Above, we assume that $\mathbb{X}_1, \mathbb{X}_2, \ldots, \mathbb{X}_n$ are i.i.d. samples from some probability distribution $\mu$ with bounded support. Meanwhile, $\varepsilon_1, \varepsilon_2, \ldots, \varepsilon_n$ are independent Gaussian noise variables such that $\mathbb{E}[\varepsilon_i|\mathbb{X}_i] = 0$ and $\mathrm{Var}(\varepsilon_i|\mathbb{X}_i) = \nu^2 I_{\bar{d}}$ for all $i \in [n]$. Next, the regression function $f_{G_*}(\cdot)$ takes the form of an MoE model with $L$ unknown experts, that is,

$$f_{G_*}(\mathbb{X}) := \sum_{j=1}^L \frac{\exp(\mathbb{X}^\top (\boldsymbol{M}_Q^0 + \boldsymbol{B}_{Q,j}^* \boldsymbol{A}_{Q,j}^*) \boldsymbol{M}_K^0 \mathbb{X} + c_j^*)}{D_f(\mathbb{X})}$$
$$\cdot (\boldsymbol{M}_V^0 + \boldsymbol{B}_{V,j}^* \boldsymbol{A}_{V,j}^*) \mathbb{X}, \qquad (11)$$

where we denote

$$D_f(\mathbb{X}) := \sum_{k=1}^L \exp(\mathbb{X}^\top (\boldsymbol{M}_Q^0 + \boldsymbol{B}_{Q,k}^* \boldsymbol{A}_{Q,k}^*) \boldsymbol{M}_K^0 \mathbb{X} + c_k^*),$$

while $G_* := \sum_{j'=1}^L \exp(c_{j'}^*) \delta_{(\boldsymbol{B}_{Q,j'}^*, \boldsymbol{A}_{Q,j'}^*, \boldsymbol{B}_{V,j'}^*, \boldsymbol{A}_{V,j'}^*)}$ represents for a *mixing measure*, that is, a combination of Dirac measures $\delta$, associated with unknown parameters $(c_{j'}^*, \boldsymbol{B}_{Q,j'}^*, \boldsymbol{A}_{Q,j'}^*, \boldsymbol{B}_{V,j'}^*, \boldsymbol{A}_{V,j'}^*)_{j'=1}^L$ in the compact parameter space $\Theta \subset \mathbb{R} \times \mathbb{R}^{\bar{d} \times r} \times \mathbb{R}^{r \times \bar{d}} \times \mathbb{R}^{\bar{d} \times r} \times \mathbb{R}^{r \times \bar{d}}$. In addition, we assume that the matrices $\boldsymbol{M}_Q^0 \in \mathbb{R}^{\bar{d} \times \bar{d}}$, $\boldsymbol{M}_K^0 \times \mathbb{R}^{\bar{d} \times \bar{d}}$, and $\boldsymbol{M}_V^0 \in \mathbb{R}^{\bar{d} \times \bar{d}}$ are given to align with the formulation in Eq. (9).

**With versus Without Reparametrization.** Subsequently, we establish the convergence rates of estimating the unknown low-rank matrices $\{\boldsymbol{B}_{Q,j'}^*, \boldsymbol{A}_{Q,j'}^*, \boldsymbol{B}_{V,j'}^*, \boldsymbol{A}_{V,j'}^*\}_{j'=1}^L$ under two scenarios, namely without shared structures among these low-rank matrices (equivalently, without reparametrization) in Section 5.1 and with shared structures among these low-rank matrices (equivalently, with reparametrization) in Section 5.2. Our ultimate goal is to demonstrate that sample efficiency of estimating these low-rank matrices under the shared structures setting is much better than that under the non-shared structures setting with a given error $\epsilon > 0$. The theory sheds light on our design of Reparameterized LoRA (RepLoRA) in Section 6.

## 5.1. Without Reparametrization: Suboptimal Sample Complexity

We begin our convergence analysis for low-rank matrices with the scenario where the LoRA reparametrization is absent. It is worth noting that we can estimate those unknown matrices via estimating the ground-truth mixing measure $G_*$, For that sake, we utilize the least square method (van de

Geer, 2000) to obtain the following estimator:

$$\widehat{G}_n \in \arg \min_{G \in \mathcal{G}_{L'}(\Theta)} \sum_{i=1}^{n} \left( \boldsymbol{Y}_i - f_G(\mathbb{X}_i) \right)^2, \quad (12)$$

where we denote by $\mathcal{G}_{L'}(\Theta) := \{G = \sum_{j'=1}^{\ell} \exp(c_{j'}) \delta_{(\boldsymbol{B}_{Q,j'}, \boldsymbol{A}_{Q,j'}, \boldsymbol{B}_{V,j'}, \boldsymbol{A}_{V,j'})} : 1 \leq \ell \leq L', (c_{j'}, \boldsymbol{B}_{Q,j'}, \boldsymbol{A}_{Q,j'}, \boldsymbol{B}_{V,j'}, \boldsymbol{A}_{V,j'}) \in \Theta\}$ the set of all mixing measures with at most $L'$ atoms. As the number of ground-truth experts $L$ is typically unknown in practice, we assume that the number of fitted experts $L'$ is large enough such that $L' > L$. Then, to determine the convergence rates of the estimator $\widehat{G}_n$, we use a loss function built upon the concept of Voronoi cells (Manole & Ho, 2022).

**Voronoi loss.** Given a mixing measure $G$ with $L' > L$ atoms, its Voronoi cell set $\{\mathcal{V}_j \equiv \mathcal{V}_j(G) : j \in [L]\}$ is generated by the atoms of $G_*$ as follows:

$$\mathcal{V}_j := \{i \in [L'] : \|\boldsymbol{H}_i - \boldsymbol{H}_j^*\| \leq \|\boldsymbol{H}_i - \boldsymbol{H}_\ell^*\|, \forall \ell \neq j\},$$

where $\boldsymbol{H} := (\boldsymbol{B}_Q, \boldsymbol{A}_Q, \boldsymbol{B}_V, \boldsymbol{A}_V)$. Then, the Voronoi loss function used for this section is defined as

$$\mathcal{D}_{1,r}(G, G_*) := \sum_{j'=1}^{L} \left| \sum_{i \in \mathcal{V}_{j'}} \exp(c_i) - \exp(c_{j'}^*) \right|$$

$$+ \sum_{j'=1}^{L} \sum_{i \in \mathcal{V}_{j'}} \exp(c_i)(\|\Delta\boldsymbol{B}_{Q,ij'}\|^r + \|\Delta\boldsymbol{A}_{Q,ij'}\|^r$$

$$+ \|\Delta\boldsymbol{B}_{V,ij'}\|^r + \|\Delta\boldsymbol{A}_{V,ij'}\|^r),$$

for $r \in \mathbb{N}$, where we denote $\Delta\boldsymbol{H}_{ij'} := \boldsymbol{H}_i - \boldsymbol{H}_{j'}^*$ for any $i, j'$. Now, we are ready to study the sample efficiency of LoRA without reparametrization in Theorem 5.1 whose proof is deferred to Appendix A.1.

**Theorem 5.1.** *The following bound holds for any $r \in \mathbb{N}$:*

$$\sup_{G \in \mathcal{G}_{L'}(\Theta) \setminus \mathcal{G}_{L-1}(\Theta)} \mathbb{E}_{f_G}[\mathcal{D}_{1,r}(\widehat{G}_n, G)] \gtrsim \frac{1}{\sqrt{n}},$$

*where $\mathbb{E}_{f_G}$ denotes the expectation taken with respect to the product measure $f_G^n$.*

Let us denote $\widehat{G}_n = \sum_{i=1}^{L_n} \exp(\hat{c}_i^n) \delta_{(\hat{\boldsymbol{B}}_{Q,i}^n, \hat{\boldsymbol{A}}_{Q,i}^n, \hat{\boldsymbol{B}}_{V,i}^n, \hat{\boldsymbol{A}}_{V,i}^n)}$. Then, it follows from the result of Theorem 5.1 and the formulation of the loss $\mathcal{D}_{1,r}$ that the convergence rates of the low-rank matrix estimators $\hat{\boldsymbol{B}}_{Q,i}^n, \hat{\boldsymbol{A}}_{Q,i}^n, \hat{\boldsymbol{B}}_{V,i}^n, \hat{\boldsymbol{A}}_{V,i}^n$ are slower than any polynomial rates $\mathcal{O}_P(n^{-1/2r})$ for $r \in \mathbb{N}$. Thus, these rates could become as slow as $\mathcal{O}_P(1/\log^\tau(n))$ for some constant $\tau > 0$ (due to the inequality $\log(n) < n$). As a consequence, we need an exponential number of data $\mathcal{O}(\exp(\epsilon^{-1/\tau}))$ to obtain the approximations of the low-rank matrices with an error $\epsilon$. This observation reflects the suboptimality of the sample complexity of the LoRA without applying the reparametrization technique.

## 5.2. With Reparametrization: Optimal Sample Complexity

In this section, we consider the scenario where the low-rank matrices share their structures with each other. In particular, we consider the case where the low-rank matrices are reparameterized as:

$$\boldsymbol{A}_Q = \boldsymbol{A}_V = \varphi_1(\boldsymbol{A}) \qquad \boldsymbol{B}_Q = \boldsymbol{B}_V = \varphi_2(\boldsymbol{B}),$$

where $\varphi_1 : \mathbb{R}^{m \times m} \to \mathbb{R}^{r \times \bar{d}}$, $\varphi_2 : \mathbb{R}^{m \times m} \to \mathbb{R}^{r \times \bar{d}}$ are some functions, $\boldsymbol{A} \in \mathbb{R}^{m \times m}, \boldsymbol{B} \in \mathbb{R}^{m' \times m'}$ are learnable matrices with given dimensions $m, m' \geq 1$. We specifically note that, for the simplicity of the theoretical development, this formulation is simplified compared to what was used in practice because we set the low-rank matrices of queries to be equal to that of values. As we will show in this section, even with this simplified formulation, reparameterization gives superior sample complexity compared to vanilla LoRA without reparameterization.

After training, the reparameterization can be discarded, and only the low-rank matrices $\boldsymbol{A}_Q, \boldsymbol{A}_V, \boldsymbol{B}_Q, \boldsymbol{B}_V$ need to be stored. We observe that the reparameterization strategy implicitly encodes a shared structure between the query and value low-rank matrices. The primary difference compared to the original LoRA is that instead of learning the low-rank matrices separately, we reparameterize those matrices as the output of the two shared structures $\varphi_1, \varphi_2$. To study the theoretical advantages of the reparametrization technique, we focus on the following two settings of the functions $\varphi_1$ and $\varphi_2$:

*(i) Simple linear reparametrization:* $\varphi_1(\boldsymbol{A}) = \boldsymbol{W}_1\boldsymbol{A}$ and $\varphi_2(\boldsymbol{B}) = \boldsymbol{W}_2\boldsymbol{B}$.

*(ii) Non-linear reparametrization:* $\varphi_1(\boldsymbol{A}) = \sigma_1(\boldsymbol{W}_1\boldsymbol{A})$ and $\varphi_2(\boldsymbol{B}) = \sigma_2(\boldsymbol{W}_2\boldsymbol{B})$, where $\sigma_1$ and $\sigma_2$ are two non-linear activation functions applied element-wise to the matrices $\boldsymbol{W}_1\boldsymbol{A}$ and $\boldsymbol{W}_2\boldsymbol{B}$.

It should be noted that the above reparametrization settings can totally be generalized to the scenarios where the matrices $\boldsymbol{A}_Q$ and $\boldsymbol{A}_V$ share only the learnable matrix $\boldsymbol{A}$. In particular, we can reformulate those matrices as $\boldsymbol{A}_Q = \boldsymbol{W}_{Q,1}A$ and $\boldsymbol{A}_V = \boldsymbol{W}_{V,1}A$ for the simple linear reparametrization and as $\boldsymbol{A}_Q = \sigma_1(\boldsymbol{W}_{Q,1}\boldsymbol{A})$ and $\boldsymbol{A}_V = \sigma_1(\boldsymbol{W}_{V,1}\boldsymbol{A})$ for the non-linear reparametrization. In order to tailor to these settings, it is necessary to include several terms involving parameters $\boldsymbol{W}_{Q,1}$ and $\boldsymbol{W}_{V,1}$ rather than merely parameters $\boldsymbol{W}_1$ as in the above settings. However, we realize that these terms not only provide no additional information on our convergence analysis but also make it unnecessarily complicated. Therefore, we assume without loss of generalization that $\boldsymbol{A}_Q = \boldsymbol{A}_V = \boldsymbol{W}_1\boldsymbol{A}$ or $\boldsymbol{A}_Q = \boldsymbol{A}_V = \sigma_1(\boldsymbol{W}_1\boldsymbol{A})$ to simplify the analysis, making it more accessible.

### 5.2.1. SIMPLE LINEAR REPARAMETERIZATION

We first take into account the simple linear reparametrization where $\boldsymbol{A}_Q = \boldsymbol{A}_V = \boldsymbol{W}_1 \boldsymbol{A}$ and $\boldsymbol{B}_Q = \boldsymbol{B}_V = \boldsymbol{W}_2 \boldsymbol{B}$. Under this setting, the ground-truth regression function in Eq. (11), which we denote now as $f_{\bar{G}_*}(\mathbb{X})$ to avoid confusion, takes the following form:

$$\sum_{j=1}^{L} \frac{\exp(\mathbb{X}^\top (\boldsymbol{M}_Q^0 + \boldsymbol{W}_{2,j}^* \boldsymbol{B}_j^* \boldsymbol{W}_{1,j}^* \boldsymbol{A}_j^*) \boldsymbol{M}_K^0 \mathbb{X} + c_j^*)}{\bar{D}_f(\mathbb{X})}$$
$$\cdot (\boldsymbol{M}_V^0 + \boldsymbol{W}_{2,j}^* \boldsymbol{B}_j^* \boldsymbol{W}_{1,j}^* \boldsymbol{A}_j^*) \mathbb{X}, \quad (13)$$

where we denote $\bar{D}_f(\mathbb{X}) = \sum_{k=1}^{L} \exp(\mathbb{X}^\top (\boldsymbol{M}_Q^0 + \boldsymbol{W}_{2,k}^* \boldsymbol{B}_k^* \boldsymbol{W}_{1,k}^* \boldsymbol{A}_k^*) \boldsymbol{M}_K^0 \mathbb{X} + c_k^*)$, while the mixing measure is of the form $\bar{G}_* := \sum_{j'=1}^{L} \exp(c_{j'}^*) \delta_{\boldsymbol{W}_{2,j'}^* \boldsymbol{B}_{j'}^*, \boldsymbol{W}_{1,j'}^* \boldsymbol{A}_{j'}^*}$. Similar to Section 5.1, we estimate the unknown low-rank matrices via estimating the ground-truth mixing measure $\bar{G}_*$ using the least square method:

$$\bar{G}_n \in \arg \min_{\bar{G} \in \bar{\mathcal{G}}_{L'}(\Theta)} \sum_{i=1}^{n} \left( \boldsymbol{Y}_i - f_{\bar{G}}(\mathbb{X}_i) \right)^2, \quad (14)$$

where $\bar{\mathcal{G}}_{L'}(\Theta) := \{ G = \sum_{i=1}^{\ell} \exp(c_i) \delta_{\boldsymbol{W}_{2,i} \boldsymbol{B}_i \boldsymbol{W}_{1,i} \boldsymbol{A}_i} : 1 \leq \ell \leq L', (c_i, \boldsymbol{W}_{2,i}, \boldsymbol{B}_i, \boldsymbol{W}_{1,i}, \boldsymbol{A}_i) \in \Theta \}$ stands for the mixing measure set. To capture the convergence behavior of the estimator, we use the following Voronoi loss tailored to the simple linear reparameterization setting given by

$$\mathcal{D}_2(\bar{G}, \bar{G}_*) := \sum_{j'=1}^{L} \Big| \sum_{i \in \mathcal{V}_{j'}} \exp(c_i) - \exp(c_{j'}^*) \Big|$$
$$+ \sum_{j' \in [L]:|\mathcal{V}_{j'}|=1} \sum_{i \in \mathcal{V}_{j'}} \exp(c_i) \| \boldsymbol{Z}_i - \boldsymbol{Z}_{j'}^* \|$$
$$+ \sum_{j' \in [L]:|\mathcal{V}_{j'}|>1} \sum_{i \in \mathcal{V}_{j'}} \exp(c_i) \| \boldsymbol{Z}_i - \boldsymbol{Z}_{j'}^* \|^2,$$

where we denote $\boldsymbol{Z} := \boldsymbol{W}_2 \boldsymbol{B} \boldsymbol{W}_1 \boldsymbol{A}$. Given the above loss function, we study the sample efficiency of the LoRA with simple linear reparametrization in Theorem 5.2.

**Theorem 5.2.** *The estimator $\bar{G}_n$ converges to the true mixing measure $\bar{G}_*$ at the following rate:*

$$\mathcal{D}_2(\bar{G}_n, \bar{G}_*) = \mathcal{O}_P(\sqrt{\log(n)/n}).$$

Proof of Theorem 5.2 is in Appendix A.2. The above bound together with the construction of the loss $\mathcal{D}_2$ indicates that the convergence rates of estimating low-rank matrices $\boldsymbol{W}_{2,j'}^* \boldsymbol{B}_{j'}^* \boldsymbol{W}_{1,j'}^* \boldsymbol{A}_{j'}^*$, for $j' \in [L]$, range from order $\mathcal{O}_P([\log(n)/n]^{\frac{1}{2}})$ to order $\mathcal{O}_P([\log(n)/n]^{\frac{1}{4}})$. Thus, it costs at most a polynomial number of data $\mathcal{O}(\epsilon^{-4})$ to approximate those low-rank matrices with the error $\epsilon$. Compared to the exponential number of data required in the LoRA without reparametrization in Section 5.1, we observe that the LoRA with linear reparametrization is much more sample efficient.

### 5.2.2. NON-LINEAR REPARAMETERIZATION

Next, we draw our attention to the LoRA with non-linear reparametrization where the low-rank matrices are parametrized as $\boldsymbol{A}_Q = \boldsymbol{A}_V = \sigma_1(\boldsymbol{W}_1 \boldsymbol{A})$ and $\boldsymbol{B}_Q = \boldsymbol{B}_V = \sigma_2(\boldsymbol{W}_2 \boldsymbol{B})$. Then, the ground-truth regression function in Eq.(11), denoted as $f_{\widetilde{G}_*}(\mathbb{X})$ in this section, admits the following form:

$$\sum_{j=1}^{N} \frac{\exp(\mathbb{X}^\top (\boldsymbol{M}_Q^0 + \sigma_2(\boldsymbol{W}_{2,j}^* \boldsymbol{B}_j^*) \sigma_1(\boldsymbol{W}_{1,j}^* \boldsymbol{A}_j^*)) \boldsymbol{M}_K^0 \mathbb{X} + c_j^*)}{\widetilde{D}_f(\mathbb{X})}$$
$$\cdot (\boldsymbol{M}_{V,j}^0 + \sigma_2(\boldsymbol{W}_{2,j}^* \boldsymbol{B}_j^*) \sigma_1(\boldsymbol{W}_{1,j}^* \boldsymbol{A}_j^*)) \mathbb{X}, \quad (15)$$

where we denote $\widetilde{D}_f(\mathbb{X}) := \sum_{k=1}^{N} \exp(\mathbb{X}^\top (\boldsymbol{M}_Q^0 + \sigma_2(\boldsymbol{W}_{2,k}^* \boldsymbol{B}_k^*) \sigma_1(\boldsymbol{W}_{1,k}^* \boldsymbol{A}_k^*)) \boldsymbol{M}_K^0 \mathbb{X} + c_k^*)$ and the mixing measure $\widetilde{G}_* := \sum_{j'=1}^{L} \exp(c_{j'}^*) \delta_{(\boldsymbol{W}_{2,j'}^* \boldsymbol{B}_{j'}^*, \boldsymbol{W}_{1,j'}^* \boldsymbol{A}_{j'}^*)}$ associated with unknown parameters $(c_{j'}^*, \boldsymbol{W}_{2,j'}^* \boldsymbol{B}_{j'}^*, \boldsymbol{W}_{1,j'}^* \boldsymbol{A}_{j'}^*)_{j'=1}^{L}$ in the parameter space $\Theta \subset \mathbb{R} \times \mathbb{R}^{d \times r} \times \mathbb{R}^{r \times d}$. The least-square estimator of the ground-truth mixing measure $G_*$ is now defined as

$$\widetilde{G}_n \in \arg \min_{\widetilde{G} \in \widetilde{\mathcal{G}}_{L'}(\Theta)} \sum_{i=1}^{n} \left( \boldsymbol{Y}_i - f_{\widetilde{G}}(\mathbb{X}_i) \right)^2. \quad (16)$$

where $\widetilde{\mathcal{G}}_{L'}(\Theta) := \{ G = \sum_{i=1}^{\ell} \exp(c_i) \delta_{(\boldsymbol{W}_{2,i} \boldsymbol{B}_i, \boldsymbol{W}_{1,i} \boldsymbol{A}_i)} : 1 \leq \ell \leq L', (c_i, \boldsymbol{W}_{2,i} \boldsymbol{B}_i, \boldsymbol{W}_{1,i} \boldsymbol{A}_i) \in \Theta \}$. For the sake of capturing the convergence rate of the estimator $\widetilde{G}_n$, the Voronoi loss function is tailored to the non-linear reparametrization setting as

$$\mathcal{D}_3(\widetilde{G}, \widetilde{G}_*) := \sum_{j'=1}^{L} \Big| \sum_{i \in \mathcal{V}_{j'}} \exp(c_i) - \exp(c_{j'}^*) \Big|$$
$$+ \sum_{\substack{j' \in [L]: \\ |\mathcal{V}_{j'}|=1, \\ i \in \mathcal{V}_{j'}}} \exp(c_i)(\|\Delta(\boldsymbol{W}_2 \boldsymbol{B})_{ij'}\| + \|\Delta(\boldsymbol{W}_1 \boldsymbol{A})_{ij'}\|)$$
$$+ \sum_{\substack{j' \in [L]: \\ |\mathcal{V}_{j'}|>1, \\ i \in \mathcal{V}_{j'}}} \exp(c_i)(\|\Delta(\boldsymbol{W}_2 \boldsymbol{B})_{ij'}\|^2 + \|\Delta(\boldsymbol{W}_1 \boldsymbol{A})_{ij'}\|^2),$$

where we denote $\Delta(\boldsymbol{W}_2 \boldsymbol{B})_{ij'} := \boldsymbol{W}_{2,i} \boldsymbol{B}_i - \boldsymbol{W}_{2,j'}^* \boldsymbol{B}_{j'}^*$ and $\Delta(\boldsymbol{W}_1 \boldsymbol{A})_{ij'} := \boldsymbol{W}_{1,i} \boldsymbol{A}_i - \boldsymbol{W}_{1,j'}^* \boldsymbol{A}_{j'}^*$ for any $i, j'$. Before presenting the main result of this section in Theorem 5.3, it is necessary to impose some mild assumptions on the activations $\sigma_1$ and $\sigma_2$. Due to the space limit, we defer those assumptions to the proof of Theorem 5.3 in Appendix A.3.

**Theorem 5.3.** *Assume that the activation functions $\sigma_1$ and $\sigma_2$ meet the assumptions specified in Appendix A.3. Then, the estimator $\widetilde{G}_n$ converges to the true mixing measure $\widetilde{G}_*$ at the following rate:*

$$\mathcal{D}_3(\widetilde{G}_n, \widetilde{G}_*) = \mathcal{O}_P(\sqrt{\log(n)/n}).$$

Theorem 5.3 suggests that the convergence rates of estimating low-rank matrices $\boldsymbol{W}_{2,j}^* \boldsymbol{B}_j^*$ and $\boldsymbol{W}_{1,j}^* \boldsymbol{A}_j^*$ are either $\mathcal{O}_P([\log(n)/n]^{\frac{1}{2}})$ or $\mathcal{O}_P([\log(n)/n]^{\frac{1}{4}})$ depending on the cardinalities of their associated Voronoi cells, or equivalently, the number of their fitted parameters. In other words, we need a polynomial number of data, $\mathcal{O}(\epsilon^{-2})$ or $\mathcal{O}(\epsilon^{-4})$, to achieve the approximations of the low-rank matrices with the error $\epsilon$ when employing the LoRA with non-linear reparametrization. Compared with the LoRA without reparametrization, which requires up to an exponential amount of data for the same task, the LoRA with non-linear reparametrization is more sample efficient.

# 6. Reparameterized Low-rank Adaptation

In the previous section, we demonstrated that vanilla LoRA without reparameterization establishes a suboptimal rate for low-rank matrix estimation while introducing shared structural reparameterization to achieve the optimal rate. Building on this theoretical insight, we introduce our method: *Reparameterized Low-Rank Adaptation (RepLoRA)*. This method is tailored explicitly for fine-tuning transformer architectures by refining the linear layers that generate queries and values (or keys, queries, and values). This paper focuses on fine-tuning the queries and values for simplicity and clarity. Recall in vanilla LoRA, the matrices that generate the queries and values are given as:

$$\boldsymbol{W}_Q' = \boldsymbol{W}_Q + \boldsymbol{B}_Q \boldsymbol{A}_Q \quad \boldsymbol{W}_V' = \boldsymbol{W}_V + \boldsymbol{B}_V \boldsymbol{A}_V, \quad (17)$$

where $\boldsymbol{B}_Q, \boldsymbol{B}_V \in \mathbb{R}^{m \times r}$ and $\boldsymbol{A}_Q, \boldsymbol{A}_V \in \mathbb{R}^{r \times n}$ are learnable low-rank matrices. Inspired by our theoretical results, RepLoRA innovatively reparameterizes $\boldsymbol{A}$ and $\boldsymbol{B}$, modeling them as outputs of two MLPs. With non-linear reparameterization, the low-rank matrices are given by:

$$[\boldsymbol{A}_Q, \boldsymbol{A}_V] = g_{\theta_A}(\boldsymbol{A}) \quad [\boldsymbol{B}_Q, \boldsymbol{B}_V] = g_{\theta_B}(\boldsymbol{B}), \quad (18)$$

where $\boldsymbol{A}, \boldsymbol{B}$ are learnable matrices, and $g_{\theta_A}, g_{\theta_B}$ are two-layer MLPs with a shared part and distinct output heads. In this approach, $\boldsymbol{A}_Q$ and $\boldsymbol{A}_V$ are derived from a shared underlying input $\boldsymbol{A}$, with distinct outputs $\boldsymbol{A}_Q$ and $\boldsymbol{A}_V$ produced by the separated heads of $g_{\theta_A}$. Similarly, $\boldsymbol{B}_Q$ and $\boldsymbol{B}_V$ follow the same structure, leveraging a shared $\boldsymbol{B}$ input. While we focus on fine-tuning the queries and values to streamline the analysis, this formulation can naturally be extended to fine-tune the keys. We implement $\boldsymbol{A}$ and $\boldsymbol{B}$ as diagonal matrices to ensure parameter efficiency. After training, the reparameterization $g_{\theta_A}$ and $g_{\theta_B}$ can be discarded, and only the fine-tuned matrices $\boldsymbol{A}_Q, \boldsymbol{A}_V, \boldsymbol{B}_Q$, and $\boldsymbol{B}_V$ need to be retained for inference. Hence, this approach does not incur any additional computational overhead for inference. An illustration of this method is provided in Figure 1.

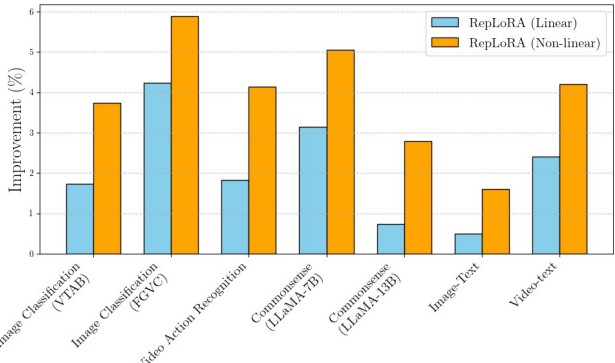

*Figure 3.* Performance improvements over LoRA. RepLoRA outperforms LoRA across all domains, with non-linear reparameterization substantially surpassing its linear counterpart.

# 7. Experiments

**Experimental Settings.** We conduct extensive experiments across multiple domains to demonstrate the versatility and effectiveness of RepLoRA in a wide range of tasks. Our evaluation spans four distinct settings: language (commonsense reasoning), image (classification), video (video action recognition), and multi-modal (image/video-text understanding). To provide a comprehensive evaluation, we compare RepLoRA against several PEFT methods, such as *Full Fine-tuning*, *Prefix Tuning* (Li & Liang, 2021), *LoRA* (Hu et al., 2022), *DoRA* (Liu et al., 2024b), and *Series Adapter* (Houlsby et al., 2019). As summarized in Figure 3, RepLoRA consistently outperforms LoRA across all settings, highlighting its robust adaptability and superior performance in various tasks.

**Evaluation Metrics.** In Parameter-Efficient Fine-Tuning (PEFT), evaluations typically focus on performance and the number of trainable parameters. The goal is to maximize performance while minimizing the parameters required. To assess this trade-off, in addition to reporting performance, we adopt the **Performance-Parameter Trade-off (PPT)** metric, proposed by Xin et al. (2024). Specifically, for a PETL algorithm $M$, the PPT metric incorporates its task performance $M_t$, the number of trainable parameters $P_M$, and a normalization constant $C$. Formally, we have:

$$\text{PPT}_M = M_t \times \exp(-\log_{10}(\frac{P_M}{C} + 1)).$$

**Commonsense Reasoning.** In our first experiment, we compare the performance of RepLoRA against LoRA and other PEFT methods using LLaMA-7B/13B (Touvron et al., 2023) on the Commonsense Reasoning task. We also include the accuracy of ChatGPT, measured with the GPT-3.5-turbo API with a zero-shot Chain of Thought approach (Wei et al., 2022). The commonsense reasoning benchmark comprises eight sub-tasks with predefined train-

*Table 1.* Top-1 Accuracy and PPT on commonsense datasets. The accuracies are reported with `LLaMA-7B` and `LLaMA-13B`.

| Model | Method | #Params (%) | BoolQ | PIQA | SIQA | HellaSwag | WinoGrande | ARC-e | ARC-c | OBQA | AVG | PPT |
|---|---|---|---|---|---|---|---|---|---|---|---|---|
| ChatGPT | - | - | 73.1 | 85.4 | 68.5 | 78.5 | 66.1 | 89.8 | 79.9 | 74.8 | 77.0 | - |
| LLaMA-7B | Prefix | 0.11 | 64.3 | 76.8 | 73.9 | 42.1 | 72.1 | 72.9 | 54.0 | 60.6 | 64.6 | 0.83 |
| | LoRA | 0.83 | 67.2 | 79.4 | 76.6 | 78.3 | 78.4 | 77.1 | 61.5 | 74.2 | 74.1 | 1.70 |
| | Adapter | 0.99 | 63.0 | 79.2 | 76.3 | 67.9 | 75.7 | 74.5 | 57.1 | 72.4 | 70.8 | 1.74 |
| | DoRA | 0.98 | 69.7 | 83.4 | 78.6 | **87.2** | 81.0 | 81.9 | 66.2 | 79.2 | 78.4 | 1.81 |
| | **RepLoRA** | 1.01 | **71.8** | **84.1** | **79.3** | 85.2 | **83.3** | **82.4** | **66.2** | **81.2** | **79.1** | **1.96** |
| LLaMA-13B | Prefix | 0.03 | 65.3 | 75.4 | 72.1 | 55.2 | 68.6 | 79.5 | 62.9 | 68.0 | 68.4 | 0.79 |
| | LoRA | 0.67 | 71.7 | 82.4 | 79.6 | 90.4 | 83.6 | 83.1 | 68.5 | 82.1 | 80.2 | 2.15 |
| | Adapter | 0.80 | 71.8 | 83.0 | 79.2 | 88.1 | 82.4 | 82.5 | 67.3 | 81.8 | 79.5 | 1.80 |
| | DoRA | 0.68 | 72.4 | 84.9 | 81.5 | **92.4** | 84.2 | 84.2 | 69.6 | 82.8 | 81.5 | 2.19 |
| | **RepLoRA** | 0.99 | **73.1** | **85.2** | **84.7** | 91.1 | **85.9** | **84.7** | **73.4** | **85.6** | **82.9** | **2.60** |

ing and testing datasets. Following the settings outlined in (Hu et al., 2023), we combine the training datasets from all eight sub-tasks into a single training dataset and evaluate performance on the individual testing datasets for each task. To ensure a fair comparison, we fine-tuned the models with RepLoRA using the same configuration as LoRA, keeping the rank fixed. As presented in Table 1, RepLoRA achieves significantly better results than LoRA across all settings, delivering substantial improvements not only in accuracy but also in the PPT score, emphasizing its parameter efficiency.

**Image Classification.** We extend our evaluation of RepLoRA to the image domain and fine-tune the ViT-B/16 architecture (Dosovitskiy, 2021), pre-trained on the `ImageNet-21K` dataset (Deng et al., 2009), on two challenging benchmarks: the `VTAB-1K` dataset suite (Zhai et al., 2019) and the `FGVC` dataset collection (Jia et al., 2022).

The `VTAB-1K` benchmark is a diverse suite of 19 datasets spanning various domains designed to test image classification and prediction capabilities. These datasets cover a wide range of tasks involving distinct semantics and object categories, organized into Natural, Specialized, and Structured domains. Each dataset includes 1,000 training examples, with an official 80/20 train-validation split, making it a rigorous test for generalization across different domains. On the other hand, the Fine-Grained Visual Classification (`FGVC`) suite, which consists of five datasets tailored for fine-grained recognition, focuses on tasks requiring subtle visual discrimination between closely related categories within specific domains. These datasets challenge models to identify nuanced differences, robustly evaluating RepLoRA's capabilities in fine-grained classification.

*Table 2.* Classification performance on `FGVC` datasets.

| Method | CUB-200 -2011 | NABirds | Oxford Flowers | Stanford Dogs | Stanford Cars | AVG | PPT |
|---|---|---|---|---|---|---|---|
| FFT | 87.3 | 82.7 | 98.8 | 89.4 | 84.5 | 88.5 | - |
| LoRA | 84.6 | 78.2 | 98.9 | 85.1 | 77.1 | 84.8 | 0.82 |
| Adapter | 87.1 | 84.3 | 98.5 | 89.8 | 68.6 | 85.6 | 0.84 |
| Prefix | 87.5 | 82.0 | 98.0 | 74.2 | **90.2** | 86.3 | 0.85 |
| DoRA | 87.3 | 80.0 | 99.1 | 87.6 | 81.9 | 87.2 | 0.88 |
| **RepLoRA** | **89.1** | **86.1** | **99.3** | **91.2** | 87.6 | **90.7** | **0.90** |

The results, summarized in Table 3 and Table 2, highlight the superior performance of RepLoRA across most settings.

On average, RepLoRA achieves a notable improvement of over 3% compared to LoRA, with notable gains exceeding 6% on datasets like `dSprites-location`. Similarly, RepLoRA outperforms all baselines on the `FGVC` datasets, with the sole exception of Prefix Tuning on the `Stanford Cars` dataset. The performance gap with LoRA is particularly significant, which was $> 6\%$ on average. On `Stanford Cars`, the improvement reaches a remarkable 10%. RepLoRA remains highly parameter-efficient despite these substantial gains, as reflected in its PPT scores.

**Video Action Recognition.** Given RepLoRA's strong performance in the image domain, we expand our experiments to the video domain. We evaluate our method against baseline approaches using the Video Swin Transformer on two datasets: `SSv2` (Goyal et al., 2017a), which offers a rich dataset with abundant data, and `HMDB51` (Kuehne et al., 2011), which presents a more challenging scenario with limited data and fewer categories. Despite the contrasting characteristics of these datasets, Table 4 indicates that RepLoRA remarkably outperforms all baselines while maintaining parameter efficiency, underscoring its adaptability and robustness across diverse data settings.

**Image/Video-Text understanding.** Having demonstrated that RepLoRA outperforms the baselines on the language and vision tasks, we attempt to see if RepLoRA remains competitive on multi-modality tasks. This experiment compares RepLoRA with LoRA and full fine-tuning (FT) on the `VL-BART` (Lewis et al., 2020). The experiments were conducted on four image-text tasks: $VQA^{v^2}$ (Goyal et al., 2017b), GQA (Hudson & Manning, 2019) for vision question-answering, $NLVR^2$ (Suhr et al., 2019) for visual reasoning, MSCOCO (Chen et al., 2015) for image captioning, and four video-text tasks from the `VALUE` benchmark (Li et al., 2021): TVQA (Lei et al., 2018) and How2QA (Li et al., 2020) for video question answering, TVC (Lei et al., 2020) and YC2C (Zhou et al., 2018) for video captioning. We follow Sung et al. (2022) and adopt the same setup of LoRA when applying RepLoRA. It is evident that RepLoRA consistently surpasses both FT and LoRA in accuracy and PPT in both Tables 5 and Table 6. In particular, RepLoRA

*Table 3.* Performance on `VTAB-1K` with `ViT-B/16` pre-trained on `ImageNet-21K`.

| | Natural | | | | | | | Specialized | | | | Structured | | | | | | | | | |
|---|---|---|---|---|---|---|---|---|---|---|---|---|---|---|---|---|---|---|---|---|---|
| Method | CIFAR100 | Caltech101 | DTD | Flower102 | Pets | SVHN | Sun397 | Camelyon | EuroSAT | Resisc45 | Retinopathy | Clevr-Count | Clevr-Dist | DMLab | KITTI | dSpr-Loc | dSpr-Ori | sNORB-Azim | sNORB-Ele | AVG | PPT |
| FFT | 68.9 | 87.7 | 64.3 | 97.2 | 86.9 | 87.4 | 38.8 | 79.7 | 95.7 | 84.2 | 73.9 | 56.3 | 58.6 | 41.7 | 65.5 | 57.5 | 46.7 | 25.7 | 29.1 | 65.5 | - |
| LoRA | 67.1 | 91.4 | 69.4 | 98.2 | 90.4 | 85.3 | 54 | 84.9 | 95.3 | 84.4 | 73.6 | 82.9 | 69.2 | 49.8 | 78.5 | 75.7 | 47.1 | 31 | 44.0 | 72.2 | 0.72 |
| Adapter | 69.2 | 90.1 | 68 | 98.8 | 89.9 | 82.8 | 54.3 | 84 | 94.9 | 81.9 | 75.5 | 80.9 | 65.3 | 48.6 | 78.3 | 74.8 | 48.5 | 29.9 | 41.6 | 71.4 | 0.71 |
| Prefix | **75.5** | 90.7 | 65.4 | 96.6 | 86 | 78.5 | 46.7 | 79.5 | 95.1 | 80.6 | 74.0 | 69.9 | 58.2 | 40.9 | 69.5 | 72.4 | 46.8 | 23.9 | 34.4 | 67.6 | 0.73 |
| **RepLoRA** | 73.2 | **94.1** | **73.3** | **99.3** | **94.4** | **89.1** | **58.9** | **89.2** | **97.5** | **87.9** | **77.8** | **85.1** | **72.6** | **55.7** | **81.2** | **81.7** | **49.2** | **35.7** | **47.3** | **75.9** | **0.74** |

*Table 4.* Performance on Video Action Recognition task.

| | | | | SSv2 | | HMDB51 | |
|---|---|---|---|---|---|---|---|
| Method | Model | Pretraining | #Params (M) | Acc@1 | PPT | Acc@1 | PPT |
| FFT | Video Swin-B | Kinetics400 | 87.64 | 50.99 | - | 68.07 | - |
| LoRA | Video Swin-B | Kinetics400 | 0.75 | 38.34 | 0.37 | 62.12 | 0.61 |
| Adapter | Video Swin-B | Kinetics400 | 1.56 | 39.09 | 0.36 | 67.52 | 0.63 |
| Prefix | Video Swin-B | Kinetics400 | 6.37 | 39.46 | 0.31 | 56.13 | 0.45 |
| **RepLoRA** | Video Swin-B | Kinetics400 | 1.45 | **46.12** | **0.41** | **68.23** | **0.64** |

exceeds LoRA's performance by nearly 2% in image-text understanding tasks and roughly 4% in video-text understanding tasks, reaching the performance of FT.

*Table 5.* Performance on image-text tasks with `VL-BART`.

| Method | #Params (%) | VQA$^{v^2}$ | GQA | NVLR$^2$ | COCO Cap | AVG | PPT |
|---|---|---|---|---|---|---|---|
| FT | 100 | **66.9** | **56.7** | 73.7 | 112.0 | 77.3 | - |
| LoRA | 5.93 | 65.2 | 53.6 | 71.9 | 115.3 | 76.5 | 0.99 |
| DoRA | 5.96 | 65.8 | 54.7 | 73.1 | 115.9 | 77.4 | 1.00 |
| RepLoRA | 6.02 | 66.5 | 55.4 | **74.2** | **116.2** | **78.1** | **1.02** |

**Enhancing Sampling Efficiency.** Our theoretical analysis has demonstrated that reparameterization achieves superior rates of sample efficiency compared to vanilla LoRA. To validate this claim, we evaluate the sample efficiency of RepLoRA on `FGVC` datasets. Following the approach of d'Ascoli et al. (2021), we subsample each class at fractions $f = \{1\%, 10\%, 30\%, 50\%, 100\%\}$ and scale the number of training epochs by $1/f$, ensuring the total number of data seen by the model remains constant. Figure 2 shows that Re-pLoRA consistently outperforms LoRA across all sampling fractions. The improvements are significant at smaller fractions, with RepLoRA achieving a remarkable **40.4%** gap at $f = 1\%$. More importantly, we emphasize that RepLoRA matches LoRA's performance with only **30%** training fraction, therefore underscoring RepLoRA's superior sample efficiency, as predicted by our theoretical analysis. We refer to Appendix C.1 for a breakdown of these results.

**Linear vs. Non-linear Reparameterization.** Another conclusion from the theoretical analysis is that even a simple linear reparameterization with a shared structure offers significant efficiency gains compared to vanilla LoRA. Furthermore, as shown in Theorems 5.2 and 5.3, incorporating non-linear reparameterization further improves the rate of low-rank matrix estimation. To validate this hypothesis, we

*Table 6.* Performance on video-text tasks with `VL-BART`.

| Method | #Params (%) | TVQA | How2QA | TVC | YC2C | AVG | PPT |
|---|---|---|---|---|---|---|---|
| FT | 100 | 76.3 | 73.9 | 45.7 | **154.0** | 87.5 | - |
| LoRA | 5.17 | 75.5 | 72.9 | 44.6 | 140.9 | 83.5 | 1.06 |
| DoRA | 5.19 | 76.3 | 74.1 | 45.8 | 145.4 | 85.4 | 1.08 |
| RepLoRA | 5.30 | **77.8** | **75.1** | **46.6** | 151.6 | **87.8** | **1.12** |

conducted empirical experiments, with the results presented in Figure 3. These results demonstrate that non-linear reparameterization outperforms the linear setting by substantial margins, underscoring its effectiveness. For a more detailed comparison of linear versus non-linear reparameterization performance, please refer to Appendix C.2.

## 8. Conclusion

We introduced a theoretical framework that bridges LoRA with MoE, offering new insights into the benefits of reparameterizing LoRA for achieving optimal sampling efficiency. Building on this theoretical foundation, we proposed RepLoRA, an effective and efficient approach to PEFT. To evaluate RepLoRA, we conducted extensive experiments across four diverse domains: image, video, text, and multi-modal tasks. RepLoRA substantially outperformed LoRA and other PEFT methods in all settings, demonstrating its adaptability and effectiveness. These results highlight the potential of reparameterized structures in enhancing efficiency and effectiveness for fine-tuning large-scale models.

## Impact Statement

This paper presents work that aims to advance the field of Machine Learning. Our work has potential societal consequences, none of which we feel must be specifically highlighted here.

## Acknowledgement

Trung Le was partly supported by ARC DP23 grant DP230101176 and by the Air Force Office of Scientific Research under award number FA9550-23-S-0001.

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

# Supplement to "RepLoRA: Reparameterizing Low-Rank Adaptation via the Perspective of Mixture of Experts"

In this supplementary material, we provide proofs of the main results in Appendix A. Details of experiments are provided in Appendix B, while additional experiments are presented in Appendix C.

## A. Proofs of Theoretical Results

This appendix provides proofs for key results in the main text.

### A.1. Proof of Theorem 5.1

The proof is divided into two steps as follows:

**Step 1.** To begin with, we demonstrate that the following limit holds for any $r \geq 1$:

$$\lim_{\varepsilon \to 0} \inf_{G \in \mathcal{G}_{L'}(\Theta) : \mathcal{D}_{1,r}(G, G_*) \leq \varepsilon} \frac{\|f_G - f_{G_*}\|_{L^2(\mu)}}{\mathcal{D}_{1,r}(G, G_*)} = 0. \tag{19}$$

Note that it is sufficient to construct a mixing measure sequence $(G_n)_{n \geq 1}$ that satisfies both $\mathcal{D}_{1,r}(G_n, G_*) \to 0$ and $\|f_{G_n} - f_{G_*}\|_{L^2(\mu)} / \mathcal{D}_{1,r}(G_n, G_*) \to 0$, as $n \to \infty$.

For that purpose, we take into account the sequence $G_n = \sum_{i=1}^{L+1} \exp(c_i^n) \delta_{(\boldsymbol{B}_{Q,i}^n, \boldsymbol{A}_{Q,i}^n, \boldsymbol{B}_{V,i}^n, \boldsymbol{A}_{V,i}^n)}$, where

- $\exp(c_1^n) = \exp(c_2^n) = \frac{1}{2} \exp(c_1^*) + \frac{1}{2n^{r+1}}$ and $\exp(c_i^n) = \exp(c_{i-1}^*)$ for any $3 \leq i \leq L+1$;

- $\boldsymbol{B}_{Q,1}^n = \boldsymbol{B}_{Q,2}^n = \boldsymbol{B}_{Q,1}^*$ and $\boldsymbol{B}_{Q,i}^n = \boldsymbol{B}_{Q,i-1}^*$ for any $3 \leq i \leq L+1$;

- $\boldsymbol{A}_{Q,1}^n = \boldsymbol{A}_{Q,2}^n = \boldsymbol{A}_{Q,1}^*$ and $\boldsymbol{A}_{Q,i}^n = \boldsymbol{A}_{Q,i-1}^*$ for any $3 \leq i \leq L+1$;

- $\boldsymbol{B}_{V,1}^n = \boldsymbol{B}_{V,1}^* + \frac{1}{n} e_{1,1}$, $\boldsymbol{B}_{V,2}^n = \boldsymbol{B}_{V,1}^* - \frac{1}{n} e_{1,1}$ and $\boldsymbol{B}_{V,i}^n = \boldsymbol{B}_{V,i-1}^*$ for any $3 \leq i \leq L+1$,

- $\boldsymbol{A}_{V,1}^n = \boldsymbol{A}_{V,2}^n = \boldsymbol{A}_{V,1}^*$ and $\boldsymbol{A}_{V,i}^n = \boldsymbol{A}_{V,i-1}^*$ for any $3 \leq i \leq L+1$;

where we denote $e_{1,1}$ as the matrix of size $\bar{d} \times \bar{d}$ such that its $(1,1)$-th element is one while all other elements are zero.

Then, we can compute the loss function $\mathcal{D}_{1,r}(G_n, G_*)$ as

$$\mathcal{D}_{1,r}(G_n, G_*) = \frac{1}{n^{r+1}} + \left[ \exp(c_1^*) + \frac{1}{n^{r+1}} \right] \cdot \frac{1}{n^r} = \mathcal{O}(n^{-r}). \tag{20}$$

It can be seen that $\mathcal{D}_{1,r}(G_n, G_*) \to 0$ as $n \to \infty$.

Subsequently, we illustrate that $\|f_{G_n} - f_{G_*}\|_{L^2(\mu)} / \mathcal{D}_{1,r}(G_n, G_*) \to 0$. In particular, let us consider the quantity

$$Q_n(\mathbb{X}) := \left[ \sum_{k=1}^{L} \exp(\mathbb{X}^\top (\boldsymbol{M}_Q^0 + \boldsymbol{B}_{Q,k}^* \boldsymbol{A}_{Q,k}^*) \mathbb{X} + c_k^*) \right] \cdot [f_{G_n}(\mathbb{X}) - f_{\bar{G}_*}(\mathbb{X})],$$

which can be decomposed as follows:

$$Q_n(\mathbb{X}) = \sum_{j=1}^{L} \sum_{i \in \mathcal{V}_j} \exp(c_i^n) \Big[ \exp(\mathbb{X}^\top (\boldsymbol{M}_Q^0 + \boldsymbol{B}_{Q,i}^n \boldsymbol{A}_{Q,i}^n) \mathbb{X})(\boldsymbol{M}_V^0 + \boldsymbol{B}_{V,i}^n \boldsymbol{A}_{V,i}^n) \mathbb{X}$$
$$- \exp(\mathbb{X}^\top (\boldsymbol{M}_Q^0 + \boldsymbol{B}_{Q,j}^* \boldsymbol{A}_{Q,j}^*) \mathbb{X})(\boldsymbol{M}_V^0 + \boldsymbol{B}_{V,j}^* \boldsymbol{A}_{V,j}^*) \mathbb{X} \Big]$$
$$- \sum_{j=1}^{L} \sum_{i \in \mathcal{V}_j} \exp(c_i^n) \Big[ \exp(\mathbb{X}^\top (\boldsymbol{M}_{Q,i}^0 + \boldsymbol{B}_{Q,i}^n \boldsymbol{A}_{Q,i}^n) \mathbb{X}) - \exp(\mathbb{X}^\top (\boldsymbol{M}_Q^0 + \boldsymbol{B}_{Q,j}^* \boldsymbol{A}_{Q,j}^*) \mathbb{X}) \Big] f_{G_n}(\mathbb{X})$$
$$+ \sum_{j=1}^{L} \Big( \sum_{i \in \mathcal{V}_j} \exp(c_i^n) - \exp(c_j^*) \Big) \exp(\mathbb{X}^\top (\boldsymbol{M}_Q^0 + \boldsymbol{B}_{Q,j}^* \boldsymbol{A}_{Q,j}^*) \mathbb{X}) \Big[ (\boldsymbol{M}_V^0 + \boldsymbol{B}_{V,j}^* \boldsymbol{A}_{V,j}^*) \mathbb{X} - f_{G_n}(\mathbb{X}) \Big]$$
$$:= A_n(\mathbb{X}) - B_n(\mathbb{X}) + C_n(\mathbb{X}).$$

It follows from the choices of $\boldsymbol{B}_{Q,i}^n, \boldsymbol{A}_{Q,i}^n, \boldsymbol{B}_{V,i}^n, \boldsymbol{A}_{V,i}^n$ and $c_i^n$ that

$$A_n(\mathbb{X}) = \sum_{i=1}^{2} \frac{1}{2} \Big[ \exp(c_1^*) + \frac{1}{n^{r+1}} \Big] \exp(\mathbb{X}^\top (\boldsymbol{M}_Q^0 + \boldsymbol{B}_{Q,1}^* \boldsymbol{A}_{Q,1}^*) \mathbb{X})(\boldsymbol{B}_{V,i}^n \boldsymbol{A}_{V,i}^n - \boldsymbol{B}_{V,1}^* \boldsymbol{A}_{V,1}^*) \mathbb{X}$$
$$= \frac{1}{2} \Big[ \exp(b_{*,1}) + \frac{1}{n^{r+1}} \Big] \exp(\mathbb{X}^\top (\boldsymbol{M}_Q^0 + \boldsymbol{B}_{Q,1}^* \boldsymbol{A}_{Q,1}^*) \mathbb{X})[(\boldsymbol{B}_{V,1}^n \boldsymbol{A}_{V,1}^n - \boldsymbol{B}_{V,1}^* \boldsymbol{A}_{V,1}^*) + (\boldsymbol{B}_{V,2}^n \boldsymbol{A}_{V,2}^n - \boldsymbol{B}_{V,1}^* \boldsymbol{A}_{V,1}^*)] \mathbb{X}$$
$$= 0,$$

where the last equality occurs as $\boldsymbol{B}_{V,1}^n \boldsymbol{A}_{V,1}^n - \boldsymbol{B}_{V,1}^* \boldsymbol{A}_{V,1}^* = \frac{1}{n} e_{1,1} \boldsymbol{A}_{V,1}^*$ and $\boldsymbol{B}_{V,2}^n \boldsymbol{A}_{V,2}^n - \boldsymbol{B}_{V,1}^* \boldsymbol{A}_{V,1}^* = -\frac{1}{n} e_{1,1} \boldsymbol{A}_{V,1}^*$.

Moreover, we can also verify that $B_n(\mathbb{X}) = 0$, and $C_n(\mathbb{X}) = \mathcal{O}(n^{-(r+1)})$. Thus, we deduce that $Q_n(\mathbb{X})/\mathcal{D}_{1,r}(G_n, G_*) \to 0$ as $n \to \infty$ for almost every $\mathbb{X}$.

As the term $\Big[ \sum_{k=1}^{L} \exp(\mathbb{X}^\top (\boldsymbol{M}_Q^0 + \boldsymbol{B}_{Q,k}^* \boldsymbol{A}_{Q,k}^*) \mathbb{X} + c_k^*) \Big]$ is bounded, we have $[f_{G_n}(\mathbb{X}) - f_{G_*}(\mathbb{X})]/\mathcal{D}_{1,r}(G_n, G_*) \to 0$ for almost every $\mathbb{X}$. This limit suggests that

$$\|f_{G_n} - f_{G_*}\|_{L^2(\mu)}/\mathcal{D}_{1,r}(G_n, G_*) \to 0$$

as $n \to \infty$. Thus, we obtain the claim in equation (19).

**Step 2.** We will establish the desired result in this step, that is,

$$\inf_{\overline{G}_n \in \mathcal{G}_{L'}(\Theta)} \sup_{G \in \mathcal{G}_{L'}(\Theta) \setminus \mathcal{G}_{L-1}(\Theta)} \mathbb{E}_{f_G}[\mathcal{D}_{1,r}(\overline{G}_n, G)] \gtrsim n^{-1/2}. \tag{21}$$

Since the noise variables $\epsilon_i$ follow from the Gaussian distribution, we get that $Y_i|\mathbb{X}_i \sim \mathcal{N}(f_{G_*}(\mathbb{X}_i), \sigma^2)$ for all $i \in [n]$. Additionally, for sufficiently small $\varepsilon > 0$ and a fixed constant $C_1 > 0$ which we will select later, we can find a mixing measure $G_*' \in \mathcal{G}_{L'}(\Theta)$ such that $\mathcal{D}_{1,r}(G_*', G_*) = 2\varepsilon$ and $\|f_{G_*'} - f_{G_*}\|_{L^2(\mu)} \le C_1 \varepsilon$ thanks to the result in equation (19). According to the Le Cam's lemma (Yu, 1997), as the Voronoi loss function $\mathcal{D}_{1,r}$ satisfies the weak triangle inequality, it follows that

$$\inf_{\overline{G}_n \in \mathcal{G}_{L'}(\Theta)} \sup_{G \in \mathcal{G}_{L'}(\Theta) \setminus \mathcal{G}_{L-1}(\Theta)} \mathbb{E}_{f_G}[\mathcal{D}_{1,r}(\overline{G}_n, G)]$$
$$\gtrsim \frac{\mathcal{D}_{1,r}(G_*', G_*)}{8} \exp(-n \mathbb{E}_{\mathbb{X} \sim \mu}[\mathrm{KL}(\mathcal{N}(f_{G_*'}(\mathbb{X}), \sigma^2), \mathcal{N}(f_{G_*}(\mathbb{X}), \sigma^2))])$$
$$\gtrsim \varepsilon \cdot \exp(-n \|f_{G_*'} - f_{G_*}\|_{L^2(\mu)}^2)$$
$$\gtrsim \varepsilon \cdot \exp(-C_1 n \varepsilon^2), \tag{22}$$

where the second inequality follows from the equality

$$\mathrm{KL}(\mathcal{N}(f_{G_*'}(\mathbb{X}), \sigma^2), \mathcal{N}(f_{G_*}(\mathbb{X}), \sigma^2)) = \frac{(f_{G_*'}(\mathbb{X}) - f_{G_*}(\mathbb{X}))^2}{2\sigma^2}.$$

Let $\varepsilon = n^{-1/2}$, then we get that $\varepsilon \cdot \exp(-C_1 n \varepsilon^2) = n^{-1/2} \exp(-C_1)$. Consequently, we achieve the desired minimax lower bound in equation (21). Related works on parameter-efficient fine-tuning techniques, low-rank adaptation, and mixture of experts are in Appendix

## A.2. Proof for Theorem 5.2

We first start with the following result regarding the convergence rate of the regression function estimation $f_{\bar{G}_n}$ to the true regression function $f_{\bar{G}_*}$:

**Proposition A.1.** *Given the least square estimator $\bar{G}_n$ in equation (14), the convergence rate of the regression function estimation $f_{\bar{G}_n}(\cdot)$ to the true regression function $f_{\bar{G}_*}(\cdot)$ under the $L^2(\mu)$ norm is parametric on the sample size, that is,*

$$\|f_{\bar{G}_n} - f_{\bar{G}_*}\|_{L^2(\mu)} = \mathcal{O}_P(\sqrt{\log(n)/n}). \tag{23}$$

Proof of Proposition A.1 is given in Appendix A.4. Given rate of $f_{\bar{G}_n}$ in Proposition A.1, our goal is to demonstrate the following inequality:

$$\inf_{G \in \bar{\mathcal{G}}_{L'}(\Theta)} \|f_G - f_{\bar{G}_*}\|_{L^2(\mu)}/\mathcal{D}_2(G, G_*) > 0.$$

We divide the proof of the above inequality into local and global parts.

### A.2.1. LOCAL PART

For the local part, we prove that

$$\lim_{\varepsilon \to 0} \inf_{G \in \mathcal{G}_{L'}(\Theta): \mathcal{D}_2(G, \bar{G}_*) \leq \varepsilon} \|f_G - f_{\bar{G}_*}\|_{L^2(\mu)}/\mathcal{D}_2(G, \bar{G}_*) > 0.$$

Assume that the above claim does not hold. It indicates that we can find a sequence of mixing measures $G_n := \sum_{j'=1}^{L'} \exp(c_{n,j'}) \delta_{\boldsymbol{B}_{n,j'} \boldsymbol{A}_{n,j'}}$ in $\bar{\mathcal{G}}_L(\Theta)$ such that

$$\begin{cases} \mathcal{D}_{2n} := \mathcal{D}_2(G_n, \bar{G}_*) \to 0, \\ \|f_{G_n} - f_{\bar{G}_*}\|_{L^2(\mu)}/\mathcal{D}_{2n} \to 0. \end{cases}$$

as $n \to \infty$. We denote $\mathcal{V}_j^n := \mathcal{V}_j(G_n)$ as a Voronoi cell of $G_n$ generated by the $j$-th components of $\bar{G}_*$. Without loss of generality, we may assume that those Voronoi cells do not depend on the sample size, i.e., $\mathcal{V}_j = \mathcal{V}_j^n$. Therefore, the Voronoi loss $\mathcal{D}_{2n}$ can be rewritten as follows:

$$\mathcal{D}_{2n} := \sum_{j'=1}^{L} \Big| \sum_{i \in \mathcal{V}_{j'}} \exp(c_{n,i}) - \exp(c_{j'}^*) \Big| + \sum_{j' \in [L]: |\mathcal{V}_{j'}| = 1} \sum_{i \in \mathcal{V}_{j'}} \exp(c_{n,i}) \|\Delta \boldsymbol{W}_{n,2ij'} \boldsymbol{B}_{n,ij'} \boldsymbol{W}_{n,1ij'} \boldsymbol{A}_{n,ij'}\|$$

$$+ \sum_{j' \in [L]: |\mathcal{V}_{j'}| > 1} \sum_{i \in \mathcal{V}_{j'}} \exp(c_{n,i}) \|\Delta \boldsymbol{W}_{n,2ij'} \boldsymbol{B}_{n,ij'} \boldsymbol{W}_{n,1ij'} \boldsymbol{A}_{n,ij'}\|^2,$$

where $\boldsymbol{W}_{n,2ij'} \Delta \boldsymbol{B}_{n,ij'} \boldsymbol{W}_{n,1ij'} \boldsymbol{A}_{n,ij'} := \boldsymbol{W}_{n,2j'} \boldsymbol{B}_{n,j'} \boldsymbol{W}_{n,1j'} \boldsymbol{A}_{n,j'} - \boldsymbol{W}_{2,i}^* \boldsymbol{B}_i^* \boldsymbol{W}_{1,i}^* \boldsymbol{A}_i^*$ for all $i \in \mathcal{V}_{j'}$.

To simplify the ensuing presentation, throughout the proof we denote $\boldsymbol{Z} := \boldsymbol{W}_2 \boldsymbol{B} \boldsymbol{W}_1 \boldsymbol{A}$ for all the matrices $\boldsymbol{W}_1, \boldsymbol{W}_2, \boldsymbol{A}$, and $\boldsymbol{B}$. Given the new notation, the Voronoi loss $\mathcal{D}_{2n}$ becomes

$$\mathcal{D}_{2n} = \sum_{j'=1}^{L} \Big| \sum_{i \in \mathcal{V}_{j'}} \exp(c_{n,i}) - \exp(c_{j'}^*) \Big| + \sum_{j' \in [L]: |\mathcal{V}_{j'}| = 1} \sum_{i \in \mathcal{V}_{j'}} \exp(c_{n,i}) \|\Delta \boldsymbol{Z}_{n,ij'}\|$$

$$+ \sum_{j' \in [L]: |\mathcal{V}_{j'}| > 1} \sum_{i \in \mathcal{V}_{j'}} \exp(c_{n,i}) \|\Delta \boldsymbol{Z}_{n,ij'}\|^2.$$

Since $\mathcal{D}_{2n} \to 0$, we have $\sum_{i \in \mathcal{V}_j} \exp(c_{n,i}) \to \exp(c_{*,j})$, $\boldsymbol{Z}_{n,i} \to \boldsymbol{Z}_j^*$ for any $i \in \mathcal{V}_j$ and $j \in [L]$. Throughout this proof, we assume without loss of generality that $M_{K,j}^0 = I_{\bar{d}}$ with a note that our techniques can be extended to the general setting of that matrix. Now, the proof of the local part is divided into three steps as follows:

**Step 1 - Taylor expansion.** First, we define

$$Q_n(\mathbb{X}) := \Big[ \sum_{k=1}^{L} \exp(\mathbb{X}^\top (\boldsymbol{M}_Q^0 + \boldsymbol{Z}_k^*)\mathbb{X} + c_k^*) \Big] \cdot [f_{G_n}(\mathbb{X}) - f_{\bar{G}_*}(\mathbb{X})].$$

Then, we can decompose the function $Q_n(\mathbb{X})$ as follows:

$$
\begin{aligned}
Q_n(\mathbb{X}) = &\sum_{j=1}^{L} \sum_{i \in \mathcal{V}_j} \exp(c_{n,i}) \Big[ \exp(\mathbb{X}^\top(\boldsymbol{M}_Q^0 + \boldsymbol{Z}_{n,i})\mathbb{X})(\boldsymbol{M}_V^0 + \boldsymbol{Z}_{n,i})\mathbb{X} - \exp(\mathbb{X}^\top(\boldsymbol{M}_Q^0 + \boldsymbol{Z}_j^*)\mathbb{X})(\boldsymbol{M}_V^0 + \boldsymbol{Z}_j^*)\mathbb{X} \Big] \\
&- \sum_{j=1}^{L} \sum_{i \in \mathcal{V}_j} \exp(c_{n,i}) \Big[ \exp(\mathbb{X}^\top(\boldsymbol{M}_{Q,i}^0 + \boldsymbol{Z}_{n,i})\mathbb{X}) - \exp(\mathbb{X}^\top(\boldsymbol{M}_Q^0 + \boldsymbol{Z}_j^*)\mathbb{X}) \Big] f_{G_n}(\mathbb{X}) \\
&+ \sum_{j=1}^{L} \Big( \sum_{i \in \mathcal{V}_j} \exp(c_{n,i}) - \exp(c_j^*) \Big) \exp(\mathbb{X}^\top(\boldsymbol{M}_Q^0 + \boldsymbol{Z}_j^*)\mathbb{X}) \Big[ (\boldsymbol{M}_V^0 + \boldsymbol{Z}_j^*)\mathbb{X} - f_{G_n}(\mathbb{X}) \Big] \\
:= &\bar{A}_n(\mathbb{X}) - \bar{B}_n(\mathbb{X}) + \bar{C}_n(\mathbb{X}).
\end{aligned}
\tag{24}
$$

**Decomposition of the function $\bar{A}_n(\mathbb{X})$.** We denote $\bar{U}(\mathbb{X}; \boldsymbol{Z}) := \exp(\mathbb{X}^\top(\boldsymbol{M}_Q^0 + \boldsymbol{Z})\mathbb{X})$ and $\bar{V}(\mathbb{X}; \boldsymbol{Z}) = (\boldsymbol{M}_V^0 + \boldsymbol{Z})\mathbb{X}$, and $F(\mathbb{X}; \boldsymbol{Z}) = \bar{U}(\mathbb{X}; \boldsymbol{Z})\bar{V}(\mathbb{X}; \boldsymbol{Z})$. Based on the number of elements in each Voronoi cells, we decompose the function $\bar{A}_n(\mathbb{X})$ as follows:

$$
\begin{aligned}
\bar{A}_n(\mathbb{X}) = &\sum_{j:|\mathcal{V}_j|=1} \sum_{i \in \mathcal{V}_j} \exp(c_{n,i}) \Big[ F(\mathbb{X}; \boldsymbol{Z}_{n,i}) - F(\mathbb{X}; \boldsymbol{Z}_j^*) \Big] \\
&+ \sum_{j:|\mathcal{V}_j|>1} \sum_{i \in \mathcal{V}_j} \exp(c_{n,i}) \Big[ F(\mathbb{X}; \boldsymbol{Z}_{n,i} - F(\mathbb{X}; \boldsymbol{Z}_j^*) \Big] \\
:= &\bar{A}_{n,1}(\mathbb{X}) + \bar{A}_{n,2}(\mathbb{X})
\end{aligned}
$$

An application of the first-order Taylor expansion leads to

$$\bar{U}(\mathbb{X}; \boldsymbol{Z}_{n,i}) = \bar{U}(\mathbb{X}; \boldsymbol{Z}_j^*) + \sum_{|\alpha|=1} (\Delta \boldsymbol{Z}_{n,ij})^\alpha \frac{\partial^{|\alpha|} \bar{U}}{\partial (\boldsymbol{Z})^\alpha}(\mathbb{X}; \boldsymbol{Z}_j^*) + \bar{R}_{ij,1}(\mathbb{X}),$$

$$\bar{V}(\mathbb{X}; \boldsymbol{Z}_{n,i}) = \bar{V}(\mathbb{X}; \boldsymbol{Z}_j^*) + \sum_{|\alpha|=1} (\Delta \boldsymbol{Z}_{n,ij})^\alpha \frac{\partial^{|\alpha|} \bar{V}}{\partial (\boldsymbol{Z})^\alpha}(\mathbb{X}; \boldsymbol{Z}_j^*) + \bar{R}_{ij,2}(\mathbb{X}),$$

for any $i$ and $j$ such that $i \in \mathcal{V}_j$ and $|\mathcal{V}_j| = 1$. Here, the functions $\bar{R}_{ij,1}(\mathbb{X})$ and $\bar{R}_{ij,2}(\mathbb{X})$ denote the Taylor remainders. Collecting the above results leads to

$$
\begin{aligned}
\bar{A}_{n,1}(\mathbb{X}) = &\sum_{j:|\mathcal{V}_j|=1} \sum_{i \in \mathcal{V}_j} \frac{\exp(c_{n,i})}{\alpha!} \sum_{|\alpha|=1} \Big\{ (\Delta \boldsymbol{Z}_{n,ij})^\alpha \frac{\partial^{|\alpha|} \bar{U}}{\partial (\boldsymbol{Z})^\alpha}(\mathbb{X}; \boldsymbol{Z}_j^*) \bar{V}(\mathbb{X}; \boldsymbol{Z}_j^*) \\
&+ (\Delta \boldsymbol{Z}_{n,ij})^\alpha \frac{\partial^{|\alpha|} \bar{V}}{\partial (\boldsymbol{Z})^\alpha}(\mathbb{X}; \boldsymbol{Z}_j^*) \bar{U}(\mathbb{X}; \boldsymbol{Z}_j^*) \Big\} + \bar{R}_{n,1}(\mathbb{X}) \\
= &\sum_{j:|\mathcal{V}_j|=1} \sum_{|\alpha|=1} \Big\{ \bar{M}_{n,j,\alpha} \frac{\partial^{|\alpha|} \bar{U}}{\partial (\boldsymbol{Z})^\alpha}(\mathbb{X}; \boldsymbol{Z}_j^*) \bar{V}(\mathbb{X}; \boldsymbol{Z}_j^*) + \bar{M}_{n,j,\alpha} \frac{\partial^{|\alpha|} \bar{V}}{\partial (\boldsymbol{Z})^\alpha}(\mathbb{X}; \boldsymbol{Z}_j^*) \bar{U}(\mathbb{X}; \boldsymbol{Z}_j^*) \Big\} + \bar{R}_{n,1}(\mathbb{X})
\end{aligned}
$$

where the function $\bar{R}_{n,1}(\mathbb{X})$ satisfies that $\bar{R}_{n,1}(\mathbb{X})/\mathcal{D}_{2n} \to 0$. It is due to the uniform Lipschitz property of the function $F$. In the above display, the formulations of the coefficients $\bar{M}_{n,j,\alpha}$ are given by:

$$\bar{M}_{n,j,\alpha} = \sum_{i \in \mathcal{V}_j} \frac{\exp(c_{n,i})}{\alpha!} (\Delta \boldsymbol{Z}_{n,ij})^\alpha,$$

for any $|\alpha| = 1$.

Moving to the function $\bar{A}_{n,2}(\mathbb{X})$, an application of the Taylor expansion up to the second order leads to

$$\bar{A}_{n,2}(\mathbb{X}) = \sum_{j:|\mathcal{V}_j|>1} \sum_{1\le|\alpha|\le2} \left\{ \bar{M}_{n,j,\alpha} \frac{\partial^{|\alpha|}\bar{U}}{\partial(\boldsymbol{Z})^\alpha}(\mathbb{X}; \boldsymbol{Z}_j^*)\bar{V}(\mathbb{X}; \boldsymbol{Z}_j^*) + \bar{M}_{n,j,\alpha} \frac{\partial^{|\alpha|}\bar{V}}{\partial(\boldsymbol{Z})^\alpha}(\mathbb{X}; \boldsymbol{Z}_j^*)\bar{U}(\mathbb{X}; \boldsymbol{Z}_j^*) \right\}$$

$$+ \sum_{|\alpha|=1,|\beta|=1} \bar{M}_{n,j,\alpha,\beta} \frac{\partial^{|\alpha|}\bar{U}}{\partial(\boldsymbol{Z})^\alpha}(\mathbb{X}; \boldsymbol{Z}_j^*) \frac{\partial^{|\beta|}\bar{V}}{\partial(\boldsymbol{Z})^\beta}(\mathbb{X}; \boldsymbol{Z}_j^*) + \bar{R}_{n,2}(\mathbb{X})$$

where the remainder $\bar{R}_{n,2}(\mathbb{X})$ satisfies that $\bar{R}_{n,2}(\mathbb{X})/\mathcal{D}_{2n} \to 0$. In this equation, the coefficients $\bar{M}_{n,j,\alpha}$ and $\bar{M}_{n,j,\alpha,\beta}$ take the following forms:

$$\bar{M}_{n,j,\alpha} = \sum_{i\in\mathcal{V}_j} \frac{\exp(c_{n,i})}{\alpha!}(\Delta\boldsymbol{Z}_{n,ij})^\alpha,$$

for any $|\alpha| = 2$ and

$$M_{n,j,\alpha,\beta} = \sum_{i\in\mathcal{V}_j} \frac{\exp(c_{n,i})}{\alpha!\beta!}(\Delta\boldsymbol{Z}_{n,ij})^{\alpha+\beta},$$

for any $|\alpha| = |\beta| = 1$. Simple algebra leads to the following formulations of the partial derivatives of $\bar{U}(\mathbb{X}; \boldsymbol{Z})$ and $\bar{V}(\mathbb{X}; \boldsymbol{Z})$:

$$\frac{\partial\bar{U}}{\partial(\boldsymbol{Z})^{(u_1v_1)}}(\mathbb{X}; \boldsymbol{Z}) = \mathbb{X}^{(u_1)}\mathbb{X}^{(v_1)}\exp(\mathbb{X}^\top(\boldsymbol{M}_Q^0 + \boldsymbol{Z})\mathbb{X}),$$

$$\frac{\partial^2\bar{U}}{\partial(\boldsymbol{Z})^{(u_1v_1)}\partial(\boldsymbol{Z})^{(u_2v_2)}}(\mathbb{X}; \boldsymbol{Z}) = \mathbb{X}^{(u_1)}\mathbb{X}^{(v_1)}\mathbb{X}^{(u_2)}\mathbb{X}^{(v_2)}\exp(\mathbb{X}^\top(\boldsymbol{M}_Q^0 + \boldsymbol{Z})\mathbb{X}),$$

$$\frac{\partial\bar{V}}{\partial(\boldsymbol{Z})^{(u_1v_1)}}(\mathbb{X}; \boldsymbol{Z}) = \mathbb{X}^{(v_1)}e_{u_1},$$

$$\frac{\partial^2\bar{V}}{\partial(\boldsymbol{Z})^{(u_1v_1)}\partial(\boldsymbol{Z})^{(u_2v_2)}}(\mathbb{X}; \boldsymbol{Z}) = \boldsymbol{0}_{\bar{d}}.$$

Plugging these formulations into the functions $\bar{A}_{n,1}(\mathbb{X})$ and $\bar{A}_{n,2}(\mathbb{X})$, we obtain that

$$\bar{A}_{n,1}(\mathbb{X}) = \sum_{j:|\mathcal{V}_j|=1} \exp(\mathbb{X}^\top(\boldsymbol{M}_Q^0 + \boldsymbol{Z}_j^*)\mathbb{X}) \left[ \sum_{u_1,v_1=1}^{\bar{d}} \bar{M}_{n,j,e_{u_1v_1}} \mathbb{X}^{(u_1)}\mathbb{X}^{(v_1)}(\boldsymbol{M}_V^0 + \boldsymbol{Z}_j^*)\mathbb{X} + \sum_{u_1,v_1=1}^{\bar{d}} \bar{M}_{n,j,e_{u_1v_1}} \mathbb{X}^{(v_1)}e_{u_1} \right]$$

$$+ \bar{R}_{n,1}(\mathbb{X}),$$

$$\bar{A}_{n,2}(\mathbb{X}) = \sum_{j:|\mathcal{V}_j|>1} \exp(\mathbb{X}^\top(\boldsymbol{M}_Q^0 + \boldsymbol{Z}_j^*)\mathbb{X}) \left[ \sum_{u_1,v_1=1}^{\bar{d}} \bar{M}_{n,j,e_{u_1v_1}} \mathbb{X}^{(u_1)}\mathbb{X}^{(v_1)}(\boldsymbol{M}_V^0 + \boldsymbol{Z}_j^*)\mathbb{X} + \sum_{u_1,v_1=1}^{\bar{d}} \bar{M}_{n,j,e_{u_1v_1}} \mathbb{X}^{(v_1)}e_{u_1} \right.$$

$$+ \sum_{u_1,v_1=1}^{\bar{d}} \sum_{u_2,v_2=1}^{\bar{d}} \bar{M}_{n,j,e_{u_1v_1}+e_{u_2v_2}} \mathbb{X}^{(u_1)}\mathbb{X}^{(v_1)}\mathbb{X}^{(u_2)}\mathbb{X}^{(v_2)}\exp(\mathbb{X}^\top(\boldsymbol{M}_Q^0 + \boldsymbol{Z}_j^*)\mathbb{X})(\boldsymbol{M}_V^0 + \boldsymbol{Z}_j^*)\mathbb{X}$$

$$+ \left. \sum_{u_1,v_1=1}^{\bar{d}} \sum_{u_2,v_2=1}^{\bar{d}} \bar{M}_{n,j,e_{u_1v_1},e_{u_2v_2}} \mathbb{X}^{(u_1)}\mathbb{X}^{(v_1)}\mathbb{X}^{(v_2)}\exp(\mathbb{X}^\top(\boldsymbol{M}_Q^0 + \boldsymbol{Z}_j^*)\mathbb{X})e_{u_2} \right] + \bar{R}_{n,2}(\mathbb{X}).$$

In these equations, $e_u$ is denoted as the vector in $\mathbb{R}^{\bar{d}}$ such that its $u$-th element is 1 while its other elements are 0 for any $1 \le u \le \bar{d}$. Furthermore, $e_{uv}$ is denoted as matrix in $\mathbb{R}^{\bar{d}\times\bar{d}}$ with its $uv$-th entry is 1 while other entries are zero.

**Decomposition of the function $\bar{B}_n(\mathbb{X})$.** Moving to the function $\bar{B}_n(\mathbb{X})$, we can decompose this function as follows:

$$
\bar{B}_n(\mathbb{X}) = \sum_{j:|\mathcal{V}_j|=1} \sum_{i \in \mathcal{V}_j} \exp(c_{n,i}) \Big[ \bar{U}(\mathbb{X}; \boldsymbol{Z}_{n,i}) - \bar{U}(\mathbb{X}; \boldsymbol{Z}_j^*) \Big] f_{G_n}(\mathbb{X})
$$
$$
+ \sum_{j:|\mathcal{V}_j|>1} \sum_{i \in \mathcal{V}_j} \exp(c_{n,i}) \Big[ \bar{U}(\mathbb{X}; \boldsymbol{Z}_{n,i}) - \bar{U}(\mathbb{X}; \boldsymbol{Z}_j^*) \Big] f_{G_n}(\mathbb{X})
$$
$$
:= \bar{B}_{n,1}(\mathbb{X}) + \bar{B}_{n,2}(\mathbb{X}).
$$

An application of the Taylor expansions up to the first order for $\bar{B}_{n,1}(\mathbb{X})$ and the second order for $\bar{B}_{n,2}(\mathbb{X})$ leads to

$$
\bar{B}_{n,1}(\mathbb{X}) = \sum_{j:|\mathcal{V}_j|=1} \sum_{|\alpha|=1} \bar{M}_{n,j,\alpha} \frac{\partial^{|\alpha|} \bar{U}}{\partial(\boldsymbol{Z})^\alpha}(\mathbb{X}; \boldsymbol{Z}_j^*) f_{G_n}(\mathbb{X}) + \bar{R}_{n,3}(\mathbb{X}),
$$
$$
\bar{B}_{n,2}(\mathbb{X}) = \sum_{j:|\mathcal{V}_j|=1} \sum_{1 \le |\alpha| \le 2} \bar{M}_{n,j,\alpha} \frac{\partial^{|\alpha|} \bar{U}}{\partial(\boldsymbol{Z})^\alpha}(\mathbb{X}; \boldsymbol{Z}_j^*) f_{G_n}(\mathbb{X}) + \bar{R}_{n,4}(\mathbb{X})
$$

where the Taylor remainders $\bar{R}_{n,3}(\mathbb{X}), \bar{R}_{n,4}(\mathbb{X})$ satisfy that $\bar{R}_{n,3}(\mathbb{X})/\mathcal{D}_{2n} \to 0$ and $\bar{R}_{n,4}(\mathbb{X})/\mathcal{D}_{2n} \to 0$. Direct calculation leads to

$$
\bar{B}_{n,1}(\mathbb{X}) = \sum_{j:|\mathcal{V}_j|=1} \exp(\mathbb{X}^\top (\boldsymbol{M}_Q^0 + \boldsymbol{Z}_j^*)\mathbb{X}) \Big[ \sum_{u_1,v_1=1}^{\bar{d}} \bar{M}_{n,j,e_{u_1 v_1}} \mathbb{X}^{(u_1)} \mathbb{X}^{(v_1)} \Big] f_{G_n}(\mathbb{X}) + \bar{R}_{n,3}(\mathbb{X}),
$$
$$
\bar{B}_{n,2}(\mathbb{X}) = \sum_{j:|\mathcal{V}_j|>1} \exp(\mathbb{X}^\top (\boldsymbol{M}_Q^0 + \boldsymbol{Z}_j^*)\mathbb{X}) \Big[ \sum_{u_1,v_1=1}^{\bar{d}} \bar{M}_{n,j,e_{u_1 v_1}} \mathbb{X}^{(u_1)} \mathbb{X}^{(v_1)}
$$
$$
+ \sum_{u_1,v_1=1}^{\bar{d}} \sum_{u_2,v_2=1}^{\bar{d}} \bar{M}_{n,j,e_{u_1 v_1}} \mathbb{X}^{(u_1)} \mathbb{X}^{(v_1)} \mathbb{X}^{(u_2)} \mathbb{X}^{(v_2)} \Big] f_{G_n}(\mathbb{X}) + \bar{R}_{n,4}(\mathbb{X}),
$$

Putting all the above results together, we can represent the function $Q_n(\mathbb{X})$ as follows:

$$
Q_n(\mathbb{X}) = \sum_{j:|\mathcal{V}_j|=1} \exp(\mathbb{X}^\top (\boldsymbol{M}_Q^0 + \boldsymbol{Z}_j^*)\mathbb{X}) \Big[ \sum_{u_1,v_1=1}^{\bar{d}} \bar{M}_{n,j,e_{u_1 v_1}} \mathbb{X}^{(u_1)} \mathbb{X}^{(v_1)} (\boldsymbol{M}_Q^0 + \boldsymbol{Z}_j^*)\mathbb{X}) + \sum_{u_1,v_1=1}^{\bar{d}} \bar{M}_{n,j,e_{u_1 v_1}} \mathbb{X}^{(v_1)} e_{u_1} \Big]
$$
$$
+ \sum_{j:|\mathcal{V}_j|>1} \exp(\mathbb{X}^\top (\boldsymbol{M}_Q^0 + \boldsymbol{Z}_j^*)\mathbb{X}) \Big[ \sum_{u_1,v_1=1}^{\bar{d}} \bar{M}_{n,j,e_{u_1 v_1}} \mathbb{X}^{(u_1)} \mathbb{X}^{(v_1)} (\boldsymbol{M}_Q^0 + \boldsymbol{Z}_j^*)\mathbb{X} + \sum_{u_1,v_1=1}^{\bar{d}} \bar{M}_{n,j,e_{u_1 v_1}} \mathbb{X}^{(v_1)} e_{u_1}
$$
$$
+ \sum_{u_1,v_1=1}^{\bar{d}} \sum_{u_2,v_2=1}^{\bar{d}} \bar{M}_{n,j,e_{u_1 v_1}+e_{u_2 v_2}} \mathbb{X}^{(u_1)} \mathbb{X}^{(v_1)} \mathbb{X}^{(u_2)} \mathbb{X}^{(v_2)} \exp(\mathbb{X}^\top (\boldsymbol{M}_Q^0 + \boldsymbol{Z}_j^*)\mathbb{X})(\boldsymbol{M}_V^0 + \boldsymbol{Z}_j^*)\mathbb{X}
$$
$$
+ \sum_{u_1,v_1=1}^{\bar{d}} \sum_{u_2,v_2=1}^{\bar{d}} \bar{M}_{n,j,e_{u_1 v_1},e_{u_2 v_2}} \mathbb{X}^{(u_1)} \mathbb{X}^{(v_1)} \mathbb{X}^{(v_2)} \exp(\mathbb{X}^\top (\boldsymbol{M}_Q^0 + \boldsymbol{Z}_j^*)\mathbb{X}) e_{u_2} \Big]
$$

$$-\sum_{j:|\mathcal{V}_j|=1}\exp(\mathbb{X}^\top(\boldsymbol{M}_Q^0+\boldsymbol{Z}_j^*)\mathbb{X})\Big[\sum_{u_1,v_1=1}^{\bar{d}}\bar{M}_{n,j,e_{u_1v_1}}\mathbb{X}^{(u_1)}\mathbb{X}^{(v_1)}\Big]f_{G_n}(\mathbb{X})$$

$$-\sum_{j:|\mathcal{V}_j|>1}\exp(\mathbb{X}^\top(\boldsymbol{M}_Q^0+\boldsymbol{Z}_j^*)\mathbb{X})\Big[\sum_{u_1,v_1=1}^{\bar{d}}\bar{M}_{n,j,e_{u_1v_1}}\mathbb{X}^{(u_1)}\mathbb{X}^{(v_1)}$$

$$+\sum_{u_1,v_1=1}^{\bar{d}}\sum_{u_2,v_2=1}^{\bar{d}}\bar{M}_{n,j,e_{u_1v_1}}\mathbb{X}^{(u_1)}\mathbb{X}^{(v_1)}\mathbb{X}^{(u_2)}\mathbb{X}^{(v_2)}\Big]f_{G_n}(\mathbb{X})$$

$$-\sum_{j=1}^{L}\bar{N}_{n,j}\exp(\mathbb{X}^\top(\boldsymbol{M}_Q^0+\boldsymbol{Z}_j^*)\mathbb{X})f_{G_n}(\mathbb{X})+\sum_{j=1}^{L}\bar{N}_{n,j}\exp(\mathbb{X}^\top(\boldsymbol{M}_Q^0+\boldsymbol{Z}_j^*)\mathbb{X})(\boldsymbol{M}_V^0+\boldsymbol{Z}_j^*)\mathbb{X}$$

$$+\bar{R}_{n,1}(\mathbb{X})+\bar{R}_{n,2}(\mathbb{X})-\bar{R}_{n,3}(\mathbb{X})-\bar{R}_{n,4}(\mathbb{X}) \tag{25}$$

where $\bar{N}_{n,j}:=\sum_{i\in\mathcal{V}_j}\exp(c_{n,i})-\exp(c_j^*)$ for any $j\in[L]$.

**Step 2 - Non-vanishing coefficients.** As indicated in equation (25), the ratio $Q_n(\mathbb{X})/\mathcal{D}_{2n}$ can be expressed as a linear combination of the following independent functions:

$$\bar{U}(\mathbb{X};\boldsymbol{Z}_j^*)\mathbb{X}^{(u_1)}\mathbb{X}^{(v_1)}(\boldsymbol{M}_V^0+\boldsymbol{Z}_j^*)\mathbb{X},\quad \bar{U}(\mathbb{X};\boldsymbol{Z}_j^*)\mathbb{X}^{(v_1)}e_{u_1},$$

$$\bar{U}(\mathbb{X};\boldsymbol{Z}_j^*)\mathbb{X}^{(u_1)}\mathbb{X}^{(v_1)}\mathbb{X}^{(u_2)}\mathbb{X}^{(v_2)}(\boldsymbol{M}_V^0+\boldsymbol{Z}_j^*)\mathbb{X},\quad \bar{U}(\mathbb{X};\boldsymbol{Z}_j^*)\mathbb{X}^{(u_1)}\mathbb{X}^{(v_1)}\mathbb{X}^{(v_2)}e_{u_2},$$

$$\bar{U}(\mathbb{X};\boldsymbol{Z}_j^*)\mathbb{X}^{(u_1)}\mathbb{X}^{(v_1)}f_{G_n}(\mathbb{X}),\quad \bar{U}(\mathbb{X};\boldsymbol{Z}_j^*)\mathbb{X}^{(u_1)}\mathbb{X}^{(v_1)}\mathbb{X}^{(u_2)}\mathbb{X}^{(v_2)}f_{G_n}(\mathbb{X}),$$

$$\bar{U}(\mathbb{X};\boldsymbol{Z}_j^*)f_{G_n}(\mathbb{X}),\quad \bar{U}(\mathbb{X};\boldsymbol{Z}_j^*)(\boldsymbol{M}_V^0+\boldsymbol{Z}_j^*)\mathbb{X},$$

for any indices $1\le j\le L$ and $1\le u_1,v_1,u_2,v_2\le\bar{d}$.

We demonstrate that at least one of the coefficients of these independent functions does not go to 0 as $n\to\infty$. Assume by contrary that all these coefficients of these linear independent functions go to 0. From equation (25), we obtain that $\bar{M}_{n,j,\alpha}/\mathcal{D}_{2n}$, $\bar{M}_{n,j,\alpha,\beta}/\mathcal{D}_{2n}$, and $\bar{N}_{n,j}/\mathcal{D}_{2n}$ go to 0 for all the coefficients $\alpha,\beta\in\mathbb{N}^{\bar{d}\times\bar{d}}$ satisfying that $1\le|\alpha|+|\beta|\le2$.

Since $N_{n,j}/\mathcal{D}_{2n}\to0$, we find that for any $j\in[L]$

$$\frac{|\sum_{i\in\mathcal{V}_j}\exp(c_{n,i})-\exp(c_j^*)|}{\mathcal{D}_{2n}}=\frac{|N_{n,j}|}{\mathcal{D}_{2n}}\to0.$$

Taking the summation of these limits leads to

$$\frac{\sum_{j=1}^{L}|\sum_{i\in\mathcal{V}_j}\exp(c_{n,i})-\exp(c_j^*)|}{\mathcal{D}_{2n}}\to0. \tag{26}$$

Now, for any indices $j\in[L]$ such that $|\mathcal{V}_j|=1$, the limits $\bar{M}_{n,j,e_{uv}}/\mathcal{D}_{2n}\to0$ lead to $\frac{\sum_{i\in\mathcal{V}_j}\exp(c_{n,i})\|\Delta\boldsymbol{Z}_{n,ij}\|_1}{\mathcal{D}_{2n}}\to0$. That result directly implies that

$$\frac{\sum_{j:|\mathcal{V}_j|=1}\sum_{i\in\mathcal{V}_j}\exp(c_{n,i})\|\Delta\boldsymbol{Z}_{n,ij}\|}{\mathcal{D}_{2n}}\to0. \tag{27}$$

Moving to indices $j\in[L]$ such that their corresponding Voronoi cells satisfy that $|\mathcal{V}_j|>1$. The limits $M_{n,j,2e_{uv}}/\mathcal{D}_{2n}\to0$ lead to

$$\frac{\sum_{i\in\mathcal{V}_j}\exp(c_{n,i})\|\Delta\boldsymbol{Z}_{n,ij}\|^2}{\mathcal{D}_{2n}}\to0. \tag{28}$$

By putting the results in equations (26), (27), and (28) together, we arrive at $1=\frac{\mathcal{D}_{2n}}{\mathcal{D}_{2n}}\to0$ as $n\to\infty$, which is a contradiction. As a consequence, at least one of the coefficients of the linear independent functions in $Q_n(\mathbb{X})/\mathcal{D}_{2n}$ does not go to 0 as $n\to\infty$.

**Step 3 - Application of the Fatou's lemma.** We denote $\bar{m}_n$ as the maximum of the absolute values of the coefficients of the linear independent functions in $Q_n(\mathbb{X})/\mathcal{D}_{2n}$. As at least one of these coefficients does not go to 0, it indicates that $1/\bar{m}_n \not\to \infty$ as $n \to \infty$. Since $\|f_{G_n} - f_{G_*}\|_{L^2(\mu)}/\mathcal{D}_{2n} \to 0$ as $n \to \infty$, we obtain $\|f_{G_n} - f_{G_*}\|_{L^2(\mu)}/(\bar{m}_n\mathcal{D}_{2n}) \to 0$. An application of the Fatou's lemma leads to:

$$0 = \lim_{n\to\infty} \frac{\|f_{G_n} - f_{\bar{G}_*}\|_{L^2(\mu)}}{\bar{m}_n\mathcal{D}_{2n}} \geq \int \liminf_{n\to\infty} \frac{\left|f_{G_n}(\mathbb{X}) - f_{\bar{G}_*}(\mathbb{X})\right|}{\bar{m}_n\mathcal{D}_{2n}} d\mu(\mathbb{X}) \geq 0.$$

That inequality demonstrates that $\liminf_{n\to\infty} \dfrac{\left|f_{G_n}(\mathbb{X}) - f_{\bar{G}_*}(\mathbb{X})\right|}{\bar{m}_n\mathcal{D}_{2n}} = 0$ for almost surely $\mathbb{X}$. As $n \to \infty$, we denote

$$\frac{\bar{M}_{n,j,\alpha}}{\bar{m}_n\mathcal{D}_{2n}} \to \bar{\lambda}_{j,\alpha}, \qquad \frac{\bar{M}_{n,j,\alpha,\beta}}{m_n\mathcal{D}_{2n}} \to \bar{\xi}_{j,\alpha,\beta}, \qquad \frac{\bar{N}_{n,j}}{m_n\mathcal{D}_{2n}} \to \bar{\tau}_j,$$

for any indices $j \in [L]$ and any coefficients $\alpha, \beta \in \mathbb{N}^{\bar{d}\times\bar{d}}$ such that $1 \leq |\alpha| + |\beta| \leq 2$. Here, we have that at least one coefficient from $\{\bar{\lambda}_{j,\alpha}, \bar{\xi}_{j,\alpha,\beta}, \bar{\tau}_j : j \in [L], \alpha, \beta \in \mathbb{N}^{\bar{d}\times\bar{d}} : 1 \leq |\alpha| + |\beta| \leq 2\}$ is different from 0 (indeed, it should be equal to 1). Given the above notations, the limit $\liminf_{n\to\infty} \dfrac{\left|f_{G_n}(\mathbb{X}) - f_{\bar{G}_*}(\mathbb{X})\right|}{m_n\mathcal{D}_{2n}} = 0$ implies that

$$\sum_{j:|\mathcal{V}_j|=1} \exp(\mathbb{X}^\top(\boldsymbol{M}_Q^0 + \boldsymbol{Z}_j^*)\mathbb{X})\Big[ \sum_{u_1,v_1=1}^{\bar{d}} \bar{\lambda}_{j,e_{u_1v_1}} \mathbb{X}^{(u_1)}\mathbb{X}^{(v_1)}(\boldsymbol{M}_Q^0 + \boldsymbol{Z}_j^*)\mathbb{X}) + \sum_{u_1,v_1=1}^{\bar{d}} \bar{M}_{n,j,e_{u_1v_1}} \mathbb{X}^{(v_1)}e_{u_1} \Big]$$

$$+ \sum_{j:|\mathcal{V}_j|>1} \exp(\mathbb{X}^\top(\boldsymbol{M}_Q^0 + \boldsymbol{Z}_j^*)\mathbb{X})\Big[ \sum_{u_1,v_1=1}^{\bar{d}} \lambda_{j,e_{u_1v_1}} \mathbb{X}^{(u_1)}\mathbb{X}^{(v_1)}(\boldsymbol{M}_Q^0 + \boldsymbol{Z}_j^*)\mathbb{X} + \sum_{u_1,v_1=1}^{\bar{d}} \bar{M}_{n,j,e_{u_1v_1}} \mathbb{X}^{(v_1)}e_{u_1}$$

$$+ \sum_{u_1,v_1=1}^{\bar{d}} \sum_{u_2,v_2=1}^{\bar{d}} \bar{\lambda}_{j,e_{u_1v_1}+e_{u_2v_2}} \mathbb{X}^{(u_1)}\mathbb{X}^{(v_1)}\mathbb{X}^{(u_2)}\mathbb{X}^{(v_2)} \exp(\mathbb{X}^\top(\boldsymbol{M}_Q^0 + \boldsymbol{Z}_j^*)\mathbb{X})(\boldsymbol{M}_V^0 + \boldsymbol{Z}_j^*)\mathbb{X}$$

$$+ \sum_{u_1,v_1=1}^{\bar{d}} \sum_{u_2,v_2=1}^{\bar{d}} \bar{\xi}_{j,e_{u_1v_1},e_{u_2v_2}} \mathbb{X}^{(u_1)}\mathbb{X}^{(v_1)}\mathbb{X}^{(v_2)} \exp(\mathbb{X}^\top(\boldsymbol{M}_Q^0 + \boldsymbol{Z}_j^*)\mathbb{X})e_{u_2} \Big]$$

$$- \sum_{j:|\mathcal{V}_j|=1} \exp(\mathbb{X}^\top(\boldsymbol{M}_Q^0 + \boldsymbol{Z}_j^*)\mathbb{X})\Big[ \sum_{u_1,v_1=1}^{\bar{d}} \bar{\lambda}_{j,e_{u_1v_1}} \mathbb{X}^{(u_1)}\mathbb{X}^{(v_1)} \Big]f_{G_n}(\mathbb{X})$$

$$- \sum_{j:|\mathcal{V}_j|>1} \exp(\mathbb{X}^\top(\boldsymbol{M}_Q^0 + \boldsymbol{Z}_j^*)\mathbb{X})\Big[ \sum_{u_1,v_1=1}^{\bar{d}} \bar{\lambda}_{j,e_{u_1v_1}} \mathbb{X}^{(u_1)}\mathbb{X}^{(v_1)}$$

$$+ \sum_{u_1,v_1=1}^{\bar{d}} \sum_{u_2,v_2=1}^{\bar{d}} \bar{\xi}_{j,e_{u_1v_1},e_{u_2v_2}} \mathbb{X}^{(u_1)}\mathbb{X}^{(v_1)}\mathbb{X}^{(u_2)}\mathbb{X}^{(v_2)} \Big]f_{G_n}(\mathbb{X})$$

$$- \sum_{j=1}^{L} \bar{\tau}_j \exp(\mathbb{X}^\top(\boldsymbol{M}_Q^0 + \boldsymbol{Z}_j^*)\mathbb{X})f_{G_n}(\mathbb{X}) + \sum_{j=1}^{L} \bar{\tau}_j \exp(\mathbb{X}^\top(\boldsymbol{M}_Q^0 + \boldsymbol{Z}_j^*)\mathbb{X})(\boldsymbol{M}_V^0 + \boldsymbol{Z}_j^*)\mathbb{X} = 0 \tag{29}$$

for almost surely $\mathbb{X}$. However, that equation implies that all the coefficients $\{\bar{\lambda}_{j,\alpha}, \bar{\xi}_{j,\alpha,\beta}, \bar{\tau}_j : j \in [L], \alpha, \beta \in \mathbb{N}^{\bar{d}\times\bar{d}} : 1 \leq |\alpha| + |\beta| \leq 2\}$ are 0. It is a contradiction. As a consequence, we obtain that

$$\lim_{\varepsilon\to0} \inf_{G\in\bar{\mathcal{G}}_{L'}(\Theta):\mathcal{D}_2(G,\bar{G}_*)\leq\varepsilon} \|f_G - f_{\bar{G}_*}\|_{L^2(\mu)}/\mathcal{D}_2(G,\bar{G}_*) > 0.$$

### A.2.2. GLOBAL PART

The result of the local part implies that we can find a positive constant $\varepsilon'$ such that

$$\inf_{G\in\bar{\mathcal{G}}_{L'}(\Theta):\mathcal{D}_2(G,\bar{G}_*)\leq\varepsilon'} \|f_G - f_{\bar{G}_*}\|_{L^2(\mu)}/\mathcal{D}_2(G,\bar{G}_*) > 0.$$

Therefore to obtain the conclusion of the theorem, we only need to prove that

$$\inf_{G \in \bar{\mathcal{G}}_{L'}(\Theta): \mathcal{D}_2(G,\bar{G}_*) > \varepsilon'} \|f_G - f_{\bar{G}_*}\|_{L^2(\mu)} / \mathcal{D}_2(G, \bar{G}_*) > 0.$$

We assume by contradiction that the above claim does not hold. It indicates that there exists a sequence of measures $G'_n := \sum_{j=1}^{\tilde{L}} \exp(c_{n,j}) \delta_{\boldsymbol{B}_{n,j} \boldsymbol{A}_{n,j}}$ in $\bar{\mathcal{G}}_{L'}(\Theta)$ such that

$$\begin{cases} \mathcal{D}_2(G'_n, \bar{G}_*) > \varepsilon' \\ \|f_{G'_n} - f_{\bar{G}_*}\|_{L^2(\mu)} / \mathcal{D}_2(G'_n, \bar{G}_*) \to 0 \end{cases}$$

as $n \to \infty$, which implies that $\|f_{G'_n} - f_{\bar{G}_*}\|_{L^2(\mu)} \to 0$ as $n \to \infty$.

Given that $\Theta$ is a compact set, there exists a mixing measure $G'$ in $\bar{\mathcal{G}}_{L'}(\Theta)$ such that one of the $G'_n$'s subsequences converges to $G'$. Since $\mathcal{D}_2(G'_n, \bar{G}_*) > \varepsilon'$, we obtain that $\mathcal{D}_2(G', \bar{G}_*) > \varepsilon'$. An application of the Fatou's lemma leads to

$$0 = \lim_{n \to \infty} \|f_{G'_n} - f_{\bar{G}_*}\|_{L^2(\mu)} \geq \int \liminf_{n \to \infty} \left\| f_{G'_n}(\mathbb{X}) - f_{\bar{G}_*}(\mathbb{X}) \right\|^2 d\mu(\mathbb{X}).$$

The above inequality indicates that $f_{G'} = f_{\bar{G}_*}$ for almost surely $\mathbb{X}$. From the identifiability property, we deduce that $G' \equiv \bar{G}_*$. It follows that $\mathcal{D}_2(G', \bar{G}_*) = 0$, contradicting the fact that $\mathcal{D}_2(G', \bar{G}_*) > \varepsilon' > 0$. Hence, the proof is completed.

**Proof for the identifiability property.** We will prove that if $f_G(\mathbb{X}) = f_{\bar{G}_*}(\mathbb{X})$ for almost surely $\mathbb{X}$, then $G \equiv \bar{G}_*$. To ease the presentation, for any mixing measure $G = \sum_{j=1}^{\tilde{L}} \exp(c_j) \delta_{\boldsymbol{B}_j \boldsymbol{A}_j} \in \mathcal{G}_{L'}(\Xi)$, we denote

$$\text{softmax}_G(u) = \frac{\exp(u)}{\sum_{j=1}^{\tilde{L}} \exp(\mathbb{X}^\top (\boldsymbol{M}_Q^0 + \boldsymbol{Z}_j)\mathbb{X} + c_j)},$$

where $u \in \{\mathbb{X}^\top (\boldsymbol{M}_Q^0 + \boldsymbol{Z}_j)\mathbb{X} + c_j : j \in [\tilde{L}]\}$. The equation $f_G(\mathbb{X}) = f_{\bar{G}_*}(\mathbb{X})$ indicates that

$$\sum_{j=1}^{L} \text{softmax}(\mathbb{X}^\top (\boldsymbol{M}_Q^0 + \boldsymbol{Z}_j^*)\mathbb{X} + c_j^*)(\boldsymbol{M}_V^0 + \boldsymbol{Z}_j^*)\mathbb{X} = \sum_{j=1}^{\tilde{L}} \text{softmax}(\mathbb{X}^\top (\boldsymbol{M}_Q^0 + \boldsymbol{Z}_j)\mathbb{X} + c_j)(\boldsymbol{M}_V^0 + \boldsymbol{Z}_j)\mathbb{X}. \quad (30)$$

That equation implies that $L = \tilde{L}$. As a consequence, we find that

$$\{\text{softmax}(\mathbb{X}^\top (\boldsymbol{M}_Q^0 + \boldsymbol{Z}_j^*)\mathbb{X} + c_j^*) : j \in [L]\} = \{\text{softmax}(\mathbb{X}^\top (\boldsymbol{M}_Q^0 + \boldsymbol{Z}_j)\mathbb{X} + c_j) : j \in [L]\},$$

for almost surely $\mathbb{X}$. By relabelling the indices, we can assume without loss of generality that for any $j \in [L]$

$$\text{softmax}(\mathbb{X}^\top (\boldsymbol{M}_Q^0 + \boldsymbol{Z}_j^*)\mathbb{X} + c_j^*) = \text{softmax}(\mathbb{X}^\top (\boldsymbol{M}_Q^0 + \boldsymbol{Z}_j)\mathbb{X} + c_j),$$

for almost surely $\mathbb{X}$. Given the invariance to translation of the softmax function, the equation (30) leads to

$$\sum_{j=1}^{L} \exp(c_j^*) \exp(\mathbb{X}^\top (\boldsymbol{M}_Q^0 + \boldsymbol{Z}_j^*)\mathbb{X})(\boldsymbol{M}_V^0 + \boldsymbol{Z}_j^*)\mathbb{X}$$
$$= \sum_{j=1}^{L} \exp(c_j) \exp(\mathbb{X}^\top (\boldsymbol{M}_Q^0 + \boldsymbol{Z}_j)\mathbb{X})(\boldsymbol{M}_V^0 + \boldsymbol{Z}_j)\mathbb{X}, \quad (31)$$

for almost surely $\mathbb{X}$.

Now, the index set $[L]$ can be partitioned into $\bar{m}$ subsets $\bar{K}_1, \bar{K}_2, \ldots, \bar{K}_{\bar{m}}$ where $\bar{m} \leq L$, such that $\exp(c_j) = \exp(c_{j'}^*)$ for any indices $j, j' \in \bar{K}_i$ and $i \in [\bar{m}]$. Thus, equation (31) can be rewritten as follows:

$$\sum_{i=1}^{\bar{m}} \sum_{j \in \bar{K}_i} \exp(c_j^*) \exp(\mathbb{X}^\top (\boldsymbol{M}_Q^0 + \boldsymbol{Z}_j^*)\mathbb{X})(\boldsymbol{M}_V^0 + \boldsymbol{Z}_j^*)\mathbb{X}$$
$$= \sum_{i=1}^{\bar{m}} \sum_{j \in \bar{K}_i} \exp(c_j) \exp(\mathbb{X}^\top (\boldsymbol{M}_Q^0 + \boldsymbol{Z}_j)\mathbb{X})(\boldsymbol{M}_V^0 + \boldsymbol{Z}_j)\mathbb{X},$$

for almost surely $\mathbb{X}$. The above equation implies that

$$\{(M_V^0 + Z_j^*)\mathbb{X} : j \in \bar{K}_i\} = \{(M_V^0 + Z_j)\mathbb{X} : j \in \bar{K}_i\},$$

for any $i \in [\bar{m}]$ and for almost surely $\mathbb{X}$. Hence, we obtain that

$$\sum_{i=1}^{\bar{m}} \sum_{j \in \bar{K}_i} \exp(c_j)\delta_{Z_j} = \sum_{i=1}^{\bar{m}} \sum_{j \in \bar{K}_i} \exp(c_j^*)\delta_{Z_j^*}.$$

As a consequence, $G \equiv G_*$ and the proof is completed.

### A.3. Proof of Theorem 5.3

Firstly, we can reduce to the case where $W_1, W_2$ are identity matrices. In particular, we may denote $\sigma_1'(X) = \sigma_1(W_1 X)$ for input $X$, and consider $\sigma_1'$ in the place of $\sigma_1$. We first start with the following result regarding the convergence rate of the regression function estimation $f_{\widetilde{G}_n}$ to the true regression function $f_{\widetilde{G}_*}$:

**Proposition A.2.** *Given the least square estimator $\widetilde{G}_n$ in equation (16), the convergence rate of the regression function estimation $f_{\widetilde{G}_n}(\cdot)$ to the true regression function $f_{\widetilde{G}_*}(\cdot)$ under the $L^2(\mu)$ norm is parametric on the sample size, that is,*

$$\|f_{\widetilde{G}_n} - f_{\widetilde{G}_*}\|_{L^2(\mu)} = \mathcal{O}_P(\sqrt{\log(n)/n}). \tag{32}$$

Given rate of $f_{\widetilde{G}_n}$ in Proposition A.2, our goal is to demonstrate the following inequality:

$$\inf_{\widetilde{G} \in \widetilde{\mathcal{G}}_{L'}(\Theta)} \|f_{\widetilde{G}} - f_{\widetilde{G}_*}\|_{L^2(\mu)}/\mathcal{D}_3(\widetilde{G}, \widetilde{G}_*) > 0.$$

We divide the proof of the above inequality into local and global parts. Before going into the proof details, let us introduce some essential assumptions on the activation function $\sigma$.

**Assumptions.** We impose the following assumptions on the activation functions $\sigma_1$ and $\sigma_2$:

*(A.1) (Algebraic independence)* If there exist parameters $(B, A)$ and $(B', A')$ such that $\sigma_2(B)\sigma_1(A) = \sigma_2(B')\sigma_1(A')$, then we obtain that $(B, A) = (B', A')$.

*(A.2) (Uniform Lipschitz)* Let $F(X; B, A) := \exp(\mathbb{X}^\top(M_Q^0 + \sigma_2(B)\sigma_1(A))\mathbb{X})(M_V^0 + \sigma_2(B)\sigma_1(A))\mathbb{X}$. Then, for any $\tau \in \{1, 2\}$, we have

$$\sum_{|\alpha|=\tau} \left| \left( \frac{\partial^{|\alpha|} F}{\partial A^{\alpha_1} \partial B^{\alpha_2}}(X; B, A) - \frac{\partial^{|\alpha|} F}{\partial A^{\alpha_1} \partial B^{\alpha_2}}(X; B', A') \right) \gamma^\alpha \right| \leq C\|(B, A) - (B', A')\|^\zeta \|\gamma\|^\tau,$$

for any vector $\gamma \in \mathbb{R}^{2dr}$ and for some positive constants $\zeta$ and $C$ which are independent of $X$ and $(B, A), (B', A')$. Here, $\alpha = (\alpha_1, \alpha_2) \in \mathbb{N}^{r \times d} \times \mathbb{N}^{d \times r}$.

*(A.3) (Strong identifiability)* For any natural number $\ell$ and distinct parameters $\{(B_j, A_j) : j \in [\ell]\}$, the functions in the set

$$\Big\{ X^{(u)}, X^{(u)} X^\top \sigma_2(B_j), X^{(u)} \sigma_1(A_j) X, X^\top \sigma_2(B_j), \sigma_1(A_j) X,$$
$$X^{(u)} X^{(v)}, X^{(u)} X^{(v)}[X^\top \sigma_2(B_j)]^2, X^{(u)} X^{(v)}[\sigma_1(A_j) X]^2,$$
$$X^{(u)} X^{(v)} X^\top \sigma_2(B_j)\sigma_1(A_j) X : j \in [\ell], \ u, v \in [d] \Big\}$$

are linearly independent for almost surely $X$.

#### A.3.1. LOCAL PART

For the local part, we prove that

$$\lim_{\varepsilon \to 0} \inf_{\widetilde{G} \in \mathcal{G}_{L'}(\Theta):\mathcal{D}_3(\widetilde{G},\widetilde{G}_*) \leq \varepsilon} \|f_{\widetilde{G}} - f_{\widetilde{G}_*}\|_{L^2(\mu)}/\mathcal{D}_3(\widetilde{G}, \widetilde{G}_*) > 0.$$

Assume that the above claim does not hold. It indicates that we can find a sequence of mixing measures $\widetilde{G}_n := \sum_{j'=1}^{L'} \exp(c_{n,j'}) \delta_{\boldsymbol{B}_{n,j'} \boldsymbol{A}_{n,j'}}$ in $\widetilde{\mathcal{G}}_{L'}(\Theta)$ such that

$$\begin{cases} \mathcal{D}_{3n} := \mathcal{D}_3(\widetilde{G}_n, \widetilde{G}_*) \to 0, \\ \|f_{\widetilde{G}_n} - f_{\widetilde{G}_*}\|_{L^2(\mu)}/\mathcal{D}_{3n} \to 0. \end{cases}$$

as $n \to \infty$. We denote $\mathcal{V}_j^n := \mathcal{V}_j(\widetilde{G}_n)$ as a Voronoi cell of $\widetilde{G}_n$ generated by the $j$-th components of $\widetilde{G}_*$. Without loss of generality, we may assume that those Voronoi cells do not depend on the sample size, i.e., $\mathcal{V}_j = \mathcal{V}_j^n$. Therefore, the Voronoi loss $\mathcal{D}_{3n}$ can be rewritten as follows:

$$\mathcal{D}_{3n} := \sum_{j'=1}^{L} \Big| \sum_{i \in \mathcal{V}_{j'}} \exp(c_{n,i}) - \exp(c_{j'}^*) \Big| + \sum_{j' \in [L]: |\mathcal{V}_{j'}|=1} \sum_{i \in \mathcal{V}_{j'}} \exp(c_{n,i})(\|\Delta \boldsymbol{B}_{n,ij'}\| + \|\Delta \boldsymbol{A}_{n,ij'}\|)$$

$$+ \sum_{j' \in [L]: |\mathcal{V}_{j'}|>1} \sum_{i \in \mathcal{V}_{j'}} \exp(c_{n,i})(\|\Delta \boldsymbol{B}_{n,ij'}\|^2 + \|\Delta \boldsymbol{A}_{n,ij'}\|^2),$$

where $\Delta \boldsymbol{B}_{n,ij'} := \boldsymbol{B}_{n,i} - \boldsymbol{B}_{j'}^*$ and $\Delta \boldsymbol{A}_{n,ij'} := \boldsymbol{A}_{n,i} - \boldsymbol{A}_{j'}^*$ for all $i \in \mathcal{V}_{j'}$ and $j' \in [L]$.

Since $\mathcal{D}_{3n} \to 0$, we have $\sum_{i \in \mathcal{V}_j} \exp(c_{n,i}) \to \exp(c_j^*)$, $\boldsymbol{B}_{n,i} \to \boldsymbol{B}_j^*$, and $\boldsymbol{A}_{n,i} \to \boldsymbol{A}_j^*$ for any $i \in \mathcal{V}_j$ and $j \in [L]$. Throughout this proof, we assume without loss of generality that $M_{K,j}^0 = I_{\bar{d}}$ with a note that our techniques can be extended to the general setting of that matrix. Now, the proof of the local part is divided into three steps as follows:

**Step 1 - Taylor expansion.** First, we define

$$Q_n(\mathbb{X}) := \Big[ \sum_{k=1}^{L} \exp(\mathbb{X}^\top (\boldsymbol{M}_Q^0 + \sigma_2(\boldsymbol{B}_k^*)\sigma_1(\boldsymbol{A}_k^*))\mathbb{X} + c_k^*) \Big] \cdot [f_{\widetilde{G}_n}(\mathbb{X}) - f_{\widetilde{G}_*}(\mathbb{X})].$$

Then, we can decompose the function $Q_n(\mathbb{X})$ as follows:

$$Q_n(\mathbb{X}) = \sum_{j=1}^{L} \sum_{i \in \mathcal{V}_j} \exp(c_{n,i}) \Big[ \exp(\mathbb{X}^\top (\boldsymbol{M}_Q^0 + \sigma_2(\boldsymbol{B}_{n,i})\sigma_1(\boldsymbol{A}_{n,i}))\mathbb{X})(\boldsymbol{M}_V^0 + \sigma_2(\boldsymbol{B}_{n,i})\sigma_1(\boldsymbol{A}_{n,i}))\mathbb{X}$$

$$- \exp(\mathbb{X}^\top (\boldsymbol{M}_Q^0 + \sigma_2(\boldsymbol{B}_j^*)\sigma_1(\boldsymbol{A}_j^*))\mathbb{X})(\boldsymbol{M}_V^0 + \sigma_2(\boldsymbol{B}_j^*)\sigma_1(\boldsymbol{A}_j^*))\mathbb{X} \Big]$$

$$- \sum_{j=1}^{L} \sum_{i \in \mathcal{V}_j} \exp(c_{n,i}) \Big[ \exp(\mathbb{X}^\top (\boldsymbol{M}_{Q,i}^0 + \sigma_2(\boldsymbol{B}_{n,i})\sigma_1(\boldsymbol{A}_{n,i}))\mathbb{X}) - \exp(\mathbb{X}^\top (\boldsymbol{M}_Q^0 + \sigma_2(\boldsymbol{B}_j^*)\sigma_1(\boldsymbol{A}_j^*))\mathbb{X}) \Big] f_{\widetilde{G}_n}(\mathbb{X})$$

$$+ \sum_{j=1}^{L} \Big( \sum_{i \in \mathcal{V}_j} \exp(c_{n,i}) - \exp(c_j^*) \Big) \exp(\mathbb{X}^\top (\boldsymbol{M}_Q^0 + \sigma_2(\boldsymbol{B}_j^*)\sigma_1(\boldsymbol{A}_j^*))\mathbb{X}) \Big[ (\boldsymbol{M}_V^0 + \sigma_2(\boldsymbol{B}_j^*)\sigma_1(\boldsymbol{A}_j^*))\mathbb{X} - f_{\widetilde{G}_n}(\mathbb{X}) \Big]$$

$$:= \widetilde{A}_n(\mathbb{X}) - \widetilde{B}_n(\mathbb{X}) + \widetilde{C}_n(\mathbb{X}). \tag{33}$$

**Decomposition of the function $\widetilde{A}_n(\mathbb{X})$.** We denote $\widetilde{U}(\mathbb{X}; \boldsymbol{B}, \boldsymbol{A}) := \exp(\mathbb{X}^\top (\boldsymbol{M}_Q^0 + \sigma_2(\boldsymbol{B})\sigma_1(\boldsymbol{A}))\mathbb{X})$ and $\widetilde{V}(\mathbb{X}; \boldsymbol{B}, \boldsymbol{A}) := (\boldsymbol{M}_V^0 + \sigma_2(\boldsymbol{B})\sigma_1(\boldsymbol{A}))\mathbb{X}$, and $F(\mathbb{X}; \boldsymbol{B}, \boldsymbol{A}) = \widetilde{U}(\mathbb{X}; \boldsymbol{B}, \boldsymbol{A})\widetilde{V}(\mathbb{X}; \boldsymbol{B}, \boldsymbol{A})$. Based on the number of elements in each Voronoi cells, we decompose the function $\widetilde{A}_n(\mathbb{X})$ as follows:

$$\widetilde{A}_n(\mathbb{X}) = \sum_{j: |\mathcal{V}_j|=1} \sum_{i \in \mathcal{V}_j} \exp(c_{n,i}) \Big[ F(\mathbb{X}; \boldsymbol{B}_{n,i}, \boldsymbol{A}_{n,i}) - F(\mathbb{X}; \boldsymbol{B}_j^*, \boldsymbol{A}_j^*) \Big]$$

$$+ \sum_{j: |\mathcal{V}_j|>1} \sum_{i \in \mathcal{V}_j} \exp(c_{n,i}) \Big[ F(\mathbb{X}; \boldsymbol{B}_{n,i}, \boldsymbol{A}_{n,i}) - F(\mathbb{X}; \boldsymbol{B}_j^*, \boldsymbol{A}_j^*) \Big]$$

$$:= \widetilde{A}_{n,1}(\mathbb{X}) + \widetilde{A}_{n,2}(\mathbb{X})$$

An application of the first-order Taylor expansion leads to

$$\widetilde{U}(\mathbb{X}; \boldsymbol{B}_{n,i}, \boldsymbol{A}_{n,i}) = \widetilde{U}(\mathbb{X}; \boldsymbol{B}_j^*, \boldsymbol{A}_j^*) + \sum_{|\alpha|=1} (\Delta \boldsymbol{A}_{n,ij})^{\alpha_1} (\Delta \boldsymbol{B}_{n,ij})^{\alpha_2} \frac{\partial^{|\alpha|}\widetilde{U}}{\partial \boldsymbol{A}^{\alpha_1} \partial \boldsymbol{B}^{\alpha_2}}(\mathbb{X}; \boldsymbol{B}_j^*, \boldsymbol{A}_j^*) + \widetilde{R}_{ij,1}(\mathbb{X}),$$

$$\widetilde{V}(\mathbb{X}; \boldsymbol{B}_{n,i}, \boldsymbol{A}_{n,i}) = \widetilde{V}(\mathbb{X}; \boldsymbol{B}_j^*, \boldsymbol{A}_j^*) + \sum_{|\alpha|=1} (\Delta \boldsymbol{A}_{n,ij})^{\alpha_1} (\Delta \boldsymbol{B}_{n,ij})^{\alpha_2} \frac{\partial^{|\alpha|}\widetilde{V}}{\partial \boldsymbol{A}^{\alpha_1} \partial \boldsymbol{B}^{\alpha_2}}(\mathbb{X}; \boldsymbol{B}_j^*, \boldsymbol{A}_j^*) + \widetilde{R}_{ij,2}(\mathbb{X}),$$

for any $i$ and $j$ such that $i \in \mathcal{V}_j$ and $|\mathcal{V}_j| = 1$. Here, the functions $\widetilde{R}_{ij,1}(\mathbb{X})$ and $\widetilde{R}_{ij,2}(\mathbb{X})$ denote the Taylor remainders. Collecting the above results leads to

$$
\begin{aligned}
\widetilde{A}_{n,1}(\mathbb{X}) &= \sum_{j:|\mathcal{V}_j|=1} \sum_{i \in \mathcal{V}_j} \frac{\exp(c_{n,i})}{\alpha!} \sum_{|\alpha|=1} \Bigg\{ (\Delta \boldsymbol{A}_{n,ij})^{\alpha_1} (\Delta \boldsymbol{B}_{n,ij})^{\alpha_2} \frac{\partial^{|\alpha|}\widetilde{U}}{\partial \boldsymbol{A}^{\alpha_1} \partial \boldsymbol{B}^{\alpha_2}}(\mathbb{X}; \boldsymbol{B}_j^*, \boldsymbol{A}_j^*) \widetilde{V}(\mathbb{X}; \boldsymbol{B}_j^*, \boldsymbol{A}_j^*) \\
&\quad + (\Delta \boldsymbol{A}_{n,ij})^{\alpha_1} (\Delta \boldsymbol{B}_{n,ij})^{\alpha_2} \frac{\partial^{|\alpha|}\widetilde{V}}{\partial \boldsymbol{A}^{\alpha_1} \partial \boldsymbol{B}^{\alpha_2}}(\mathbb{X}; \boldsymbol{B}_j^*, \boldsymbol{A}_j^*) \widetilde{U}(\mathbb{X}; \boldsymbol{B}_j^*, \boldsymbol{A}_j^*) \Bigg\} + \widetilde{R}_{n,1}(\mathbb{X}) \\
&= \sum_{j:|\mathcal{V}_j|=1} \sum_{|\alpha|=1} \Bigg\{ \widetilde{M}_{n,j,\alpha} \frac{\partial^{|\alpha|}\widetilde{U}}{\partial \boldsymbol{A}^{\alpha_1} \partial \boldsymbol{B}^{\alpha_2}}(\mathbb{X}; \boldsymbol{B}_j^*, \boldsymbol{A}_j^*) \widetilde{V}(\mathbb{X}; \boldsymbol{B}_j^*, \boldsymbol{A}_j^*) \\
&\quad + \widetilde{M}_{n,j,\alpha} \frac{\partial^{|\alpha|}\widetilde{V}}{\partial \boldsymbol{A}^{\alpha_1} \partial \boldsymbol{B}^{\alpha_2}}(\mathbb{X}; \boldsymbol{B}_j^*, \boldsymbol{A}_j^*) \widetilde{U}(\mathbb{X}; \boldsymbol{B}_j^*, \boldsymbol{A}_j^*) \Bigg\} + \widetilde{R}_{n,1}(\mathbb{X})
\end{aligned}
$$

where the function $\widetilde{R}_{n,1}(\mathbb{X})$ satisfies that $\widetilde{R}_{n,1}(\mathbb{X})/\mathcal{D}_{3n} \to 0$. It is due to the uniform Lipschitz property of the function $F$. In the above display, the formulations of the coefficients $\widetilde{M}_{n,j,\alpha}$ are given by:

$$\widetilde{M}_{n,j,\alpha_1,\alpha_2} = \sum_{i \in \mathcal{V}_j} \frac{\exp(c_{n,i})}{\alpha!} (\Delta \boldsymbol{A}_{n,ij})^{\alpha_1} (\Delta \boldsymbol{B}_{n,ij})^{\alpha_2},$$

for any $|\alpha| = 1$.

Moving to the function $\widetilde{A}_{n,2}(\mathbb{X})$, an application of the Taylor expansion up to the second order leads to

$$
\begin{aligned}
\widetilde{A}_{n,2}(\mathbb{X}) &= \sum_{j:|\mathcal{V}_j|>1} \sum_{1 \le |\alpha| \le 2} \Bigg\{ \widetilde{M}_{n,j,\alpha_1,\alpha_2} \frac{\partial^{|\alpha|}\widetilde{U}}{\partial \boldsymbol{A}^{\alpha_1} \partial \boldsymbol{B}^{\alpha_2}}(\mathbb{X}; \boldsymbol{B}_j^*, \boldsymbol{A}_j^*) \widetilde{V}(\mathbb{X}; \boldsymbol{B}_j^*, \boldsymbol{A}_j^*) \\
&\quad + \widetilde{M}_{n,j,\alpha_1,\alpha_2} \frac{\partial^{|\alpha|}\widetilde{V}}{\partial \boldsymbol{A}^{\alpha_1} \partial \boldsymbol{B}^{\alpha_2}}(\mathbb{X}; \boldsymbol{B}_j^*, \boldsymbol{A}_j^*) \widetilde{U}(\mathbb{X}; \boldsymbol{B}_j^*, \boldsymbol{A}_j^*) \Bigg\} \\
&\quad + \sum_{|\alpha|=1,|\beta|=1} \widetilde{M}_{n,j,\alpha_1,\beta_1,\alpha_2,\beta_2} \frac{\partial^{|\alpha|}\widetilde{U}}{\partial \boldsymbol{A}^{\alpha_1} \partial \boldsymbol{B}^{\alpha_2}}(\mathbb{X}; \boldsymbol{B}_j^*, \boldsymbol{A}_j^*) \frac{\partial^{|\beta|}\widetilde{V}}{\partial \boldsymbol{A}^{\beta_1} \partial \boldsymbol{B}^{\beta_2}}(\mathbb{X}; \boldsymbol{B}_j^*, \boldsymbol{A}_j^*) + \widetilde{R}_{n,2}(\mathbb{X})
\end{aligned}
$$

where the remainder $\widetilde{R}_{n,2}(\mathbb{X})$ satisfies that $\widetilde{R}_{n,2}(\mathbb{X})/\mathcal{D}_{3n} \to 0$. In this equation, the coefficients $\widetilde{M}_{n,j,\alpha_1,\alpha_2}$ and $\widetilde{M}_{n,j,\alpha_1,\beta_1,\alpha_2,\beta_2}$ take the following forms:

$$\widetilde{M}_{n,j,\alpha_1,\alpha_2} = \sum_{i \in \mathcal{V}_j} \frac{\exp(c_{n,i})}{\alpha!} (\Delta \boldsymbol{A}_{n,ij})^{\alpha_1} (\Delta \boldsymbol{B}_{n,ij})^{\alpha_2},$$

for any $|\alpha| = 2$ and

$$\widetilde{M}_{n,j,\alpha_1,\beta_1,\alpha_2,\beta_2} = \sum_{i \in \mathcal{V}_j} \frac{\exp(c_{n,i})}{\alpha!\beta!} (\Delta \boldsymbol{A}_{n,ij})^{\alpha_1+\beta_1} (\Delta \boldsymbol{B}_{n,ij})^{\alpha_2+\beta_2},$$

for any $|\alpha| = |\beta| = 1$. Simple algebra leads to the following formulations of the partial derivatives of $\widetilde{U}(\mathbb{X}; \boldsymbol{B}, \boldsymbol{A})$ and $\widetilde{V}(\mathbb{X}; \boldsymbol{B}, \boldsymbol{A})$:

$$\frac{\partial \widetilde{U}}{\partial \boldsymbol{A}^{(u)}}(\mathbb{X}; \boldsymbol{B}, \boldsymbol{A}) = \mathbb{X}^{(u)} \sigma_1'(\boldsymbol{A}^{(u)}) \mathbb{X}^\top \sigma_2(\boldsymbol{B}) \exp(\mathbb{X}^\top (\boldsymbol{M}_Q^0 + \sigma_2(\boldsymbol{B})\sigma_1(\boldsymbol{A}))\mathbb{X}),$$

$$\frac{\partial \widetilde{U}}{\partial \boldsymbol{B}^{(u)}}(\mathbb{X}; \boldsymbol{B}, \boldsymbol{A}) = \mathbb{X}^{(u)} \sigma_2'(\boldsymbol{B}^{(u)}) \sigma_1(\boldsymbol{A}) \mathbb{X} \exp(\mathbb{X}^\top (\boldsymbol{M}_Q^0 + \sigma_2(\boldsymbol{B})\sigma_1(\boldsymbol{A}))\mathbb{X}),$$

$$\frac{\partial^2 \widetilde{U}}{\partial \boldsymbol{A}^{(u)} \partial \boldsymbol{A}^{(v)}}(\mathbb{X}; \boldsymbol{B}, \boldsymbol{A}) = \left[ \mathbb{X}^{(u)} \mathbb{X}^{(v)} \sigma_1'(\boldsymbol{A}^{(u)}) \sigma_1'(\boldsymbol{A}^{(v)}) \left( \mathbb{X}^\top \sigma_2(\boldsymbol{B}) \right)^2 + \mathbf{1}_{\{u=v\}} \mathbb{X}^{(u)} \sigma_1''(\boldsymbol{A}^{(u)}) \mathbb{X}^\top \sigma_2(\boldsymbol{B}) \right]$$
$$\times \exp(\mathbb{X}^\top (\boldsymbol{M}_Q^0 + \sigma_2(\boldsymbol{B})\sigma_1(\boldsymbol{A}))\mathbb{X}),$$

$$\frac{\partial^2 \widetilde{U}}{\partial \boldsymbol{B}^{(u)} \partial \boldsymbol{B}^{(v)}}(\mathbb{X}; \boldsymbol{B}, \boldsymbol{A}) = \left[ \mathbb{X}^{(u)} \mathbb{X}^{(v)} \sigma_2'(\boldsymbol{B}^{(u)}) \sigma_2'(\boldsymbol{B}^{(v)}) \left( \sigma_1(\boldsymbol{A}) \mathbb{X} \right)^2 + \mathbf{1}_{\{u=v\}} \mathbb{X}^{(u)} \sigma_2''(\boldsymbol{B}^{(u)}) \sigma_1(\boldsymbol{A}) \mathbb{X} \right]$$
$$\times \exp(\mathbb{X}^\top (\boldsymbol{M}_Q^0 + \sigma_2(\boldsymbol{B})\sigma_1(\boldsymbol{A}))\mathbb{X}),$$

$$\frac{\partial^2 \widetilde{U}}{\partial \boldsymbol{A}^{(u)} \partial \boldsymbol{B}^{(v)}}(\mathbb{X}; \boldsymbol{B}, \boldsymbol{A}) = \left[ \mathbb{X}^{(u)} \mathbb{X}^{(v)} \sigma_1'(\boldsymbol{A}^{(u)}) \sigma_2'(\boldsymbol{B}^{(v)}) + \mathbb{X}^{(u)} \mathbb{X}^{(v)} \sigma_1'(\boldsymbol{A}^{(u)}) \sigma_2'(\boldsymbol{B}^{(v)}) \mathbb{X}^\top \sigma_2(\boldsymbol{B})\sigma_1(\boldsymbol{A})\mathbb{X} \right]$$
$$\times \exp(\mathbb{X}^\top (\boldsymbol{M}_Q^0 + \sigma_2(\boldsymbol{B})\sigma_1(\boldsymbol{A}))\mathbb{X}),$$

$$\frac{\partial \widetilde{V}}{\partial \boldsymbol{A}^{(u)}}(\mathbb{X}; \boldsymbol{B}, \boldsymbol{A}) = \mathbb{X}^{(u)} \sigma_1'(\boldsymbol{A}^{(u)}) \sigma_2(\boldsymbol{B}),$$

$$\frac{\partial \widetilde{V}}{\partial \boldsymbol{B}^{(u)}}(\mathbb{X}; \boldsymbol{B}, \boldsymbol{A}) = \sigma_1(A) \mathbb{X} \sigma_2'(\boldsymbol{B}^{(u)}) e_u,$$

$$\frac{\partial^2 \widetilde{V}}{\partial \boldsymbol{A}^{(u)} \partial \boldsymbol{A}^{(v)}}(\mathbb{X}; \boldsymbol{B}, \boldsymbol{A}) = \mathbf{1}_{\{u=v\}} \mathbb{X}^{(u)} \sigma_1''(\boldsymbol{A}^{(u)}) \sigma_2(\boldsymbol{B}),$$

$$\frac{\partial^2 \widetilde{V}}{\partial \boldsymbol{B}^{(u)} \partial \boldsymbol{B}^{(v)}}(\mathbb{X}; \boldsymbol{B}, \boldsymbol{A}) = \mathbf{1}_{\{u=v\}} \sigma_1(A) \mathbb{X} \sigma_2''(\boldsymbol{B}^{(u)}) e_u,$$

$$\frac{\partial^2 \widetilde{V}}{\partial \boldsymbol{A}^{(u)} \partial \boldsymbol{B}^{(v)}}(\mathbb{X}; \boldsymbol{B}, \boldsymbol{A}) = \mathbb{X}^{(u)} \sigma_1'(\boldsymbol{A}^{(u)}) \sigma_2'(\boldsymbol{B}^{(v)}) e_v.$$

Plugging these formulations into the functions $\widetilde{A}_{n,1}(\mathbb{X})$ and $\widetilde{A}_{n,2}(\mathbb{X})$, we obtain that

$$\widetilde{A}_{n,1}(\mathbb{X}) = \sum_{j:|\mathcal{V}_j|=1} \exp(\mathbb{X}^\top (\boldsymbol{M}_Q^0 + \sigma_2(\boldsymbol{B}_j^*)\sigma_1(\boldsymbol{A}_j^*))\mathbb{X}) \Big[ \left( L_{n,1,j}^\top \mathbb{X}\mathbb{X}^\top \sigma_2(\boldsymbol{B}_j^*) + L_{n,2,j}^\top \mathbb{X}\sigma_1(\boldsymbol{A}_j^*)\mathbb{X} \right)$$
$$(\boldsymbol{M}_V^0 + \sigma_2(\boldsymbol{B}_j^*)\sigma_1(\boldsymbol{A}_j^*))\mathbb{X} + L_{n,1,j}^\top \mathbb{X}\sigma_2(\boldsymbol{B}_j^*) + \sigma_1(\boldsymbol{A}_j^*)\mathbb{X}L_{n,2,j} \Big] + \widetilde{R}_{n,1}(\mathbb{X}),$$

$$\widetilde{A}_{n,2}(\mathbb{X}) = \sum_{j:|\mathcal{V}_j|>1} \exp(\mathbb{X}^\top (\boldsymbol{M}_Q^0 + \sigma_2(\boldsymbol{B}_j^*)\sigma_1(\boldsymbol{A}_j^*))\mathbb{X}) \Big[ \left( L_{n,1,j}^\top \mathbb{X}\mathbb{X}^\top \sigma_2(\boldsymbol{B}_j^*) + L_{n,2,j}^\top \mathbb{X}\sigma_1(\boldsymbol{A}_j^*)\mathbb{X} \right.$$
$$+ \mathbb{X}^\top L_{n,3,j} \mathbb{X}(\mathbb{X}^\top \sigma_2(\boldsymbol{B}_j^*))^2 + L_{n,4,j}^\top \mathbb{X}\mathbb{X}^\top \sigma_2(\boldsymbol{B}_j^*) + \mathbb{X}^\top L_{n,5,j} \mathbb{X}(\sigma_1(\boldsymbol{A}_j^*)\mathbb{X})^2 + L_{n,6,j}^\top \mathbb{X}\sigma_1(\boldsymbol{A}_j^*)\mathbb{X}$$
$$+ \mathbb{X}^\top L_{n,7,j} \mathbb{X} + \mathbb{X}^\top L_{n,7,j} \mathbb{X}\mathbb{X}^\top \sigma_2(\boldsymbol{B}_j^*)\sigma_1(\boldsymbol{A}_j^*)\mathbb{X} \big) \times (\boldsymbol{M}_V^0 + \sigma_2(\boldsymbol{B}_j^*)\sigma_1(\boldsymbol{A}_j^*))\mathbb{X} + L_{n,1,j}^\top \mathbb{X}\sigma_2(\boldsymbol{B}_j^*)$$
$$+ \sigma_1(\boldsymbol{A}_j^*)\mathbb{X}L_{n,2,j} + L_{n,4,j}^\top \mathbb{X}\sigma_2(\boldsymbol{B}_j^*) + \sigma_1(\boldsymbol{A}_j^*)\mathbb{X}L_{n,6,j} + L_{n,7,j}^\top \mathbb{X} \Big] + \widetilde{R}_{n,2}(\mathbb{X}),$$

where the formulations of $L_{n,1,j}, L_{n,2,j}, \ldots, L_{n,6,j}$ are given by:

$$
\begin{aligned}
L_{n,1,j} &:= (\widetilde{M}_{n,j,e_u,0_d}\sigma_1'(\boldsymbol{A}^{(u)}))_{u=1}^{\bar{d}}, \\
L_{n,2,j} &:= (\widetilde{M}_{n,j,0_d,e_u}\sigma_2'(\boldsymbol{B}^{(u)}))_{u=1}^{\bar{d}}, \\
L_{n,3,j} &:= (\widetilde{M}_{n,j,e_u+e_v,0_d}\sigma_1'(\boldsymbol{A}^{(u)})\sigma_1'(\boldsymbol{A}^{(v)}))_{u,v=1}^{\bar{d}}, \\
L_{n,4,j} &:= (\widetilde{M}_{n,j,2e_u,0_d}\sigma_1''(\boldsymbol{A}^{(u)}))_{u=1}^{\bar{d}}, \\
L_{n,5,j} &:= (\widetilde{M}_{n,j,0_d,e_u+e_v}\sigma_2'(\boldsymbol{B}^{(u)})\sigma_2'(\boldsymbol{B}^{(v)}))_{u,v=1}^{\bar{d}}, \\
L_{n,6,j} &:= (\widetilde{M}_{n,j,0_d,2e_u}\sigma_2''(\boldsymbol{B}^{(u)}))_{u=1}^{\bar{d}}, \\
L_{n,7,j} &:= (\widetilde{M}_{n,j,e_u,e_v}\sigma_1'(\boldsymbol{A}^{(u)})\sigma_2'(\boldsymbol{B}^{(v)}))_{u,v=1}^{\bar{d}}.
\end{aligned}
$$

In these equations, $e_u$ is denoted as the vector in $\mathbb{R}^{\bar{d}}$ such that its $u$-th element is 1 while its other elements are 0 for any $1 \leq u \leq \bar{d}$. Furthermore, $e_{uv}$ is denoted as matrix in $\mathbb{R}^{\bar{d}\times\bar{d}}$ with its $uv$-th entry is 1 while other entries are zero.

**Decomposition of the function $\widetilde{B}_n(\mathbb{X})$.** Moving to the function $\widetilde{B}_n(\mathbb{X})$, we can decompose this function as follows:

$$
\begin{aligned}
\widetilde{B}_n(\mathbb{X}) &= \sum_{j:|\mathcal{V}_j|=1}\sum_{i\in\mathcal{V}_j}\exp(c_{n,i})\Big[\widetilde{U}(\mathbb{X};\boldsymbol{B}_{n,i},\boldsymbol{A}_{n,i}) - \widetilde{U}(\mathbb{X};\boldsymbol{B}_j^*\boldsymbol{A}_j^*)\Big]f_{\widetilde{G}_n}(\mathbb{X}) \\
&\quad + \sum_{j:|\mathcal{V}_j|>1}\sum_{i\in\mathcal{V}_j}\exp(c_{n,i})\Big[\widetilde{U}(\mathbb{X};\boldsymbol{B}_{n,i},\boldsymbol{A}_{n,i}) - \widetilde{U}(\mathbb{X};\boldsymbol{B}_j^*\boldsymbol{A}_j^*)\Big]f_{\widetilde{G}_n}(\mathbb{X}) \\
&:= \widetilde{B}_{n,1}(\mathbb{X}) + \widetilde{B}_{n,2}(\mathbb{X}).
\end{aligned}
$$

An application of the Taylor expansions up to the first order for $\widetilde{B}_{n,1}(\mathbb{X})$ and the second order for $\widetilde{B}_{n,2}(\mathbb{X})$ leads to

$$
\begin{aligned}
\widetilde{B}_{n,1}(\mathbb{X}) &= \sum_{j:|\mathcal{V}_j|=1}\sum_{|\alpha|=1}\widetilde{M}_{n,j,\alpha_1,\alpha_2}\frac{\partial^{|\alpha|}\widetilde{U}}{\partial\boldsymbol{A}^{\alpha_1}\partial\boldsymbol{B}^{\alpha_2}}(\mathbb{X};\boldsymbol{B}_j^*,\boldsymbol{A}_j^*)f_{\widetilde{G}_n}(\mathbb{X}) + \widetilde{R}_{n,3}(\mathbb{X}), \\
\widetilde{B}_{n,2}(\mathbb{X}) &= \sum_{j:|\mathcal{V}_j|=1}\sum_{1\leq|\alpha|\leq 2}\widetilde{M}_{n,j,\alpha_1,\alpha_2}\frac{\partial^{|\alpha|}\widetilde{U}}{\partial\boldsymbol{A}^{\alpha_1}\partial\boldsymbol{B}^{\alpha_2}}(\mathbb{X};\boldsymbol{B}_j^*,\boldsymbol{A}_j^*)f_{\widetilde{G}_n}(\mathbb{X}) + \widetilde{R}_{n,4}(\mathbb{X})
\end{aligned}
$$

where the Taylor remainders $\widetilde{R}_{n,3}(\mathbb{X}), \widetilde{R}_{n,4}(\mathbb{X})$ satisfy that $\widetilde{R}_{n,3}(\mathbb{X})/\mathcal{D}_{3n} \to 0$ and $\widetilde{R}_{n,4}(\mathbb{X})/\mathcal{D}_{3n} \to 0$. Direct calculation leads to

$$
\begin{aligned}
\widetilde{B}_{n,1}(\mathbb{X}) &= \sum_{j:|\mathcal{V}_j|=1}\exp(\mathbb{X}^\top(\boldsymbol{M}_Q^0 + \sigma_2(\boldsymbol{B}_j^*)\sigma_1(\boldsymbol{A}_j^*))\mathbb{X})\Big[L_{n,1,j}^\top\mathbb{X}\mathbb{X}^\top\sigma_2(\boldsymbol{B}_j^*) + L_{n,2,j}^\top\mathbb{X}\sigma_1(\boldsymbol{A}_j^*)\mathbb{X}\Big]f_{\widetilde{G}_n}(\mathbb{X}) + \widetilde{R}_{n,3}(\mathbb{X}), \\
\widetilde{B}_{n,2}(\mathbb{X}) &= \sum_{j:|\mathcal{V}_j|>1}\exp(\mathbb{X}^\top(\boldsymbol{M}_Q^0 + \sigma_2(\boldsymbol{B}_j^*)\sigma_1(\boldsymbol{A}_j^*))\mathbb{X})\Big[L_{n,1,j}^\top\mathbb{X}\mathbb{X}^\top\sigma_2(\boldsymbol{B}_j^*) + L_{n,2,j}^\top\mathbb{X}\sigma_1(\boldsymbol{A}_j^*)\mathbb{X} \\
&\quad + \mathbb{X}^\top L_{n,3,j}\mathbb{X}(\mathbb{X}^\top\sigma_2(\boldsymbol{B}_j^*))^2 + L_{n,4,j}^\top\mathbb{X}\mathbb{X}^\top\sigma_2(\boldsymbol{B}_j^*) + \mathbb{X}^\top L_{n,5,j}\mathbb{X}(\sigma_1(\boldsymbol{A}_j^*)\mathbb{X})^2 + L_{n,6,j}^\top\mathbb{X}\sigma_1(\boldsymbol{A}_j^*)\mathbb{X} \\
&\quad + \mathbb{X}^\top L_{n,7,j}\mathbb{X} + \mathbb{X}^\top L_{n,7,j}\mathbb{X}\mathbb{X}^\top\sigma_2(\boldsymbol{B}_j^*)\sigma_1(\boldsymbol{A}_j^*)\mathbb{X}\Big]f_{\widetilde{G}_n}(\mathbb{X}) + \widetilde{R}_{n,4}(\mathbb{X}),
\end{aligned}
$$

Putting all the above results together, we can represent the function $Q_n(\mathbb{X})$ as follows:

$$Q_n(\mathbb{X}) = \sum_{j:|\mathcal{V}_j|=1} \exp(\mathbb{X}^\top(\boldsymbol{M}_Q^0 + \sigma_2(\boldsymbol{B}_j^*)\sigma_1(\boldsymbol{A}_j^*))\mathbb{X})\Big[\big(L_{n,1,j}^\top\mathbb{X}\mathbb{X}^\top\sigma_2(\boldsymbol{B}_j^*) + L_{n,2,j}^\top\mathbb{X}\sigma_1(\boldsymbol{A}_j^*)\mathbb{X}\big)(\boldsymbol{M}_V^0 \tag{34}$$

$$+ \sigma_2(\boldsymbol{B}_j^*)\sigma_1(\boldsymbol{A}_j^*))\mathbb{X} + L_{n,1,j}^\top\mathbb{X}\sigma_2(\boldsymbol{B}_j^*) + \sigma_1(\boldsymbol{A}_j^*)\mathbb{X}L_{n,2,j}\Big]$$

$$+ \sum_{j:|\mathcal{V}_j|>1} \exp(\mathbb{X}^\top(\boldsymbol{M}_Q^0 + \sigma_2(\boldsymbol{B}_j^*)\sigma_1(\boldsymbol{A}_j^*))\mathbb{X})\Big[\big(L_{n,1,j}^\top\mathbb{X}\mathbb{X}^\top\sigma_2(\boldsymbol{B}_j^*) + L_{n,2,j}^\top\mathbb{X}\sigma_1(\boldsymbol{A}_j^*)\mathbb{X} \tag{35}$$

$$+ \mathbb{X}^\top L_{n,3,j}\mathbb{X}(\mathbb{X}^\top\sigma_2(\boldsymbol{B}_j^*))^2 + L_{n,4,j}^\top\mathbb{X}\mathbb{X}^\top\sigma_2(\boldsymbol{B}_j^*) + \mathbb{X}^\top L_{n,5,j}\mathbb{X}(\sigma_1(\boldsymbol{A}_j^*)\mathbb{X})^2 + L_{n,6,j}^\top\mathbb{X}\sigma_1(\boldsymbol{A}_j^*)\mathbb{X}$$

$$+ \mathbb{X}^\top L_{n,7,j}\mathbb{X} + \mathbb{X}^\top L_{n,7,j}\mathbb{X}\mathbb{X}^\top\sigma_2(\boldsymbol{B}_j^*)\sigma_1(\boldsymbol{A}_j^*)\mathbb{X}) \times (\boldsymbol{M}_V^0 + \sigma_2(\boldsymbol{B}_j^*)\sigma_1(\boldsymbol{A}_j^*))\mathbb{X} + L_{n,1,j}^\top\mathbb{X}\sigma_2(\boldsymbol{B}_j^*)$$

$$+ \sigma_1(\boldsymbol{A}_j^*)\mathbb{X}L_{n,2,j} + L_{n,4,j}^\top\mathbb{X}\sigma_2(\boldsymbol{B}_j^*) + \sigma_1(\boldsymbol{A}_j^*)\mathbb{X}L_{n,6,j} + L_{n,7,j}^\top\mathbb{X}\Big]$$

$$- \sum_{j:|\mathcal{V}_j|=1} \exp(\mathbb{X}^\top(\boldsymbol{M}_Q^0 + \sigma_2(\boldsymbol{B}_j^*)\sigma_1(\boldsymbol{A}_j^*))\mathbb{X})\Big[L_{n,1,j}^\top\mathbb{X}\mathbb{X}^\top\sigma_2(\boldsymbol{B}_j^*) + L_{n,2,j}^\top\mathbb{X}\sigma_1(\boldsymbol{A}_j^*)\mathbb{X}\Big]f_{\widetilde{G}_n}(\mathbb{X})$$

$$- \sum_{j:|\mathcal{V}_j|>1} \exp(\mathbb{X}^\top(\boldsymbol{M}_Q^0 + \sigma_2(\boldsymbol{B}_j^*)\sigma_1(\boldsymbol{A}_j^*))\mathbb{X})\Big[L_{n,1,j}^\top\mathbb{X}\mathbb{X}^\top\sigma_2(\boldsymbol{B}_j^*) + L_{n,2,j}^\top\mathbb{X}\sigma_1(\boldsymbol{A}_j^*)\mathbb{X} \tag{36}$$

$$+ \mathbb{X}^\top L_{n,3,j}\mathbb{X}(\mathbb{X}^\top\sigma_2(\boldsymbol{B}_j^*))^2 + L_{n,4,j}^\top\mathbb{X}\mathbb{X}^\top\sigma_2(\boldsymbol{B}_j^*) + \mathbb{X}^\top L_{n,5,j}\mathbb{X}(\sigma_1(\boldsymbol{A}_j^*)\mathbb{X})^2$$

$$+ L_{n,6,j}^\top\mathbb{X}\sigma_1(\boldsymbol{A}_j^*)\mathbb{X} + \mathbb{X}^\top L_{n,7,j}\mathbb{X} + \mathbb{X}^\top L_{n,7,j}\mathbb{X}\mathbb{X}^\top\sigma_2(\boldsymbol{B}_j^*)\sigma_1(\boldsymbol{A}_j^*)\mathbb{X}\Big]f_{\widetilde{G}_n}(\mathbb{X})$$

$$+ \sum_{j=1}^{L} \widetilde{N}_{n,j}\exp(\mathbb{X}^\top(\boldsymbol{M}_Q^0 + \sigma_2(\boldsymbol{B}_j^*)\sigma_1(\boldsymbol{A}_j^*))\mathbb{X})\Big[(\boldsymbol{M}_V^0 + \boldsymbol{B}_j^*\boldsymbol{A}_j^*)\mathbb{X} - f_{\widetilde{G}_n}(\mathbb{X})\Big]$$

$$+ \widetilde{R}_{n,1}(\mathbb{X}) + \widetilde{R}_{n,2}(\mathbb{X}) - \widetilde{R}_{n,3}(\mathbb{X}) - \widetilde{R}_{n,4}(\mathbb{X}), \tag{37}$$

where $\widetilde{N}_{n,j} := \sum_{i\in\mathcal{V}_j}\exp(c_{n,i}) - \exp(c_j^*)$ for any $j \in [L]$.

**Step 2 - Non-vanishing coefficients.** As indicated in equation (37), the ratio $Q_n(\mathbb{X})/\mathcal{D}_{3n}$ can be expressed as a linear combination of the following independent functions:

$$\widetilde{U}(\mathbb{X};\boldsymbol{B}_j^*\boldsymbol{A}_j^*)\mathbb{X}^{(u)}\mathbb{X}^\top\sigma_2(\boldsymbol{B}_j^*)\widetilde{V}(\mathbb{X};\boldsymbol{B}_j^*,\boldsymbol{A}_j^*), \quad \widetilde{U}(\mathbb{X};\boldsymbol{B}_j^*\boldsymbol{A}_j^*)\mathbb{X}^{(u)}\sigma_1(\boldsymbol{A}_j^*)\mathbb{X}\widetilde{V}(\mathbb{X};\boldsymbol{B}_j^*,\boldsymbol{A}_j^*), \quad \widetilde{U}(\mathbb{X};\boldsymbol{B}_j^*\boldsymbol{A}_j^*)\mathbb{X}^{(u)}\sigma_2(\boldsymbol{B}_j^*)$$

$$\widetilde{U}(\mathbb{X};\boldsymbol{B}_j^*\boldsymbol{A}_j^*)\sigma_1(\boldsymbol{A}_j^*)\mathbb{X}e_u, \quad \widetilde{U}(\mathbb{X};\boldsymbol{B}_j^*\boldsymbol{A}_j^*)\mathbb{X}^{(u)}\mathbb{X}^{(v)}(\mathbb{X}^\top\sigma_2(\boldsymbol{B}_j^*))^2\widetilde{V}(\mathbb{X};\boldsymbol{B}_j^*,\boldsymbol{A}_j^*),$$

$$\widetilde{U}(\mathbb{X};\boldsymbol{B}_j^*\boldsymbol{A}_j^*)\mathbb{X}^{(u)}\mathbb{X}^{(v)}(\sigma_1(\boldsymbol{A}_j^*)\mathbb{X})^2\widetilde{V}(\mathbb{X};\boldsymbol{B}_j^*,\boldsymbol{A}_j^*), \quad \widetilde{U}(\mathbb{X};\boldsymbol{B}_j^*\boldsymbol{A}_j^*)\mathbb{X}^{(u)}\sigma_1(\boldsymbol{A}_j^*)\mathbb{X})\widetilde{V}(\mathbb{X};\boldsymbol{B}_j^*,\boldsymbol{A}_j^*),$$

$$\widetilde{U}(\mathbb{X};\boldsymbol{B}_j^*\boldsymbol{A}_j^*)\mathbb{X}^{(u)}\mathbb{X}^{(v)}\widetilde{V}(\mathbb{X};\boldsymbol{B}_j^*,\boldsymbol{A}_j^*), \quad \widetilde{U}(\mathbb{X};\boldsymbol{B}_j^*\boldsymbol{A}_j^*)\mathbb{X}^{(u)}\mathbb{X}^{(v)}\mathbb{X}^\top\sigma_2(\boldsymbol{B}_j^*)\sigma_1(\boldsymbol{A}_j^*)\mathbb{X}\widetilde{V}(\mathbb{X};\boldsymbol{B}_j^*,\boldsymbol{A}_j^*),$$

$$\widetilde{U}(\mathbb{X};\boldsymbol{B}_j^*\boldsymbol{A}_j^*)\mathbb{X}^{(u)}\mathbb{X}^\top\sigma_2(\boldsymbol{B}_j^*)f_{\widetilde{G}_n}(\mathbb{X}), \quad \widetilde{U}(\mathbb{X};\boldsymbol{B}_j^*\boldsymbol{A}_j^*)\mathbb{X}^{(u)}\sigma_1(\boldsymbol{A}_j^*)\mathbb{X}f_{\widetilde{G}_n}(\mathbb{X}),$$

$$\widetilde{U}(\mathbb{X};\boldsymbol{B}_j^*\boldsymbol{A}_j^*)\mathbb{X}^{(u)}\mathbb{X}^{(v)}(\mathbb{X}^\top\sigma_2(\boldsymbol{B}_j^*))^2 f_{\widetilde{G}_n}(\mathbb{X}), \quad \widetilde{U}(\mathbb{X};\boldsymbol{B}_j^*\boldsymbol{A}_j^*)\mathbb{X}^{(u)}\mathbb{X}^\top\sigma_2(\boldsymbol{B}_j^*)f_{\widetilde{G}_n}(\mathbb{X}),$$

$$\widetilde{U}(\mathbb{X};\boldsymbol{B}_j^*\boldsymbol{A}_j^*)\mathbb{X}^{(u)}\mathbb{X}^{(v)}(\sigma_1(\boldsymbol{A}_j^*)\mathbb{X})^2 f_{\widetilde{G}_n}(\mathbb{X}), \quad \widetilde{U}(\mathbb{X};\boldsymbol{B}_j^*\boldsymbol{A}_j^*)\mathbb{X}^{(u)}\sigma_1(\boldsymbol{A}_j^*)\mathbb{X})f_{\widetilde{G}_n}(\mathbb{X}),$$

$$\widetilde{U}(\mathbb{X};\boldsymbol{B}_j^*\boldsymbol{A}_j^*)\mathbb{X}^{(u)}\mathbb{X}^{(v)}f_{\widetilde{G}_n}(\mathbb{X}), \quad \widetilde{U}(\mathbb{X};\boldsymbol{B}_j^*\boldsymbol{A}_j^*)\mathbb{X}^{(u)}\mathbb{X}^{(v)}\mathbb{X}^\top\sigma_2(\boldsymbol{B}_j^*)\sigma_1(\boldsymbol{A}_j^*)\mathbb{X}f_{\widetilde{G}_n}(\mathbb{X}),$$

$$\widetilde{U}(\mathbb{X};\boldsymbol{B}_j^*\boldsymbol{A}_j^*)\widetilde{V}(\mathbb{X};\boldsymbol{B}_j^*,\boldsymbol{A}_j^*), \quad \widetilde{U}(\mathbb{X};\boldsymbol{B}_j^*\boldsymbol{A}_j^*)f_{\widetilde{G}_n}(\mathbb{X}),$$

for any indices $1 \leq j \leq L$ and $1 \leq u_1, v_1, u_2, v_2 \leq \bar{d}$.

We demonstrate that at least one of the coefficients of these independent functions does not go to 0 as $n \to \infty$. Assume by contrary that all these coefficients of these linear independent functions go to 0. From equation (37), we obtain that $\widetilde{M}_{n,j,\alpha_1,\alpha_2}/\mathcal{D}_{3n}$, $\widetilde{M}_{n,j,\alpha_1,\beta_1,\alpha_2,\beta_2}/\mathcal{D}_{3n}$, and $\widetilde{N}_{n,j}/\mathcal{D}_{3n}$ go to 0 for all the coefficients $\alpha_1, \beta_1, \alpha_2, \beta_2 \in \mathbb{N}^{\bar{d}\times\bar{d}}$ satisfying that $1 \leq |\alpha_1| + |\beta_1| + |\alpha_2| + |\beta_2| \leq 2$.

Since $\widetilde{N}_{n,j}/\mathcal{D}_{3n} \to 0$, we find that for any $j \in [L]$

$$\frac{|\sum_{i \in \mathcal{V}_j} \exp(c_{n,i}) - \exp(c_j^*)|}{\mathcal{D}_{3n}} = \frac{|\widetilde{N}_{n,j}|}{\mathcal{D}_{3n}} \to 0.$$

Taking the summation of these limits leads to

$$\frac{\sum_{j=1}^{L} |\sum_{i \in \mathcal{V}_j} \exp(c_{n,i}) - \exp(c_j^*)|}{\mathcal{D}_{3n}} \to 0. \tag{38}$$

Now, for any index $j \in [L]$ such that $|\mathcal{V}_j| = 1$, the limits $\widetilde{M}_{n,j,e_u,0_d}/\mathcal{D}_{3n} \to 0$ lead to $\frac{\sum_{i \in \mathcal{V}_j} \exp(c_{n,i}) \|\Delta \boldsymbol{A}_{n,ij}\|_1}{\mathcal{D}_{3n}} \to 0$ as $n \to \infty$. Due to the equivalence between the $\ell_1$-norm and the $\ell_2$-norm, this result directly implies that

$$\frac{\sum_{j:|\mathcal{V}_j|=1} \sum_{i \in \mathcal{V}_j} \exp(c_{n,i}) \|\Delta \boldsymbol{A}_{n,ij}\|}{\mathcal{D}_{3n}} \to 0.$$

Similarly, since $\widetilde{M}_{n,j,0_d,e_u}/\mathcal{D}_{3n} \to 0$, we also get that $\frac{\sum_{j:|\mathcal{V}_j|=1} \sum_{i \in \mathcal{V}_j} \exp(c_{n,i}) \|\Delta \boldsymbol{B}_{n,ij}\|}{\mathcal{D}_{3n}} \to 0$. Thus, we obtain that

$$\frac{\sum_{j:|\mathcal{V}_j|=1} \sum_{i \in \mathcal{V}_j} \exp(c_{n,i})(\|\Delta \boldsymbol{A}_{n,ij}\| + \|\Delta \boldsymbol{B}_{n,ij}\|)}{\mathcal{D}_{3n}} \to 0 \tag{39}$$

Moving to indices $j \in [L]$ such that their corresponding Voronoi cells satisfy that $|\mathcal{V}_j| > 1$. The limits $\widetilde{M}_{n,j,2e_u,0_d}/\mathcal{D}_{3n} \to 0$ and $\widetilde{M}_{n,j,0_d,2e_u}/\mathcal{D}_{3n} \to 0$ induces that

$$\frac{\sum_{j:|\mathcal{V}_j|>1} \sum_{i \in \mathcal{V}_j} \exp(c_{n,i})(\|\Delta \boldsymbol{A}_{n,ij}\|^2 + \|\Delta \boldsymbol{B}_{n,ij}\|^2)}{\mathcal{D}_{3n}} \to 0 \tag{40}$$

By putting the results in equations (38), (39), and (40) together, we arrive at $1 = \frac{\mathcal{D}_{3n}}{\mathcal{D}_{3n}} \to 0$ as $n \to \infty$, which is a contradiction. As a consequence, at least one of the coefficients of the linear independent functions in $Q_n(\mathbb{X})/\mathcal{D}_{3n}$ does not go to 0 as $n \to \infty$.

**Step 3 - Application of the Fatou's lemma.** We denote $\widetilde{m}_n$ as the maximum of the absolute values of the coefficients of the linear independent functions in $Q_n(\mathbb{X})/\mathcal{D}_{3n}$. As at least one of these coefficients does not go to 0, it indicates that $1/\widetilde{m}_n \not\to \infty$ as $n \to \infty$. Since $\|f_{\widetilde{G}_n} - f_{\widetilde{G}_*}\|_{L^2(\mu)}/\mathcal{D}_{3n} \to 0$ as $n \to \infty$, we obtain $\|f_{\widetilde{G}_n} - f_{\widetilde{G}_*}\|_{L^2(\mu)}/(\widetilde{m}_n \mathcal{D}_{3n}) \to 0$. An application of the Fatou's lemma leads to:

$$0 = \lim_{n \to \infty} \frac{\|f_{\widetilde{G}_n} - f_{\widetilde{G}_*}\|_{L^2(\mu)}}{\widetilde{m}_n \mathcal{D}_{3n}} \geq \int \liminf_{n \to \infty} \frac{\left| f_{\widetilde{G}_n}(\mathbb{X}) - f_{\widetilde{G}_*}(\mathbb{X}) \right|}{\widetilde{m}_n \mathcal{D}_{3n}} d\mu(\mathbb{X}) \geq 0.$$

That inequality demonstrates that $\liminf_{n \to \infty} \frac{\left| f_{\widetilde{G}_n}(\mathbb{X}) - f_{\widetilde{G}_*}(\mathbb{X}) \right|}{\widetilde{m}_n \mathcal{D}_{3n}} = 0$ for almost surely $\mathbb{X}$. As $n \to \infty$, we denote

$$\frac{\widetilde{N}_{n,j}}{\widetilde{m}_n \mathcal{D}_{3n}} \to \tilde{\lambda}_{0,j}, \quad \frac{L_{n,\tau,j}}{\widetilde{m}_n \mathcal{D}_{3n}} \to \tilde{\lambda}_{\tau,j},$$

for any indices $j \in [L]$ and $\tau \in [7]$. Here, at least one element of the set $\{\tilde{\lambda}_{0,j}, \tilde{\lambda}_{\tau,j} : j \in [L], \tau \in [7]\}$ is different from 0.

Given the above notations, the limit $\liminf_{n \to \infty} \dfrac{\left| f_{\widetilde{G}_n}(\mathbb{X}) - f_{\widetilde{G}_*}(\mathbb{X}) \right|}{m_n \mathcal{D}_{3n}} = 0$ implies that

$$
= \sum_{j:|\mathcal{V}_j|=1} \exp(\mathbb{X}^\top (\boldsymbol{M}_Q^0 + \sigma_2(\boldsymbol{B}_j^*)\sigma_1(\boldsymbol{A}_j^*))\mathbb{X}) \Big[ (\tilde{\lambda}_{1,j}^\top \mathbb{X}\mathbb{X}^\top \sigma_2(\boldsymbol{B}_j^*) + \tilde{\lambda}_{2,j}^\top \mathbb{X}\sigma_1(\boldsymbol{A}_j^*)\mathbb{X})(\boldsymbol{M}_V^0 + \sigma_2(\boldsymbol{B}_j^*)\sigma_1(\boldsymbol{A}_j^*))\mathbb{X}
$$

$$
+ \tilde{\lambda}_{1,j}^\top \mathbb{X}\sigma_2(\boldsymbol{B}_j^*) + \sigma_1(\boldsymbol{A}_j^*)\mathbb{X}\tilde{\lambda}_{2,j} \Big]
$$

$$
+ \sum_{j:|\mathcal{V}_j|>1} \exp(\mathbb{X}^\top (\boldsymbol{M}_Q^0 + \sigma_2(\boldsymbol{B}_j^*)\sigma_1(\boldsymbol{A}_j^*))\mathbb{X}) \Big[ (\tilde{\lambda}_{1,j}^\top \mathbb{X}\mathbb{X}^\top \sigma_2(\boldsymbol{B}_j^*) + \tilde{\lambda}_{2,j}^\top \mathbb{X}\sigma_1(\boldsymbol{A}_j^*)\mathbb{X} + \mathbb{X}^\top \tilde{\lambda}_{3,j} \mathbb{X}(\mathbb{X}^\top \sigma_2(\boldsymbol{B}_j^*))^2
$$

$$
+ \tilde{\lambda}_{4,j}^\top \mathbb{X}\mathbb{X}^\top \sigma_2(\boldsymbol{B}_j^*) + \mathbb{X}^\top \tilde{\lambda}_{5,j} \mathbb{X}(\sigma_1(\boldsymbol{A}_j^*)\mathbb{X})^2 + \tilde{\lambda}_{6,j}^\top \mathbb{X}\sigma_1(\boldsymbol{A}_j^*)\mathbb{X} + \mathbb{X}^\top \tilde{\lambda}_{7,j} \mathbb{X} + \mathbb{X}^\top \tilde{\lambda}_{7,j} \mathbb{X}\mathbb{X}^\top \sigma_2(\boldsymbol{B}_j^*)\sigma_1(\boldsymbol{A}_j^*)\mathbb{X})
$$

$$
\times (\boldsymbol{M}_V^0 + \sigma_2(\boldsymbol{B}_j^*)\sigma_1(\boldsymbol{A}_j^*))\mathbb{X} + \tilde{\lambda}_{1,j}^\top \mathbb{X}\sigma_2(\boldsymbol{B}_j^*) + \sigma_1(\boldsymbol{A}_j^*)\mathbb{X}\tilde{\lambda}_{2,j} + \tilde{\lambda}_{4,j}^\top \mathbb{X}\sigma_2(\boldsymbol{B}_j^*) + \sigma_1(\boldsymbol{A}_j^*)\mathbb{X}\tilde{\lambda}_{6,j} + \tilde{\lambda}_{7,j}^\top \mathbb{X} \Big]
$$

$$
- \sum_{j:|\mathcal{V}_j|=1} \exp(\mathbb{X}^\top (\boldsymbol{M}_Q^0 + \sigma_2(\boldsymbol{B}_j^*)\sigma_1(\boldsymbol{A}_j^*))\mathbb{X}) \Big[ \tilde{\lambda}_{1,j}^\top \mathbb{X}\mathbb{X}^\top \sigma_2(\boldsymbol{B}_j^*) + \tilde{\lambda}_{2,j}^\top \mathbb{X}\sigma_1(\boldsymbol{A}_j^*)\mathbb{X} \Big] f_{\widetilde{G}_*}(\mathbb{X})
$$

$$
- \sum_{j:|\mathcal{V}_j|>1} \exp(\mathbb{X}^\top (\boldsymbol{M}_Q^0 + \sigma_2(\boldsymbol{B}_j^*)\sigma_1(\boldsymbol{A}_j^*))\mathbb{X}) \Big[ \tilde{\lambda}_{1,j}^\top \mathbb{X}\mathbb{X}^\top \sigma_2(\boldsymbol{B}_j^*) + \tilde{\lambda}_{2,j}^\top \mathbb{X}\sigma_1(\boldsymbol{A}_j^*)\mathbb{X} + \mathbb{X}^\top \tilde{\lambda}_{3,j} \mathbb{X}(\mathbb{X}^\top \sigma_2(\boldsymbol{B}_j^*))^2
$$

$$
+ \tilde{\lambda}_{4,j}^\top \mathbb{X}\mathbb{X}^\top \sigma_2(\boldsymbol{B}_j^*) + \mathbb{X}^\top \tilde{\lambda}_{5,j} \mathbb{X}(\sigma_1(\boldsymbol{A}_j^*)\mathbb{X})^2 + \tilde{\lambda}_{6,j}^\top \mathbb{X}\sigma_1(\boldsymbol{A}_j^*)\mathbb{X} + \mathbb{X}^\top \tilde{\lambda}_{7,j} \mathbb{X} + \mathbb{X}^\top \tilde{\lambda}_{7,j} \mathbb{X}\mathbb{X}^\top \sigma_2(\boldsymbol{B}_j^*)\sigma_1(\boldsymbol{A}_j^*)\mathbb{X} \Big] f_{\widetilde{G}_*}(\mathbb{X})
$$

$$
+ \sum_{j=1}^{L} \tilde{\lambda}_{0,j} \exp(\mathbb{X}^\top (\boldsymbol{M}_Q^0 + \sigma_2(\boldsymbol{B}_j^*)\sigma_1(\boldsymbol{A}_j^*))\mathbb{X}) \Big[ (\boldsymbol{M}_V^0 + \boldsymbol{B}_j^* \boldsymbol{A}_j^*)\mathbb{X} - f_{\widetilde{G}_*}(\mathbb{X}) \Big] = 0 \tag{41}
$$

for almost surely $\mathbb{X}$. However, that equation implies that all the coefficients $\{\tilde{\lambda}_{0,j}, \tilde{\lambda}_{\tau,j} : j \in [L], \tau \in [7]\}$ are 0. It is a contradiction. As a consequence, we obtain that

$$
\lim_{\varepsilon \to 0} \inf_{\widetilde{G} \in \mathcal{G}_{L'}(\Theta):\mathcal{D}_3(\widetilde{G},\widetilde{G}_*) \leq \varepsilon} \| f_{\widetilde{G}} - f_{\widetilde{G}_*} \|_{L^2(\mu)} / \mathcal{D}_3(\widetilde{G}, \widetilde{G}_*) > 0.
$$

### A.3.2. GLOBAL PART

The result of the local part implies that we can find a positive constant $\varepsilon'$ such that

$$
\inf_{\widetilde{G} \in \mathcal{G}_{L'}(\Theta):\mathcal{D}_3(\widetilde{G},\widetilde{G}_*) \leq \varepsilon'} \| f_{\widetilde{G}} - f_{\widetilde{G}_*} \|_{L^2(\mu)} / \mathcal{D}_3(\widetilde{G}, \widetilde{G}_*) > 0.
$$

Therefore to obtain the conclusion of the theorem, we only need to prove that

$$
\inf_{\widetilde{G} \in \mathcal{G}_{L'}(\Theta):\mathcal{D}_3(\widetilde{G},\widetilde{G}_*) > \varepsilon'} \| f_{\widetilde{G}} - f_{\widetilde{G}_*} \|_{L^2(\mu)} / \mathcal{D}_3(\widetilde{G}, \widetilde{G}_*) > 0.
$$

We assume by contradiction that the above claim does not hold. It indicates that there exists a sequence of measures $\widetilde{G}'_n := \sum_{j=1}^{L} \exp(c_{n,j})\delta_{(\boldsymbol{B}_{n,j},\boldsymbol{A}_{n,j})}$ in $\widetilde{\mathcal{G}}_{L'}(\Theta)$ such that

$$
\begin{cases}
\mathcal{D}_3(\widetilde{G}'_n, \widetilde{G}_*) > \varepsilon' \\
\| f_{\widetilde{G}'_n} - f_{\widetilde{G}_*} \|_{L^2(\mu)} / \mathcal{D}_3(\widetilde{G}'_n, \widetilde{G}_*) \to 0
\end{cases}
$$

as $n \to \infty$, which implies that $\| f_{\widetilde{G}'_n} - f_{\widetilde{G}_*} \|_{L^2(\mu)} \to 0$ as $n \to \infty$.

Given that $\Theta$ is a compact set, there exists a mixing measure $\widetilde{G}'$ in $\widetilde{\mathcal{G}}_{L'}(\Theta)$ such that one of the $\widetilde{G}'_n$'s subsequences converges to $\widetilde{G}'$. Since $\mathcal{D}_3(\widetilde{G}'_n, \widetilde{G}_*) > \varepsilon'$, we obtain that $\mathcal{D}_3(\widetilde{G}', \widetilde{G}_*) > \varepsilon'$. An application of the Fatou's lemma leads to

$$
0 = \lim_{n \to \infty} \| f_{\widetilde{G}'_n} - f_{\widetilde{G}_*} \|_{L^2(\mu)} \geq \int \liminf_{n \to \infty} \left\| f_{\widetilde{G}'_n}(\mathbb{X}) - f_{\widetilde{G}_*}(\mathbb{X}) \right\|^2 d\mu(\mathbb{X}).
$$

The above inequality indicates that $f_{\widetilde{G}'} = f_{\widetilde{G}_*}$ for almost surely $\mathbb{X}$. From the identifiability property, we deduce that $\widetilde{G}' \equiv \widetilde{G}_*$. It follows that $\mathcal{D}_3(\widetilde{G}', \widetilde{G}_*) = 0$, contradicting the fact that $\mathcal{D}_3(\widetilde{G}', \widetilde{G}_*) > \varepsilon' > 0$. Hence, the proof is completed.

**Proof for the identifiability property.** We will prove that if $f_{\widetilde{G}}(\mathbb{X}) = f_{\widetilde{G}_*}(\mathbb{X})$ for almost surely $\mathbb{X}$, then $G \equiv \widetilde{G}_*$. To ease the presentation, for any mixing measure $\widetilde{G} = \sum_{j=1}^{\tilde{L}} \exp(c_j) \delta_{(\boldsymbol{B}_j, \boldsymbol{A}_j)} \in \mathcal{G}_{L'}(\Theta)$, we denote

$$
\mathrm{softmax}_G(u) = \frac{\exp(u)}{\sum_{j=1}^{\tilde{L}} \exp(\mathbb{X}^\top (\boldsymbol{M}_Q^0 + \sigma_2(\boldsymbol{B}_j)\sigma_1(\boldsymbol{A}_j))\mathbb{X} + c_j)},
$$

where $u \in \{\mathbb{X}^\top (\boldsymbol{M}_Q^0 + \sigma_2(\boldsymbol{B}_j)\sigma_1(\boldsymbol{A}_j))\mathbb{X} + c_j : j \in [\tilde{L}]\}$. The equation $f_{\widetilde{G}}(\mathbb{X}) = f_{\widetilde{G}_*}(\mathbb{X})$ indicates that

$$
\sum_{j=1}^{\tilde{L}} \mathrm{softmax}(\mathbb{X}^\top (\boldsymbol{M}_Q^0 + \sigma_2(\boldsymbol{B}_j)\sigma_1(\boldsymbol{A}_j))\mathbb{X} + c_j)(\boldsymbol{M}_V^0 + \sigma_2(\boldsymbol{B}_j)\sigma_1(\boldsymbol{A}_j))\mathbb{X}
$$

$$
= \sum_{j=1}^{L} \mathrm{softmax}(\mathbb{X}^\top (\boldsymbol{M}_Q^0 + \sigma_2(\boldsymbol{B}_j^*)\sigma_1(\boldsymbol{A}_j^*))\mathbb{X} + c_j^*)(\boldsymbol{M}_V^0 + \sigma_2(\boldsymbol{B}_j^*)\sigma_1(\boldsymbol{A}_j^*))\mathbb{X} \quad (42)
$$

That equation implies that $\tilde{L} = L$. As a consequence, we find that

$$
\{\mathrm{softmax}(\mathbb{X}^\top (\boldsymbol{M}_Q^0 + \sigma_2(\boldsymbol{B}_j)\sigma_1(\boldsymbol{A}_j))\mathbb{X} + c_j) : j \in [\tilde{L}]\} = \{\mathrm{softmax}(\mathbb{X}^\top (\boldsymbol{M}_Q^0 + \sigma_2(\boldsymbol{B}_j^*)\sigma_1(\boldsymbol{A}_j^*))\mathbb{X} + c_j^*) : j \in [L]\}
$$

for almost surely $\mathbb{X}$. By relabelling the indices, we can assume without loss of generality that for any $j \in [L]$

$$
\mathrm{softmax}(\mathbb{X}^\top (\boldsymbol{M}_Q^0 + \sigma_2(\boldsymbol{B}_j^*)\sigma_1(\boldsymbol{A}_j^*))\mathbb{X} + c_j^*) = \mathrm{softmax}(\mathbb{X}^\top (\boldsymbol{M}_Q^0 + \sigma_2(\boldsymbol{B}_j)\sigma_1(\boldsymbol{A}_j))\mathbb{X} + c_j),
$$

for almost surely $\mathbb{X}$. Given the invariance to translation of the softmax function, the equation (42) leads to

$$
\sum_{j=1}^{\tilde{L}} \exp(c_j) \exp(\mathbb{X}^\top (\boldsymbol{M}_Q^0 + \sigma_2(\boldsymbol{B}_j)\sigma_1(\boldsymbol{A}_j))\mathbb{X})(\boldsymbol{M}_V^0 + \sigma_2(\boldsymbol{B}_j)\sigma_1(\boldsymbol{A}_j))\mathbb{X}
$$

$$
= \sum_{j=1}^{L} \exp(c_j^*) \exp(\mathbb{X}^\top (\boldsymbol{M}_Q^0 + \sigma_2(\boldsymbol{B}_j^*)\sigma_1(\boldsymbol{A}_j^*))\mathbb{X})(\boldsymbol{M}_V^0 + \sigma_2(\boldsymbol{B}_j^*)\sigma_1(\boldsymbol{A}_j^*))\mathbb{X}, \quad (43)
$$

for almost surely $\mathbb{X}$.

Now, the index set $[L]$ can be partitioned into $\tilde{m}$ subsets $\tilde{K}_1, \tilde{K}_2, \ldots, \tilde{K}_{\tilde{m}}$ where $\tilde{m} \leq L$, such that $\exp(c_j) = \exp(c_{j'}^*)$ for any indices $j, j' \in \tilde{K}_i$ and $i \in [\tilde{m}]$. Thus, equation (43) can be rewritten as follows:

$$
\sum_{i=1}^{\tilde{m}} \sum_{j \in \tilde{K}_i} \exp(c_j) \exp(\mathbb{X}^\top (\boldsymbol{M}_Q^0 + \sigma_2(\boldsymbol{B}_j)\sigma_1(\boldsymbol{A}_j))\mathbb{X})(\boldsymbol{M}_V^0 + \sigma_2(\boldsymbol{B}_j)\sigma_1(\boldsymbol{A}_j))\mathbb{X}
$$

$$
= \sum_{i=1}^{\tilde{m}} \sum_{j \in \tilde{K}_i} \exp(c_j^*) \exp(\mathbb{X}^\top (\boldsymbol{M}_Q^0 + \sigma_2(\boldsymbol{B}_j^*)\sigma_1(\boldsymbol{A}_j^*))\mathbb{X})(\boldsymbol{M}_V^0 + \sigma_2(\boldsymbol{B}_j^*)\sigma_1(\boldsymbol{A}_j^*))\mathbb{X},
$$

for almost surely $\mathbb{X}$. The above equation implies that

$$
\{(\boldsymbol{M}_V^0 + \sigma_2(\boldsymbol{B}_j)\sigma_1(\boldsymbol{A}_j))\mathbb{X} : j \in \tilde{K}_i\} = \{(\boldsymbol{M}_V^0 + \sigma_2(\boldsymbol{B}_j^*)\sigma_1(\boldsymbol{A}_j^*))\mathbb{X} : j \in \tilde{K}_i\},
$$

for any $i \in [\tilde{m}]$ and for almost surely $\mathbb{X}$. Since the activation functions $\sigma_1$ and $\sigma_2$ are algebraically independent, the above result indicates that

$$
\sum_{i=1}^{\tilde{m}} \sum_{j \in \tilde{K}_i} \exp(c_j) \delta_{(\boldsymbol{B}_j, \boldsymbol{A}_j)} = \sum_{i=1}^{\tilde{m}} \sum_{j \in \tilde{K}_i} \exp(c_j^*) \delta_{(\boldsymbol{B}_j^*, \boldsymbol{A}_j^*)}.
$$

As a consequence, $G \equiv G_*$ and the proof is completed.

## A.4. Proof of Proposition A.1

Recall from the setting that $(\boldsymbol{X}_1, Y_1), (\boldsymbol{X}_2, Y_2), \ldots, (\boldsymbol{X}_n, Y_n) \in \mathbb{R}^{\bar{d}} \times \mathbb{R}^{\bar{d}}$ are i.i.d. samples from the following regression model:

$$Y_i = f_{\bar{G}_*}(\boldsymbol{X}_i) + \varepsilon_i, \quad i = 1, 2, \ldots, n,$$

where the Gaussian noises $\varepsilon_1, \ldots, \varepsilon_n$ are i.i.d. and satisfy that $\mathbb{E}[\varepsilon_i | \boldsymbol{X}_i] = 0$ and $\mathrm{Var}(\varepsilon_i | \boldsymbol{X}_i) = \sigma^2 I_{\bar{d}}$ for all $i \in [n]$. Furthermore, $f_{\bar{G}_*}(.)$ admits the following form:

$$f_{\bar{G}_*}(\boldsymbol{X}) := \sum_{j=1}^{L} \frac{\exp(\mathbb{X}^{\top}(\boldsymbol{M}_Q^0 + \boldsymbol{W}_{2,j}^* \boldsymbol{B}_j^* \boldsymbol{W}_{1,j}^* \boldsymbol{A}_j^*) \boldsymbol{M}_K^0 \mathbb{X} + c_j^*)}{\bar{D}_f(\mathbb{X})} \cdot (\boldsymbol{M}_V^0 + \boldsymbol{W}_{2,j}^* \boldsymbol{B}_j^* \boldsymbol{W}_{1,j}^* \boldsymbol{A}_j^*) \mathbb{X},$$

where we denote $\bar{D}_f(\mathbb{X}) = \sum_{k=1}^{L} \exp(\mathbb{X}^{\top}(\boldsymbol{M}_Q^0 + \boldsymbol{W}_{2,k}^* \boldsymbol{B}_k^* \boldsymbol{W}_{1,k}^* \boldsymbol{A}_k^*) \boldsymbol{M}_K^0 \mathbb{X} + c_k^*)$. Finally, the least-square estimator $\bar{G}_n$ takes the following form:

$$\bar{G}_n := \arg \min_{G \in \bar{\mathcal{G}}_{L'}(\Theta)} \sum_{i=1}^{n} \|Y_i - f_G(\boldsymbol{X}_i)\|^2,$$

From the Gaussianity assumption of $\varepsilon_i | \boldsymbol{X}_i$ for all $i \in [n]$, we have $Y_i | \boldsymbol{X}_i \sim \mathcal{N}(f_{\bar{G}_*}(\boldsymbol{X}_i), \sigma^2 I_{\bar{d}})$ for all $i \in [n]$. Therefore, the least square estimator $\bar{G}_n$ is indeed a maximum likelihood estimator with respect to the data $Y_1 | \boldsymbol{X}_1, \ldots, Y_n | \boldsymbol{X}_n$, which takes the following form:

$$\bar{G}_n \in \arg \max_{G \in \bar{\mathcal{G}}_{L'}(\Theta)} \frac{1}{n} \sum_{i=1}^{n} \log(p(Y_i | f_G(\boldsymbol{X}_i), \sigma^2 I_{\bar{d}})).$$

Here, $p(Y_i | f_G(\boldsymbol{X}_i), \sigma^2 I_{\bar{d}})$ stands for multivariate Gaussian distribution with mean $f_G(\boldsymbol{X})$ and covariance matrix $\sigma^2 I_{\bar{d}}$. An application of Theorem 7.4 from (van de Geer, 2000) leads to

$$h(p(Y | f_{\bar{G}_n}(\boldsymbol{X}), \sigma^2 I_{\bar{d}}), p(Y | f_{\bar{G}_*}(\boldsymbol{X}), \sigma^2 I_{\bar{d}})) = \mathcal{O}_P(\sqrt{\log(n)/n}),$$

where $h$ stands for the Hellinger distance. As the Hellinger distance between two multivariate Gaussian distributions has closed-form expression, direct calculation yields that

$$h^2(p(Y | f_{\bar{G}_n}(\boldsymbol{X}), \sigma^2 I_{\bar{d}}), p(Y | f_{\bar{G}_*}(\boldsymbol{X}), \sigma^2 I_{\bar{d}})) = 1 - \exp\left\{ -\frac{1}{8\sigma^2} \|f_{\bar{G}_n}(\boldsymbol{X}) - f_{\bar{G}_*}(\boldsymbol{X})\|^2 \right\}.$$

Therefore, for sufficiently large $n$, for some universal constant $C$ the above inequality leads to

$$\|f_{\bar{G}_n}(\boldsymbol{X}) - f_{\bar{G}_*}(\boldsymbol{X})\|^2 \le 8\sigma^2 \log\left(\frac{1}{1 - C\log(n)/n}\right)$$

$$= 8\sigma^2 \log\left(1 + \frac{C\log(n)/n}{1 - C\log(n)/n}\right)$$

$$\le 8\sigma^2 \cdot \frac{C\log(n)/n}{1 - C\log(n)/n}$$

$$\le 16\sigma^2 C\log(n)/n.$$

That inequality is equivalent to

$$\|f_{\bar{G}_n}(\boldsymbol{X}) - f_{\bar{G}_*}(\boldsymbol{X})\| = \mathcal{O}_P(\sqrt{\log(n)/n}).$$

As a consequence, we find that

$$\|f_{\bar{G}_n} - f_{\bar{G}_*}\|_{L^2(\mu)} = \mathcal{O}_P(\sqrt{\log(n)/n}).$$

The proof of the proposition is completed.

# B. Experimental Details

## B.1. Hyperparameters

For the vision tasks, we use grid search to tune the learning rate in the range of $\{0.001, 0.005, 0.01, 0.05, 0.1\}$, and the weight decay in the range of $\{0.0001, 0.0005, 0.001, 0.01, 0.1\}$. Other hyperparameters are reported in the tables below:

*Table 7.* Hyperparameter configurations of RepLoRA for `ViT-B/16` on the vision tasks.

| Hyperparameters (RepLoRA) | Classification | Video-Action Recognition |
|---|---|---|
| Rank $r$ | | 8 |
| $\alpha$ | | 8 |
| Dropout | | 0 |
| Base Optimizer | | AdamW |
| Lr Scheduler | | Cosine |
| Batch size | 64 | 512 |
| Warmup steps | | 100 |
| Epochs | 100 | 90 |

*Table 8.* Hyperparameter configurations of RepLoRA for `LLaMA-7B/13B` on the commonsense reasoning tasks.

| Hyperparameters (RepLoRA) | LLaMA-7B | | LLaMA-13B | |
|---|---|---|---|---|
| Rank $r$ | 16 | 32 | 16 | 32 |
| $\alpha$ | 32 | 64 | 32 | 64 |
| Dropout | | | 0.05 | |
| Base Optimizer | | | AdamW | |
| LR | $2.00E-04$ | $1.00E-04$ | $2.00E-04$ | $1.00E-04$ |
| Lr Scheduler | | | Linear | |
| Batch size | | | 32 | |
| Warmup steps | | | 100 | |
| Epochs | | | 3 | |

*Table 9.* Hyperparameter configurations of RepLoRA for `VL-BART` on the Image/Video-Text Understanding tasks.

| Hyperparameters (RepLoRA) | Image-Text | Video-Text |
|---|---|---|
| Rank $r$ | | 128 |
| $\alpha$ | | 128 |
| Dropout | | 0 |
| Base Optimizer | | AdamW |
| LR | $1.00E-03$ | $3.00E-04$ |
| Lr Scheduler | | Linear |
| Batch size | 300 | 40 |
| Warmup ratio | | 0.1 |
| Epochs | 20 | 7 |

# C. Additional Experiments

## C.1. Sample Efficiency on the FGVC Datasets

## C.2. Linear vs Non-linear Reparameterization

Recall that in our practical method, the low-rank matrices are given by:

*Table 10.* Detail statistic of RepLoRA and LoRA sample efficiencies on five `FGVC` datasets.

|         | $f$  | `CUB_200_2011` | `NABirds` | `OxfordFlower` | `StanfordDogs` | `StanfordCars` | **AVG** |
|---------|------|----------------|-----------|----------------|----------------|----------------|---------|
|         | 0.01 | 10.1           | 2.2       | 15.1           | 12.1           | 13.3           | 10.56   |
|         | 0.1  | 70.6           | 50.3      | 71.6           | 65.3           | 60.2           | 63.6    |
| LoRA    | 0.3  | 75.1           | 70.8      | 85.3           | 80.2           | 73.3           | 76.94   |
|         | 0.5  | 80.1           | 76.9      | 96.2           | 85             | 75.5           | 82.74   |
|         | 1    | 84.6           | 78.2      | 98.9           | 85.1           | 77.1           | 84.78   |
|         | 0.01 | 50.2           | 40.1      | 55.3           | 51.2           | 59.8           | 51.32   |
|         | 0.1  | 80.2           | 79.6      | 85.9           | 80.8           | 79.1           | 81.12   |
| RepLoRA | 0.3  | 85.3           | 85.9      | 93.1           | 79.1           | 81.6           | 85.0    |
|         | 0.5  | 87.9           | 86        | 98.9           | 86.3           | 84.9           | 88.8    |
|         | 1    | 89.1           | 86.1      | 99.3           | 91.2           | 87.6           | 90.66   |

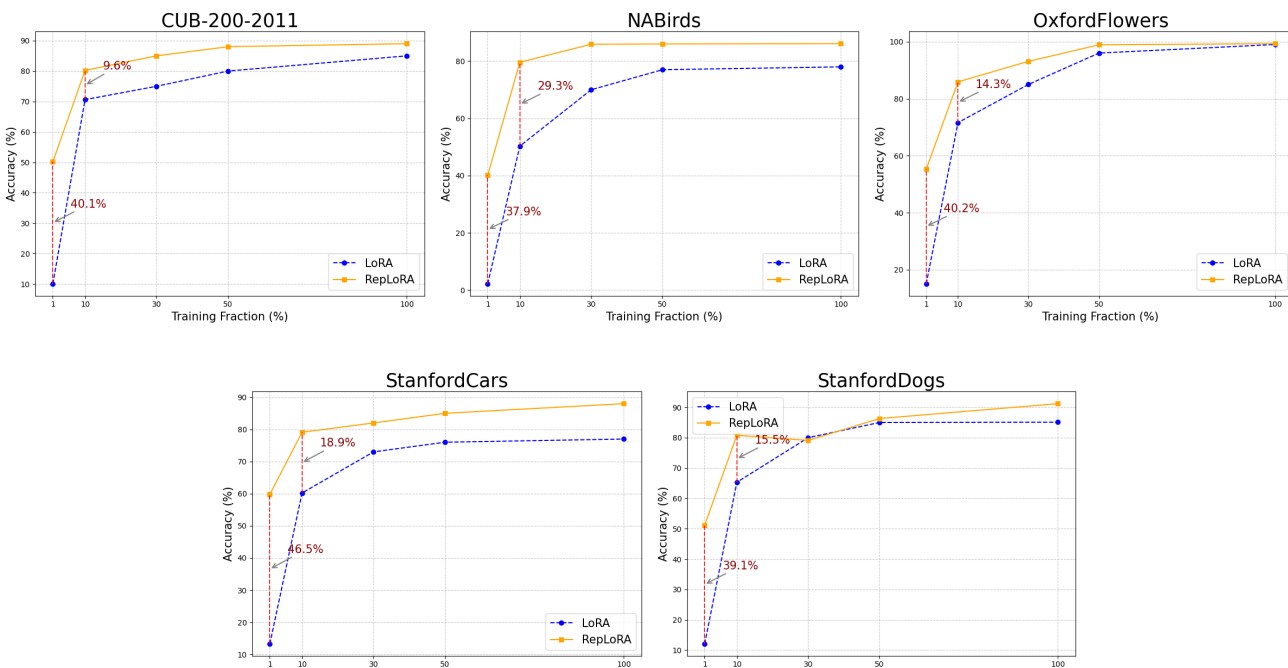

*Figure 4.* Visualization of sample efficiency of LoRA and RepLoRA on five `FGVC` Datasets.

$$A_Q = \sigma_1^{A}(A), B_Q = \sigma_1^{B}(B)$$
$$A_V = \sigma_2^{A}(A), B_V = \sigma_2^{B}(B)$$

For the linear reparameterization, $\sigma_1^{A}, \sigma_2^{A}, \sigma_1^{B}, \sigma_2^{B}$ were implemented with linear layers without activation. For the nonlinear reparameterization setting, these functions were implemented with a two-layer neural network with a hidden dimension of $64$ and sigmoid activation functions on all settings.

**Detail results.** The tables below report all results for linear vs. non-linear reparameterization

*Table 11.* Image classification accuracy of Linear vs. Non-linear Reparameterization on `VTAB-1K`.

| Method | CIFAR100 | Caltech101 | DTD | Flower102 | Pets | SVHN | Sun397 | Camelyon | EuroSAT | Resisc45 | Retinopathy | Clevr-Count | Clevr-Dist | DMLab | KITTI | dSpr-Loc | dSpr-Ori | sNORB-Azim | sNORB-Ele | AVG |
|---|---|---|---|---|---|---|---|---|---|---|---|---|---|---|---|---|---|---|---|---|
| LoRA | 67.1 | 91.4 | 69.4 | 98.2 | 90.4 | 85.3 | 54 | 84.9 | 95.3 | 84.4 | 73.6 | 82.9 | 69.2 | 49.8 | 78.5 | 75.7 | 47.1 | 31 | 44 | 72.2 |
| RepLoRA (Linear) | 70.1 | 93.1 | 71.7 | 98.9 | 93.3 | 89 | 56 | **90.4** | 95.8 | 86.3 | 75.6 | 83.2 | 70.6 | 54.6 | 76.7 | 80.6 | 48 | 31.3 | 39.9 | 73.9 |
| RepLoRA (Non-linear) | **73.2** | **94.1** | **73.3** | **99.3** | **94.4** | **89.1** | **58.9** | 89.2 | **97.5** | **87.9** | **77.8** | **85.1** | **72.6** | **55.7** | **81.2** | **81.7** | **49.2** | **35.7** | **47.3** | **75.9** |

*Table 12.* Image classification accuracy of Linear vs. Non-linear Reparameterization on `FGVC` datasets.

| Method | CUB_200_2011 | NABirds | OxfordFlower | StanfordDogs | StanfordCars | AVG |
|---|---|---|---|---|---|---|
| LoRA | 84.6 | 78.2 | 98.9 | 85.1 | 77.1 | 84.7 |
| RepLoRA (Linear) | 88.6 | 85.2 | 98.1 | 89.9 | 83.3 | 89.0 |
| RepLoRA (Non-linear) | **89.1** | **86.1** | **99.3** | **91.2** | **87.6** | **90.7** |

*Table 13.* Video action recognition performance of Linear vs. Non-linear Reparameterization on `SSv2` and `HMDB51` datasets.

| Method | Model | Pretraining | #Params (M) | SSv2 Acc@1 | SSv2 PPT | HMDB51 Acc@1 | HMDB51 PPT |
|---|---|---|---|---|---|---|---|
| LoRA | Video Swin-B | Kinetics400 | 0.75 | 38.34 | 0.37 | 62.12 | 0.60 |
| RepLoRA (Linear) | Video Swin-B | Kinetics400 | 0.91 | 41.89 | 0.40 | 66.01 | 0.63 |
| RepLoRA (Non-linear) | Video Swin-B | Kinetics400 | 1.45 | **43.12** | **0.41** | **68.23** | **0.64** |

*Table 14.* Performance of performance of Linear vs. Non-linear Reparameterization on the Commonsense Reasoning task.

| | PEFT Method | BoolQ | PIQA | SIQA | HellaSwag | WinoGrande | ARC-e | ARC-c | OBQA | AVG |
|---|---|---|---|---|---|---|---|---|---|---|
| LLaMA-7B | LoRA | 67.2 | 79.4 | 76.6 | 78.3 | 78.4 | 77.1 | 61.5 | 74.2 | 74.0875 |
| | RepLoRA (Linear) | 67.1 | 81.7 | **79.3** | 77.9 | 79.6 | 78.4 | 64.1 | 77.4 | 75.6875 |
| | RepLoRA (Non-linear) | **71.8** | **84.1** | 78.9 | **85.2** | **83.3** | **82.4** | **66.2** | **81.2** | **79.1375** |
| LLaMA-13B | LoRA | 71.7 | 82.4 | 79.6 | 90.4 | 83.6 | 83.1 | 68.5 | 82.1 | 80.175 |
| | RepLoRA (Linear) | 72.6 | 82.2 | 82.3 | 90.4 | 84.1 | 82.9 | 67.9 | 83.9 | 80.7875 |
| | RepLoRA (Non-linear) | **73.1** | **85.2** | **84.7** | **91.1** | **85.9** | **84.7** | **73.4** | **85.6** | **82.9625** |

*Table 15.* Performance of Linear vs Non-linear reparameterization on the image-text understanding task on `VL-BART`.

| Method | VQA | GQA | NVLR | COCO Cap | Avg. |
|---|---|---|---|---|---|
| LoRA | 65.2 | 53.6 | 71.9 | 115.3 | 76.5 |
| RepLoRA (Linear) | 65.5 | 55 | 72.3 | 115.9 | 77.2 |
| RepLoRA (Non-linear) | **66.5** | **55.4** | **74.2** | **116.2** | **78.1** |

*Table 16.* Performance of Linear vs Non-linear reparameterization on the video-text understanding task on `VL-BART`.

| Method | TVQA | How2QA | TVC | YC2C | Avg. |
|---|---|---|---|---|---|
| LoRA | 75.5 | 72.9 | 44.6 | 140.9 | 83.5 |
| RepLoRA (Linear) | 76.3 | 73.4 | 44.9 | 143.2 | 84.5 |
| RepLoRA (Non-linear) | **77.8** | **75.1** | **46.6** | **151.6** | **87.8** |

