# OpenReview forum: "RepLoRA: Reparameterizing Low-rank Adaptation via the Perspective of Mixture of Experts"
_ICML.cc/2025/Conference — ICML 2025 poster_

### Official Review · Reviewer_ykAM · 2025-03-10

**Overall Recommendation:** 4

**Summary:**

This work studies a new variant of LoRA. First, the authors show that under certain settings, LoRA requires exponential sample complexity. Then, they introduce a simple reparameterization strategy, which builds a single generator for Q, V layers. The generator can be a single layer with or without activations. Using this reparameterization, the authors show that the sample complexity can be reduced to polynomial scale, which reveals the advantage of the new method. Experiments are conducted on multiple domains including LLMs, images/videos and multi-modal datasets. The proposed method consistently outperforms LoRA.

## Update after rebuttal

As the authors' address most concerns of all reviewers, I would keep the score.

**Claims And Evidence:**

Yes, the claims are supported by theoretical and empirical analysis.

**Essential References Not Discussed:**

No.

**Experimental Designs Or Analyses:**

The experimental design is sound, including datasets and pre-trained models of different domains.

**Methods And Evaluation Criteria:**

Yes.

**Other Comments Or Suggestions:**

In Line 349, PETL is not introduced.

**Other Strengths And Weaknesses:**

### Strengths

1. This work studies LoRA from the view of MoE, and theoretical analysis about the sample complexity. These results may be useful for future research in this field.

2. The proposed method is simple and theoretically sound. It may be also applied to other variants of LoRA.

3. Experiments are also conducted on diverse domains. The proposed method seems good.

### Weakness

1. While the method is simple and advantageous over LoRA, it would be better to compare with some more advanced variants of LoRA, especially some work that applies hypernetworks to generate adapters, which is very similar to this one. For example the following.

    https://openreview.net/forum?id=iP8ig954Uz

2. The parameters of RepLoRA are higher than LoRA. Did the authors try experiments with comparable parameter sizes?

**Questions For Authors:**

See above.

**Relation To Broader Scientific Literature:**

The key contribution is to improve LoRA by reparameterization. It may contribute to the field of parameter-efficient fine-tuning, including many other variants of LoRA.

**Theoretical Claims:**

I did not check the proof.

---

> ### Author Rebuttal · Authors · 2025-03-29
>
> We thank the reviewer for the feedback and would like to address the concerns as follows:
>
> + **Regarding the comparison with other variants of LoRA:** Following the reviewer’s suggestion, we conducted an additional experiment on the image classification task using the FGVC dataset to compare RepLoRA with VeRA [1] and DoRA [2]. The results are presented below:
>
> | Methods | CUB-200-2011 | NABirds | Oxford Flowers | Stanford Dogs | Stanford Cars | AVG | PPT |
> | -------- | -------- | -------- | -------- | -------- | -------- | -------- | -------- |
> | LoRA     | 84.6     | 78.2     |98.9     | 85.1     | 77.1     |84.8     | 0.82 |
> | DoRA     | 87.3     | 80.0     |99.1     | 87.6     | 81.9     |87.2     |0.88 |
> | VeRA     | 85.1     | 79.2     |97.4     | 87.3     | 76.3     |85.1     |0.88 |
> | RepLoRA     | **89.1**     | **86.1**     | **99.3**    | **91.2**     | **87.6**     | **90.7**     |**0.90** |
>
> Furthermore, we would like to highlight that RepLoRA has already been compared to other methods that utilize hypernetworks to generate adapters in the main paper. Specifically, in the main text, we included a comparison with prefix tuning [3], which leverages an MLP to generate the adapters, specifically the prepended prompts. Building on the reviewer’s helpful suggestion, we will also incorporate comparisons with VeRA and DoRA in the revised version to provide a more comprehensive evaluation.
>
> + **Regarding the comparisons with comparable parameter sizes:**  In all experiments, we reparameterized the low-rank adapters using MLPs with a hidden dimension of $h=64$, which we identified as the optimal setting through extensive tuning in terms of both PPT and accuracy. To address this concern, we also conducted experiments with a reduced hidden dimension of $h=8$, resulting in a comparable parameter size. The results, presented below, show a slight overall drop; however, RepLoRA still significantly outperforms vanilla LoRA, demonstrating its robustness and practical benefits under tighter parameter constraints:
>
> | Methods | CUB-200-2011 | NABirds | Oxford Flowers | Stanford Dogs | Stanford Cars | AVG | PPT |
> | -------- | -------- | -------- | -------- | -------- | -------- | -------- | -------- |
> | LoRA     | 84.6     | 78.2     |98.9     | 85.1     | 77.1     |84.8     | 0.82 |
> | RepLoRA $(h=8)$     | 87.5     | 85.8     |98.4     | **91.6**     | 83.0     |   89.2  |0.90 |
> | RepLoRA $(h=64)$    | **89.1**     | **86.1**     | **99.3**    | 91.2     | **87.6**     | **90.7**     |**0.90** |
>
> Following the reviewer's suggestion, we will include this finding in the appendix of the final manuscript.
>
> **References**
>
> [1] VeRA: Vector-based Random Matrix Adaptation. ICLR. 2024
>
> [2] DoRA: Weight-Decomposed Low-Rank Adaptation. ICML. 2024
>
> [3] Prefix-Tuning: Optimizing Continuous Prompts for Generation. ACL. 2021

---

> > ### Comment · Reviewer_ykAM · 2025-04-03
> >
> > Thanks for the authors' response. Most of my concerns are addressed. I would keep the score.

---

> > > ### Author Response · Authors · 2025-04-03
> > >
> > > Dear Reviewer ykAM,
> > >
> > > We are glad to hear that our response addresses your concerns, and we would like to thank you so much for keeping the positive rating of 4 (Accept), which we really appreciate. If you have any further concerns, please feel free to let us know. We will keep revising the manuscript based on the feedback from you and other reviewers.
> > >
> > > Thank you,
> > >
> > > The Authors

---

### Official Review · Reviewer_8a5d · 2025-03-13

**Overall Recommendation:** 4

**Summary:**

The authors proposed two reparametrizations of LoRA under which the convergence rate improves from $\mathcal{O}_P(\frac{1}{\log^{\tau}(n)})$ of vanilla LoRA to $\mathcal{O}_P(\sqrt{\frac{\log (n)}{n}})$. Empirical results demonstrate that both reparametrizations outperform vanilla LoRA on real datasets.

**Claims And Evidence:**

The claim that the proposed reparametrizations are more sample efficient are backed up by formal theorem statements and their proofs. However, there is a discrepancy between the theoretical results and the experimental setup: while Theorems 4.1-3 assume $A_Q$ and $A_V$ are tied, the experiments only assume that they result from the same low rank matrices. Therefore the theoretical results cannot fully explain the empirical successes.

**Essential References Not Discussed:**

Discussion of previous empirical observations on LoRA's performance (_e.g._, [Biderman et al., 2024](https://arxiv.org/abs/2405.09673)) can further improve the paper.

**Experimental Designs Or Analyses:**

A main message of this paper is that parameter estimation of the proposed reparameterizations is much more sample efficient. Besides the discrepancy between what is being proven and what is being implemented, another issue is that the authors did not explicitly evaluate whether the reparametrization hurts expressiveness in real world tasks when data is abundant. One way to help answer this question is to evaluate on pretraining tasks where there is much more training data.

**Methods And Evaluation Criteria:**

The experiment setup seems correct. However, I would love to see experiments that validate the theoretical results: namely experiments that tie $A_Q$ and $A_V$ together.

**Other Comments Or Suggestions:**

- Minor editing errors on L218, L175 (right column), and L716.
- The paper is both dense and packed; proof sketches in the main text can provide useful guidance for the reader.
- Choice of nonlinearity function used is missing in the manuscript.

**Other Strengths And Weaknesses:**

This paper explores the sample efficiency aspects of LoRA which is a novel and significant contribution.

**Questions For Authors:**

n/a

**Relation To Broader Scientific Literature:**

This work adds on the ongoing discussion on how to best train LoRA models ([Yen et al., 2025](https://openreview.net/forum?id=VpWki1v2P8), [Hayou et al., 2024](https://arxiv.org/abs/2402.12354)).

**Theoretical Claims:**

I checked proof for Theorem 4.1 which seems correct. However the implications in L187-196 (right column) should be formalized into a proposition and proven. I did not carefully check proofs for Theorems 4.2-3 which also appear in the appendix.

---

> ### Author Rebuttal · Authors · 2025-03-29
>
> We thank the reviewer for the valuable feedback and would like to address the concerns raised as follows:
>
> + **Regarding the discrepancy between the theoretical results and the experiment setup**: Thanks for this feedback. We would like to clarify that the assumption in Section 4.2 that $A_Q=A_V$ is for simplicity and can totally be generalized to the setting where they share only the learnable matrix $A$. In particular, we can reformulate those matrices as $A_Q=W_{Q,1}A, A_V=W_{V,1}A$ for the simple linear reparametrization and as $A_Q=\sigma_1(W_{Q,1}A), A_V=\sigma_1(W_{V,1}A)$ for the non-linear reparametrization. Tailored to that setting, we need to add several additional terms involving parameters $W_{Q,1}$ and $W_{V,1}$ rather than merely parameters $W_{1}$ as in the current setting, making the convergence analysis unnecessarily complicated. Therefore, we assume without loss of generalization that $A_{Q}=A_{V}=W_{1}A$ or $A_{Q}=A_{V}=\sigma_1(W_{1}A)$ in Section 4.2 to simplify the analysis, making it more accessible.
>
> + **Regarding the experiment tying $A_Q$ and $A_V$:** To further support our theoretical findings, we conduct experiments on FGVC datasets by tying $A_Q$ with $A_V$ and $B_Q$ with $B_V$ in RepLoRA. The results, summarized in the table below, show that tying $A_Q$ and $A_V$ slightly reduces the model's expressiveness, resulting in a modest drop in performance compared to the original RepLoRA with untied matrices. However, tied RepLoRA still significantly outperforms vanilla LoRA, which reinforces the practical value of our approach, even under constrained parameterization.
>
>
> | Methods | CUB-200-2011 | NABirds | Oxford Flowers | Stanford Dogs | Stanford Cars | AVG |
> | -------- | -------- | -------- | -------- | -------- | -------- | -------- |
> | LoRA     | 84.6     | 78.2     |98.9     | 85.1     | 77.1     |84.8     |
> | RepLoRA (tied) | 87.2 | 83.8 |99.0     | 85.6     | 85.4     |88.9     |
> | RepLoRA (untied) | **89.1** | **86.1** | **99.3** | **91.2** | **87.6** | **90.7**  |
>
> + **Regarding the implications of Theorem 4.1:** Thanks for your comment. We would like to emphasize that the results in lines 187-196 (right column) are consequences of Theorem 4.1 in our paper. In particular, by combining the result of Theorem 4.1 and the formulation of the Voronoi loss $\mathcal{D}_{1,r}$ in lines 167-173 (right column), we deduce the convergence rates of low-rank matrices estimation are slower than polynomial rates $O(n^{1-/2r})$ for all $r\geq 1$, where $n$ is the sample size. Due to the inequality $\log(n)<n$, these rates are even slower than the order $O(1/\log^{\tau}(n))$ for some positive constant $\tau$. As a result, to achieve a given approximation error $\epsilon=O(1/\log^{\tau}(n))$ of estimating the low-rank matrices, we need exponentially many data points $O(\exp(\epsilon^{-1/\tau}))$. We will consider formulating those implications into a corollary following Theorem 4.1 in the revision of our manuscript.
>
>
> + **Regarding the activation function**: The nonlinearity was implemented using the sigmoid function. We appreciate the reviewer’s suggestion and will include this detail in the final revision.

---

> > ### Comment · Reviewer_8a5d · 2025-04-03
> >
> > Thank you for the response! I really appreciate the new parameter tying experiments.
> >
> > Since a main message of this paper is the proposed affine reparametrization leads to better sample efficiency, I still think Thms 4.2 and 4.3 have to be modified to untie $A_Q$ and $A_V$, and similarly for $B_Q$ and $B_V$. The full proof can appear in the appendix in case there are space issues.
> >
> > If the modification is too difficult, at the very least the authors should show a counter example where $\{A, B\}_Q$, $\{A, B\}_V$ are tied but sample complexity is still super polynomial.

---

> > > ### Author Response · Authors · 2025-04-03
> > >
> > > Dear Reviewer 8a5d,
> > >
> > > Thank you so much for your response, which we really appreciate. **We would like to confirm that the results of Theorem 4.2 and Theorem 4.3 can totally be generalized to the setting where $A_Q,A_V$ are untied and $B_Q,B_V$ are untied without any technical issues**. Under that scenario, these matrices are formulated as
> > >
> > > $A_Q=\sigma_1(W_{Q,1}A), A_V=\sigma_1(W_{V,1}A)$
> > >
> > > and
> > >
> > > $B_Q=\sigma_2(W_{Q,2}B), B_V=\sigma_2(W_{V,2}B)$
> > >
> > > for the non-linear reparametrization setting. In response to these formulation changes, it is necessary to modify the Voronoi loss function $D_3(\tilde{G},\tilde{G}_{\ast})$ defined in lines 277-287 as *(we have to break down the loss formulation as the tex compiler does not allow us to type in full, we are really sorry for this inconvenience)*
> > >
> > > $D_3(\tilde{G},\tilde{G}_{\ast})=$
> > >
> > > $\sum_{j=1}^{L}\Big|\sum_{i\in\mathcal{V}_j}\exp(c_i)-\exp(c^{\ast}_j)\Big|$
> > >
> > > $+\sum_{j:|\mathcal{V}_j|=1, i\in\mathcal{V}_j}\exp(c_i)$
> > >
> > > $\Big(||\Delta (W_{V,2}B)_{ij}||$
> > >
> > > $+||\Delta (W_{Q,1}A)_{ij}||$
> > >
> > > $+||\Delta (W_{V,2}B)_{ij}|| $
> > >
> > > $+||\Delta (W_{V,1}A)_{ij}\|\Big)$
> > >
> > > $+\sum_{j:|\mathcal{V}_j|>1, i\in\mathcal{V}_j}\exp(c_i)$
> > >
> > > $\Big(||\Delta (W_{V,2}B)_{ij}||^2$
> > >
> > > $+||\Delta (W_{Q,1}A)_{ij}||^2$
> > >
> > > $+||\Delta (W_{V,2}B)_{ij}||^2$
> > >
> > > $+||\Delta (W_{V,1}A)_{ij}\|^2\Big).$
> > >
> > > Then, by employing the same arguments as in Appendix A.3, we obtain the estimation rates of the low-rank matrices through the bound $D_3(\tilde{G}n,\tilde{G}_{\ast})$ as in Theorem 4.3. It can be seen that the convergence behavior of the low-rank matrix estimation remains unchanged compared to that in the current paper. Additionally, the result for the simple linear reparametrization (Theorem 4.2) can also be generalized analogously. Therefore, we simplify the presentation of the convergence analysis by tying $A_Q$ and $A_V$, $B_Q$ and $B_V$, which helps reduce several inessential terms in the Voronoi loss function. However, **as per your suggestion, we will consider modifying the settings of Theorem 4.2 and Theorem 4.3 as above and including respective proofs in the revision of our manuscript.**
> > >
> > > Thank you,
> > >
> > > The Authors

---

### Official Review · Reviewer_yfWw · 2025-03-14

**Overall Recommendation:** 3

**Summary:**

This paper proposes RepLoRA, a method that reparameterizes the low-rank matrices of LoRA using a lightweight MLP. RepLoRA surpasses baseline LoRA by up to 40.0% and achieves similar results with baseline with only 30.0% of the training data. Additionally, this work provides a theoretical analysis of LoRA from the perspective of a mixture of experts, demonstrating that reparameterization can reduce the data needed to achieve a desired estimation error from an exponential scale to a polynomial scale. Experiments across various tasks, including language (commonsense reasoning), image (classification), video (video action recognition), and multi-modal (image/video-text understanding), demonstrate the effectiveness of the proposed method.

**Claims And Evidence:**

Most of the claims are supported by cited works.

**Essential References Not Discussed:**

Most of the relevant related works have already been cited.

**Experimental Designs Or Analyses:**

The experiments robustly demonstrate the effectiveness of the proposed method.

**Methods And Evaluation Criteria:**

The proposed method is effective and the evaluation criteria are valid.

**Other Comments Or Suggestions:**

see Strengths And Weaknesses

**Other Strengths And Weaknesses:**

Strengths
1.This work provides an insightful analysis of the impact of LoRA on multi-head self-attention layers from the perspective of a mixture of experts (MoE), offering valuable inspiration.
2.The proposed RepLoRA achieves outstanding results across various tasks, including commonsense reasoning, image classification, video action recognition, and image/video-text understanding.

Weaknesses
1.It would be better to provide experimental analysis for the theoritical proofs, such as the convergence curve w./w.o reparametrization.
2.Lack of comparison with similar works, such as [a,b,c]
[a] DoRA: Weight-Decomposed Low-Rank Adaptation
[b] VERA: VECTOR-BASED RANDOM MATRIX ADAPTATION.
[c] LORA-FA: MEMORY-EFFICIENT LOW-RANK ADAPTATION FOR LARGE LANGUAGE MODELS FINE-TUNING

**Questions For Authors:**

see Strengths And Weaknesses

**Relation To Broader Scientific Literature:**

Low-rank Adaptation (LoRA) has gained significant traction as a method for fine-tuning large-scale foundation models, yet its theoretical underpinnings have remained relatively unexplored. This paper contributes to the broader scientific literature by providing a theoretical analysis of LoRA through its connection to Mixture of Experts models. By situating LoRA within this framework, they demonstrate that simple reparameterizations of LoRA matrices can significantly expedite the low-rank matrix estimation process. Specifically, the findings show that reparameterization can reduce the data required to achieve a desired estimation error from an exponential to a polynomial scale, thereby enhancing sample efficiency.

**Theoretical Claims:**

I cannot verify the correctness of the proofs due to my non-mathematical background. Please refer to other reviewers for the correctness check.

---

> ### Author Rebuttal · Authors · 2025-03-29
>
> We sincerely thank the reviewer for the constructive feedback and would like to address your concerns as follows:
>
> **Regarding the analysis of the theoretical results:** Our theoretical analysis demonstrates that LoRA with reparameterization offers superior *sample efficiency* compared to LoRA without reparameterization. To empirically analyze this theoretical claim, we dedicated the final experiment in the experimental section to validate these theoretical findings. The results show that reparameterization in LoRA significantly improves *sample efficiency*, as illustrated in Figure 2.
>
> **Regarding the comparison with related works:** Following the reviewer’s suggestion, we have included additional comparisons with VeRA [1] and DoRA [2] on the image classification task using the FGVC dataset, as detailed below:
>
>
> | Methods | CUB-200-2011 | NABirds | Oxford Flowers | Stanford Dogs | Stanford Cars | AVG | PPT |
> | -------- | -------- | -------- | -------- | -------- | -------- | -------- | -------- |
> | LoRA     | 84.6     | 78.2     |98.9     | 85.1     | 77.1     |84.8     | 0.82 |
> | DoRA     | 87.3     | 80.0     |99.1     | 87.6     | 81.9     |87.2     |0.88 |
> | VeRA     | 85.1     | 79.2     |97.4     | 87.3     | 76.3     |85.1     |0.88 |
> | RepLoRA     | **89.1**     | **86.1**     | **99.3**    | **91.2**     | **87.6**     | **90.7**     |**0.90** |
>
> The results demonstrate that RepLoRA outperforms both LoRA variants by large margins, emphasizing its practical advantages. In response to the reviewer’s suggestion, we will include this finding in the final revision.
>
> **References**
>
> [1] VeRA: Vector-based Random Matrix Adaptation. ICLR. 2024
>
> [2] DoRA: Weight-Decomposed Low-Rank Adaptation. ICML. 2024

---

### Official Review · Reviewer_zZoN · 2025-03-14

**Overall Recommendation:** 3

**Summary:**

The apper combines LoRA into multi-head parts of MSA and treats different heads as experts to build a mixtural of experts. In addition, authors use lightweight MLP to conduct reparameter opertations, which improves sampling efficiency while reducing data requirements compared to the original LoRA.

**Claims And Evidence:**

The claims in the submission are generally well-supported.

**Essential References Not Discussed:**

None

**Experimental Designs Or Analyses:**

The experimental designs are comprehensive, with LLama7B/13B baselines and extensive datasets. By experimental analysis, the proposed methods are verified in both aspects in performance and parameters.

**Methods And Evaluation Criteria:**

The proposed method offers a novel insight, reducing  the data needed to achieve a desired estimation error from an exponential
scale to a polynomial scale.

+ Authors measure classification accuracy trends at different training scales.
+ Linear and nonlinear RepLoRA modules are ablated on 7 application scenarios.
+ Based on the baseline LLama7B/13B, the proposed methods are verified to be effective on multiple datasets

**Other Comments Or Suggestions:**

The proposed method is a general method, but the current work focuses on verifying it in combination with LoRA on LLama. If possible, more experiments are better.

**Other Strengths And Weaknesses:**

+ Strengths
1) The proposed framework serves as a theoretical foundation that underpins the various methodologies and processes within LoRA scope.
2) Authors measure classification accuracy trends at different training scales.
3) Authors perform experiments on multiple scenarios for linear and non-linear RepLoRA modules.
4) LLama7B/13B are classical LLMs. The proposed methods utilize  LLama7B/13B to verified to be effective on multiple datasets


- Weaknesses
1) The experimental situation with more LoRA adapters is unknown.
2) Should compare LoRA with MoE works [1, 2 , 3]

[1] Mixture-of-loras: An efficient multitask tuning for large language models. COLING. 2024.

[2] Mixture-of-subspaces in low-rank adaptation. EMNLP. 2024.

[3] MoR: Mixture of Ranks for Low-Rank Adaptation Tuning. Arxiv. 2024

**Questions For Authors:**

1) Please see weaknesses.
2) The cost of training or inference is unclear, such as GPU memory usage, training/inference time, and flops.

**Relation To Broader Scientific Literature:**

This paper is closely related to the relevant research in fields of LoRA, MOE, and PEFT. If possible, authors can further illustrate the related mathematical basis/thinking in scientific literature.

**Theoretical Claims:**

Theoretical proofs are provided. Based on it, authors discuss LoRA with MoE for achieving optimal sampling. Furthermore, RepLoRA was proposed with an effective and efficient approach to PEFT.

---

> ### Author Rebuttal · Authors · 2025-03-29
>
> We appreciate the reviewer’s insightful feedback. In response to their concern, we’ve expanded our analysis to include additional comparisons with three LoRA adapters: VeRA [1], DoRA [2], and MoR [3], as suggested by the reviewer. These experiments were carried out on the image classification task using the FGVC datasets. For MoR, we specifically report results with 8 experts. The detailed results are presented below:
>
> | Method | CUB-200-2011 | NABirds | Oxford Flowers | Stanford Dogs | Stanford Cars | AVG  | PPT |
> | -------- | -------- | -------- | -------- | -------- | -------- | -------- | -------- |
> | LoRA     | 84.6     | 78.2     |98.9     | 85.1     | 77.1     |84.8     |0.82 |
> | DoRA     | 87.3     | 80.0     |99.1     | 87.6     | 81.9     |87.2     |0.88 |
> | VeRA     | 85.1     | 79.2     |97.4     | 87.3     | 76.3     |85.1     |0.88 |
> | MoR     |  87.6    |  82.5    | 99.3    |  89.7    |   84.7   |  88.8    | 0.89 |
> | RepLoRA     | **89.1**     | **86.1**     | **99.3**    | **91.2**     | **87.6**     | **90.7**     |**0.90** |
>
> Following the reviewer’s suggestion, we will incorporate these results into the final revision.
>
> When it comes to training, RepLoRA introduces only a minimal increase in parameters compared to LoRA, as seen in the number of parameters and PPT. As a result, RepLoRA does not introduce any significant additional training time, FLOPs, or memory usage when compared to LoRA. Additionally, as highlighted in the main text, the reparameterization matrices can be discarded during inference, making the inference process identical to that of LoRA.
>
> **References**
>
> [1] VeRA: Vector-based Random Matrix Adaptation. ICLR. 2024
>
> [2] DoRA: Weight-Decomposed Low-Rank Adaptation. ICML. 2024
>
> [3] MoR: Mixture of Ranks for Low-Rank Adaptation Tuning. Arxiv. 2024

---

> > ### Comment · Reviewer_zZoN · 2025-04-03
> >
> > Most of my doubts have been cleared. Can reparameterization be used in other LoRA works to alleviate their suboptimal rate for low-rank matrix estimation?
> >
> > Finally, I cannot offer any further advice on the mathematical derivation and analysis, so I keep my score.

---

> > > ### Author Response · Authors · 2025-04-03
> > >
> > > Dear Reviewer zZoN,
> > >
> > > We would like to thank you for your response and for maintaining the positive rating of 3, which we really appreciate.
> > >
> > > **Regarding the reparametrization for LoRA variants:** We have shown in this work that the convergence rates of low-rank matrix estimations in the original LoRA method [1] are suboptimal. Therefore, we propose the reparametrization strategy for LoRA based on its connection to Mixture-of-Experts (MoE) for improving the low-rank matrix estimation rates. On the other hand, there have not been any work analyzing that the convergence behavior of low-rank matrix estimations in other LoRA variants, namely VeRA [2] and DoRA [3], are suboptimal. However, if these rates were still suboptimal and VeRA or DoRA can be also linked to MoE in a similar fashion to the case of LoRA, then we believe that the reparametrization method would help alleviate the supoptimal rate for estimating low-rank matrices in VeRA and DoRA. Since this direction stays beyond the scope of our work, we leave it for future development.
> > >
> > > **References**
> > >
> > > [1] LoRA: Low-Rank Adaptation of Large Language Models. ICLR, 2022
> > >
> > > [2] VeRA: Vector-based Random Matrix Adaptation. ICLR, 2024
> > >
> > > [3] DoRA: Weight-Decomposed Low-Rank Adaptation. ICML, 2024

---

### Decision · Program_Chairs · 2025-05-01

**Decision:**

Accept (poster)

**Comment:**

This paper presents a compelling perspective on rethinking LoRA as a specific mechanism of Mixture of Experts (MoE) with a reparameterization scheme. All the reviewers have provided positive feedback regarding this paper. During the rebuttal phase, the authors offered additional justifications and experiments to support their methodology. Overall, I recommend accepting this paper. However, I strongly encourage the authors to incorporate their responses and any relevant discussions into the final manuscript.